# Would I have gotten that reward? Long-term credit assignment by counterfactual contribution analysis

**Alexander Meulemans**[*1]**, Simon Schug**[*1]**, Seijin Kobayashi**[*1]
**Nathaniel D Daw**[2,3,4]**, Gregory Wayne**[2]

[1]Department of Computer Science, ETH Zürich
[2]Google DeepMind
[3]Princeton Neuroscience Institute, Princeton University
[4]Department of Psychology, Princeton University
`{ameulema, sschug, seijink}@ethz.ch`

## Abstract

To make reinforcement learning more sample efficient, we need better credit assignment methods that measure an action's influence on future rewards. Building upon Hindsight Credit Assignment (HCA) [1], we introduce Counterfactual Contribution Analysis (COCOA), a new family of model-based credit assignment algorithms. Our algorithms achieve precise credit assignment by measuring the contribution of actions upon obtaining subsequent rewards, by quantifying a counterfactual query: 'Would the agent still have reached this reward if it had taken another action?'. We show that measuring contributions w.r.t. rewarding *states*, as is done in HCA, results in spurious estimates of contributions, causing HCA to degrade towards the high-variance REINFORCE estimator in many relevant environments. Instead, we measure contributions w.r.t. rewards or learned representations of the rewarding objects, resulting in gradient estimates with lower variance. We run experiments on a suite of problems specifically designed to evaluate long-term credit assignment capabilities. By using dynamic programming, we measure ground-truth policy gradients and show that the improved performance of our new model-based credit assignment methods is due to lower bias and variance compared to HCA and common baselines. Our results demonstrate how modeling action contributions towards rewarding outcomes can be leveraged for credit assignment, opening a new path towards sample-efficient reinforcement learning.[2]

## 1   Introduction

Reinforcement learning (RL) faces two central challenges: exploration and credit assignment [2]. We need to explore to discover rewards and we need to reinforce the actions that are instrumental for obtaining these rewards. Here, we focus on the credit assignment problem and the intimately linked problem of estimating policy gradients. For long time horizons, obtaining the latter is notoriously difficult as it requires measuring how each action influences expected subsequent rewards. As the number of possible trajectories grows exponentially with time, future rewards come with a considerable variance stemming from stochasticity in the environment itself and from the stochasticity of interdependent future actions leading to vastly different returns [3–5].

Monte Carlo estimators such as REINFORCE [6] therefore suffer from high variance, even after variance reduction techniques like subtracting a baseline [6–9]. Similarly, in Temporal Difference methods such as Q-learning, this high variance in future rewards results in a high bias in the value estimates, requiring exponentially many updates to correct for it [5]. Thus, a common technique to

---

[*]Equal contribution; ordering determined by coin flip.
[2]Code available at `https://github.com/seijin-kobayashi/cocoa`

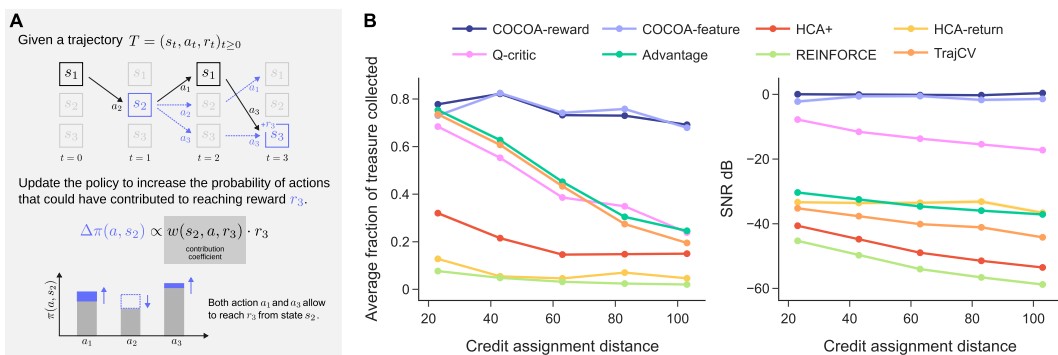

Figure 1: **Counterfactual Contribution Analysis enables long-term credit assignment.** (A) Given a sample trajectory that eventually results in a rewarding outcome, we estimate the policy gradient by considering the contribution of actions along the trajectory towards arriving at a rewarding outcome. In this example, we measure how much more likely the rewarding outcome with reward $r_3$ is when following action $a_1$ versus the counterfactual actions $a_2$ and $a_3$ in state $s_2$. This is quantified through the contribution coefficient $w(s_2, a_1, r_3)$ which is used to update all possible action probabilities of the policy $\pi(a \mid s_2)$. (B) In the linear key-to-door environment increasing the distance between picking up the key and opening the door that leads to reward necessitates credit assignment over increasing time spans. COCOA consistently achieves good performance (left) compared to HCA and baselines which deteriorate when increasing the distance between an action and the resulting rewarding outcome. This is reflected in a higher signal-to-noise ratio of the policy gradient estimator of COCOA compared to baselines (right).

reduce variance and bias is to discount rewards that are far away in time resulting in a biased estimator which ignores long-term dependencies [10–13]. Impressive results have nevertheless been achieved in complex environments [14–16] at the cost of requiring billions of environment interactions, making these approaches sample inefficient.

Especially in settings where obtaining such large quantities of data is costly or simply not possible, model-based RL that aims to simulate the dynamics of the environment is a promising alternative. While learning such a world model is a difficult problem by itself, when successful it can be used to generate a large quantity of synthetic environment interactions. Typically, this synthetic data is combined with model-free methods to improve the action policy [17–19]. A notable exception to simply using world models to generate more data are the Stochastic Value Gradient method [20] and the closely related Dreamer algorithms [21–23]. These methods perform credit assignment by backpropagating policy gradients through the world model. Crucially, this approach only works for environments with a continuous state-action space, as otherwise sensitivities of the value with respect to past actions are undefined [20, 22, 24]. Intuitively, we cannot compute sensitivities of discrete choices such as a yes / no decision as the agent cannot decide 'yes' a little bit more or less.

Building upon Hindsight Credit Assignment (HCA) [1], we develop Counterfactual Contribution Analysis (COCOA), a family of algorithms that use models for credit assignment compatible with discrete actions. We measure the *contribution* of an action upon subsequent rewards by asking a counterfactual question: 'would the agent still have achieved the rewarding outcome, if it had taken another action?' (c.f. Fig. 1A). We show that measuring contributions towards achieving a future *state*, as is proposed in HCA, leads to spurious contributions that do not reflect a contribution towards a reward. This causes HCA to degrade towards the high-variance REINFORCE method in most environments. Instead, we propose to measure contributions directly on rewarding outcomes and we develop various new ways of learning these contributions from observations. The resulting algorithm differs from value-based methods in that it measures the contribution of an action to individual rewards, instead of estimating the full expected sum of rewards. This crucial difference allows our contribution analysis to disentangle different tasks and ignore uncontrollable environment influences, leading to a gradient estimator capable of long-term credit assignment (c.f. Fig. 1B). We introduce a new method for analyzing policy gradient estimators which uses dynamic programming to allow comparing to ground-truth policy gradients. We leverage this to perform a detailed bias-variance analysis of all proposed methods and baselines showing that our new model-based credit assignment algorithms achieve low variance and bias, translating into improved performance (c.f. Fig. 1C).

## 2 Background and notation

We consider an undiscounted Markov decision process (MDP) defined as the tuple $(\mathcal{S}, \mathcal{A}, p, p_r)$, with $\mathcal{S}$ the state space, $\mathcal{A}$ the action space, $p(S_{t+1} \mid S_t, A_t)$ the state-transition distribution and $p_r(R \mid S, A)$ the reward distribution with bounded reward values $r$. We use capital letters for random variables and lowercase letters for the values they take. The policy $\pi(A \mid S)$, parameterized by $\theta$, denotes the probability of taking action $A$ at state $S$. We consider an undiscounted infinite-horizon setting with a zero-reward absorbing state $s_\infty$ that the agent eventually reaches: $\lim_{t \to \infty} p(S_t = s_\infty) = 1$. Both the discounted and episodic RL settings are special cases of this setting. (c.f. App. B), and hence all theoretical results proposed in this work can be readily applied to both (c.f. App J).

We use $\mathcal{T}(s, \pi)$ and $\mathcal{T}(s, a, \pi)$ as the distribution over trajectories $T = (S_t, A_t, R_t)_{t \geq 0}$ starting from $S_0 = s$ and $(S_0, A_0) = (s, a)$ respectively, and define the return $Z_t = \sum_{t=0}^{\infty} R_t$. The value function $V^\pi(s) = \mathbb{E}_{T \sim \mathcal{T}(s,\pi)}[Z_t]$ and action value function $Q^\pi(s, a) = \mathbb{E}_{T \sim \mathcal{T}(s,a,\pi)}[Z_t]$ are the expected return when starting from state $s$, or state $s$ and action $a$ respectively. Note that these infinite sums have finite values due to the absorbing zero-reward state (c.f. App. B).

The objective of reinforcement learning is to maximize the expected return $V^\pi(s_0)$, where we assume the agent starts from a fixed state $s_0$. Policy gradient algorithms optimize $V^\pi(s_0)$ by repeatedly estimating its gradient $\nabla_\theta V(s_0)$ w.r.t. the policy parameters. REINFORCE [6] (c.f. Tab. 1) is the canonical policy gradient estimator, however, it has a high variance resulting in poor parameter updates. Common techniques to reduce the variance are (i) subtracting a baseline, typically a value estimate, from the sum of future rewards [2, 25] (c.f. 'Advantage' in Tab. 1); (ii) replacing the sum of future rewards with a learned action value function $Q$ [2, 3, 25, 26] (c.f. 'Q-critic' in Tab. 1); and (iii) using temporal discounting. Note that instead of using a discounted formulation of MDPs, we treat the discount factor as a variance reduction technique in the undiscounted problem [10, 11, 13] as this more accurately reflects its practical use [4, 27]. Rearranging the summations of REINFORCE with discounting lets us interpret temporal discounting as a credit assignment heuristic, where for each reward, past actions are reinforced proportional to their proximity in time.

$$\hat{\nabla}_\theta^{\text{REINFORCE},\gamma} V^\pi(s_0) = \sum_{t \geq 0} R_t \sum_{k \leq t} \gamma^{t-k} \nabla_\theta \log \pi(A_k \mid S_k), \quad \gamma \in [0, 1]. \tag{1}$$

Crucially, long-term dependencies between actions and rewards are exponentially suppressed, thereby reducing variance at the cost of disabling long-term credit assignment [4, 28]. The aim of this work is to replace the heuristic of time discounting by principled *contribution coefficients* quantifying how much an action contributed towards achieving a reward, and thereby introducing new policy gradient estimators with reduced variance, without jeopardizing long-term credit assignment.

HCA [1] makes an important step in this direction by introducing a new gradient estimator:

$$\hat{\nabla}_\theta^{\text{HCA}} V^\pi = \sum_{t \geq 0} \sum_{a \in \mathcal{A}} \nabla_\theta \pi(a \mid S_t) \Big( r(S_t, a) + \sum_{k \geq 1} \frac{p^\pi(A_t = a \mid S_t = s, S' = S_{t+k})}{\pi(a \mid S_t)} R_{t+k} \Big) \tag{2}$$

with $r(s, a)$ a reward model, and the *hindsight* ratio $\frac{p^\pi(a \mid S_t = s, S' = S_{t+k})}{\pi(a \mid S_t)}$ measuring how important action $a$ was to reach the state $S'$ at some point in the future. Although the hindsight ratio delivers precise credit assignment w.r.t. reaching future states, it has a failure mode of practical importance, creating the need for an updated theory of model-based credit assignment which we will detail in the next section.

## 3 Counterfactual Contribution Analysis

To formalize the 'contribution' of an action upon subsequent rewards, we generalize the theory of HCA [1] to measure contributions on *rewarding outcomes* instead of states. We introduce unbiased policy gradient estimators that use these contribution measures, and show that HCA suffers from high variance, making the generalization towards rewarding outcomes crucial for obtaining low-variance estimators. Finally, we show how we can estimate contributions using observational data.

### 3.1 Counterfactual contribution coefficients

To assess the contribution of actions towards rewarding outcomes, we propose to use counterfactual reasoning: 'how does taking action $a$ influence the probability of obtaining a rewarding outcome, compared to taking alternative actions $a'$?'.

Table 1: Comparison of policy gradient estimators.

| Method | Policy gradient estimator ($\hat{\nabla}_\theta V^\pi(s_0)$) |
|---|---|
| REINFORCE | $\sum_{t\geq 0} \nabla_\theta \log \pi(A_t \mid S_t) \sum_{k\geq 0} R_{t+k}$ |
| Advantage | $\sum_{t\geq 0} \nabla_\theta \log \pi(A_t \mid S_t) \left( \sum_{k\geq 0} R_{t+k} - V(S_t) \right)$ |
| Q-critic | $\sum_{t\geq 0} \sum_{a\in\mathcal{A}} \nabla_\theta \pi(a \mid S_t) Q(S_t, a)$ |
| HCA-Return | $\sum_{t\geq 0} \nabla_\theta \log \pi(A_t \mid S_t) \left( 1 - \frac{\pi(A_t\mid S_t)}{p^\pi(A_t\mid S_t, Z_t)} \right) Z_t$ |
| TrajCV | $\sum_{t\geq 0} \nabla_\theta \log \pi(A_t \mid S_t) \left( Z_t - Q(A_t, S_t) - \sum_{t'>t} (Q(S_{t'}, A_{t'}) - V(S_{t'})) \right) + \ldots$ $\ldots \sum_{a\in\mathcal{A}} \nabla_\theta \pi(a \mid S_t) Q(S_t, a)$ |
| COCOA | $\sum_{t\geq 0} \nabla_\theta \log \pi(A_t \mid S_t) R_t + \sum_{a\in\mathcal{A}} \nabla_\theta \pi(a \mid S_t) \sum_{k\geq 1} w(S_t, a, U_{t+k}) R_{t+k}$ |
| HCA+ | $\sum_{t\geq 0} \nabla_\theta \log \pi(A_t \mid S_t) R_t + \sum_{a\in\mathcal{A}} \nabla_\theta \pi(a \mid S_t) \sum_{k\geq 1} w(S_t, a, S_{t+k}) R_{t+k}$ |

**Definition 1** (Rewarding outcome). A rewarding outcome $U' \sim p(U' \mid s', a', r')$ is a probabilistic encoding of the state-action-reward triplet.

If action $a$ contributed towards the reward, the probability of obtaining the rewarding outcome following action $a$ should be higher compared to taking alternative actions. We quantify the contribution of action $a$ taken in state $s$ upon rewarding outcome $u'$ as

$$w(s, a, u') = \frac{\sum_{k\geq 1} p^\pi(U_{t+k} = u' \mid S_t = s, A_t = a)}{\sum_{k\geq 1} p^\pi(U_{t+k} = u' \mid S_t = s)} - 1 = \frac{p^\pi(A_t = a \mid S_t = s, U' = u')}{\pi(a \mid s)} - 1 \quad (3)$$

From a given state, we compare the probability of reaching the *rewarding outcome* $u'$ at any subsequent point in time, given we take action $a$ versus taking counterfactual actions according to the policy $\pi$, as $p^\pi(U_{t+k} = u' \mid S_t = s) = \sum_{a'} \pi(a' \mid s) p^\pi(U_{t+k} = u' \mid S_t = s, A_t = a')$. Subtracting this ratio by one results in an intuitive interpretation of the *contribution coefficient* $w(s, a, u')$: if the coefficient is positive/negative, performing action $a$ results in a higher/lower probability of obtaining rewarding outcomes $u'$, compared to following the policy $\pi$. Using Bayes' rule, we can convert the counterfactual formulation of the contribution coefficients into an equivalent *hindsight* formulation (right-hand side of Eq. 3), where the hindsight distribution $p^\pi(A_t = a \mid S_t = s, U' = u')$ reflects the probability of taking action $a$ in state $s$, given that we encounter the rewarding outcome $u'$ at *some future point in time*. We refer the reader to App. C for a full derivation.

**Choice of rewarding outcome.** For $u' = s'$, we recover state-based HCA [1].[3] In the following, we show that a better choice is to use $u' = r'$, or an encoding $p(u' \mid s', a')$ of the underlying object that causes the reward. Both options lead to gradient estimators with lower variance (c.f. Section 3.3), while using the latter becomes crucial when different underlying rewarding objects have the same scalar reward (c.f. Section 4).

### 3.2 Policy gradient estimators

We now show how the contribution coefficients can be used to learn a policy. Building upon HCA [1], we propose the Counterfactual Contribution Analysis (COCOA) policy gradient estimator

$$\hat{\nabla}_\theta^U V^\pi(s_0) = \sum_{t\geq 0} \nabla_\theta \log \pi(A_t \mid S_t) R_t + \sum_{a\in\mathcal{A}} \nabla_\theta \pi(a \mid S_t) \sum_{k\geq 1} w(S_t, a, U_{t+k}) R_{t+k}. \quad (4)$$

When comparing to the discounted policy gradient of Eq. 1, we see that the temporal discount factors are substituted by the contribution coefficients, replacing the time heuristic with fine-grained credit assignment. Importantly, the contribution coefficients enable us to evaluate all counterfactual actions instead of only the observed ones, further increasing the quality of the gradient estimator (c.f. Fig. 1A). The contribution coefficients allow for various different gradient estimators (c.f. App. C). For example, independent action samples can replace the sum over all actions, making it applicable to large action spaces. Here, we use the gradient estimator of Eq. 4, as our experiments consist of small action spaces where enumerating all actions is feasible. When $U = S$, the above estimator is almost equivalent to

---

[3]Note that Harutyunyan et al. [1] also introduce an estimator based on the return. Here, we focus on the state-based variant as only this variant uses contribution coefficients to compute the policy gradient (Eq. 4). The return-based variant instead uses the hindsight distribution as an action-dependent baseline as shown in Tab. 1. Importantly, the return-based estimator is biased in many relevant environments (c.f. Appendix L).

the state-based HCA estimator of Eq. 2, except that it does not need a learned reward model $r(s, a)$. We use the notation HCA+ to refer to this simplified version of the HCA estimator. Theorem 1 below shows that the COCOA gradient estimator is unbiased, as long as the encoding $U$ is fully predictive of the reward, thereby generalizing the results of Harutyunyan et al. [1] to arbitrary rewarding outcome encodings.

**Definition 2** (Fully predictive). A rewarding outcome $U$ is fully predictive of the reward $R$, if the following conditional independence condition holds for all $k \geq 0$: $p^\pi(R_k = r \mid S_0 = s, A_0 = a, U_k = u) = p^\pi(R = r \mid U = u)$, where the right-hand side does not depend on the time $k$.

**Theorem 1.** *Assuming that $U$ is fully predictive of the reward (c.f. Definition 2), the COCOA policy gradient estimator $\hat{\nabla}_\theta^U V^\pi(s_0)$ is unbiased, when using the ground-truth contribution coefficients of Eq. 3, that is*

$$\nabla_\theta V^\pi(s_0) = \mathbb{E}_{T \sim \mathcal{T}(s_0, \pi)} \hat{\nabla}_\theta^U V^\pi(s_0).$$

### 3.3 Optimal rewarding outcome encoding for low-variance gradient estimators

Theorem 1 shows that the COCOA gradient estimators are unbiased for all rewarding outcome encodings $U$ that are fully predictive of the reward. The difference between specific rewarding outcome encodings manifests itself in the variance of the resulting gradient estimator. Proposition 2 shows that for $U' = S'$ as chosen by HCA there are many cases where the variance of the resulting policy gradient estimator degrades to the high-variance REINFORCE estimator [6]:

**Proposition 2.** *In environments where each action sequence leads deterministically to a different state, we have that the HCA+ estimator is equal to the REINFORCE estimator (c.f. Tab. 1).*

In other words, when all previous actions can be perfectly decoded from a given state, they trivially all contribute to reaching this state. The proof of Propostion 2 follows immediately from observing that $p^\pi(a \mid s, s') = 1$ for actions $a$ along the observed trajectory, and zero otherwise. Substituting this expression into the contribution analysis gradient estimator (4) recovers REINFORCE. A more general issue underlies this special case: State representations need to contain detailed features to allow for a capable policy but the same level of detail is detrimental when assigning credit to actions for reaching a particular state since at some resolution almost every action will lead to a slightly different outcome. Measuring the contribution towards reaching a specific state ignores that the same rewarding outcome could be reached in slightly different states, hence overvaluing the importance of previous actions and resulting in *spurious contributions*. Many commonly used environments, such as pixel-based environments, continuous environments, and partially observable MDPs exhibit this property to a large extent due to their fine-grained state representations (c.f. App. G). Hence, our generalization of HCA to rewarding outcomes is a crucial step towards obtaining practical low-variance gradient estimators with model-based credit assignment.

**Using rewards as rewarding outcomes yields lowest-variance estimators.** The following Theorem 3 shows in a simplified setting that (i) the variance of the REINFORCE estimator is an upper bound on the variance of the COCOA estimator, and (ii) the variance of the COCOA estimator is smaller for rewarding outcome encodings $U$ that contain less information about prior actions. We formalize this with the conditional independence relation of Definition 2 by replacing $R$ with $U'$: encoding $U$ contains less or equal information than encoding $U'$, if $U'$ is fully predictive of $U$. Combined with Theorem 1 that states that an encoding $U$ needs to be fully predictive of the reward $R$, we have that taking the reward $R$ as our rewarding outcome encoding $U$ results in the gradient estimator with the lowest variance of the COCOA family.

**Theorem 3.** *Consider an MDP where only the states at a single (final) time step contain a reward, and where we optimize the policy only at a single (initial) time step. Furthermore, consider two rewarding outcome encodings $U$ and $U'$, where $S$ is fully predictive of $U'$, $U'$ fully predictive of $U$, and $U$ fully predictive of $R$. Then, the following relation holds between the policy gradient estimators:*

$$\mathbb{V}[\hat{\nabla}_\theta^R V^\pi(s_0)] \preccurlyeq \mathbb{V}[\hat{\nabla}_\theta^U V^\pi(s_0)] \preccurlyeq \mathbb{V}[\hat{\nabla}_\theta^{U'} V^\pi(s_0)] \preccurlyeq \mathbb{V}[\hat{\nabla}_\theta^S V^\pi(s_0)] \preccurlyeq \mathbb{V}[\hat{\nabla}_\theta^{REINF} V^\pi(s_0)]$$

*with $\hat{\nabla}_\theta^X V^\pi(s_0)$ the COCOA estimator (4) using $U = X$, $\mathbb{V}[Y]$ the covariance matrix of $Y$ and $A \preccurlyeq B$ indicating that $B - A$ is positive semi-definite.*

As Theorem 3 considers a simplified setting, we verify empirically whether the same arguments hold more generally. We construct a tree environment where we control the amount of information a

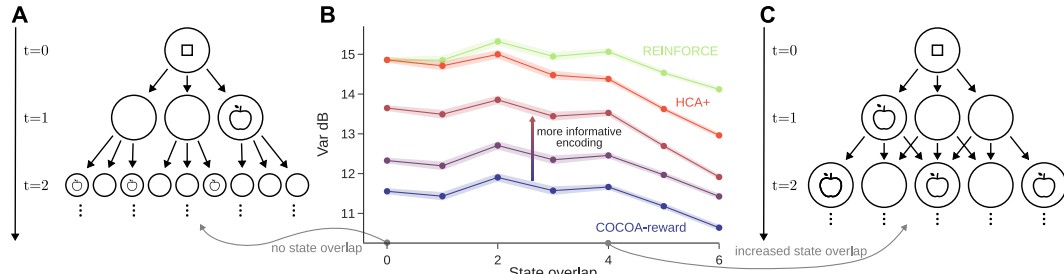

Figure 2: **HCA suffers from spurious contributions which can be alleviated by using less informative rewarding outcome encodings.** (A) and (C): Schematic of the tree environment where we parametrically adjust the amount of overlap between states by varying the amount of shared children of two neighboring nodes. We can decrease the information content of the rewarding outcome encoding $u = f(s, a)$ by grouping state-action pairs that share the same reward value. (B) Normalized variance in dB using ground-truth coefficients and a random uniform policy (shaded region represents standard error over 10 random environments) comparing REINFORCE, HCA, COCOCA-reward and various degrees of intermediate grouping.

state contains about the previous actions by varying the overlap of the children of two neighbouring nodes (c.f. Fig 2), and assign a fixed random reward to each state-action pair. We compute the ground-truth contribution coefficients by leveraging dynamic programming (c.f. Section 4). Fig. 2B shows that the variance of HCA is as high as REINFORCE for zero state overlap, but improves when more states overlap, consistent with Proposition 2 and Theorem 3. To investigate the influence of the information content of $U$ on the variance, we consider rewarding outcome encodings $U$ with increasing information content, which we quantify with how many different values of $u$ belong to the same reward $r$. Fig. 2B shows that by increasing the information content of $U$, we interpolate between the variance of COCOA with $u = r$ and HCA+, consistent with Theorem 3.

**Why do rewarding outcome encodings that contain more information than the reward lead to higher variance?** To provide a better intuition on this question we use the following theorem:

**Theorem 4.** *The policy gradient on the expected number of occurrences $O^\pi(u', s) = \sum_{k \geq 1} p^\pi(U_k = u' \mid S_0 = s)$ is proportional to*

$$\nabla_\theta O^\pi(u', s) \propto \mathbb{E}_{S'' \sim \mathcal{T}(\pi, s)} \sum_{a \in \mathcal{A}} \nabla_\theta \pi(a \mid S'') w(S'', a, u') O^\pi(u', S'')$$

Recall that the COCOA gradient estimator consists of individual terms that credit past actions $a$ at state $s$ for a current reward $r$ encountered in $u$ according to $\sum_{a \in \mathcal{A}} \nabla_\theta \pi(a \mid s) w(s, a, u') r'$ (c.f. Eq. 4). Theorem 4 indicates that each such term aims to increase the average number of times we encounter $u'$ in a trajectory starting from $s$, proportional to the corresponding reward $r'$. If $U' = R'$, this update will correctly make all underlying states with the same reward $r'$ more likely while decreasing the likeliness of all states for which $u' \neq r'$. Now consider the case where our rewarding outcome encoding contains a bit more information, i.e. $U' = f(R', \Delta S')$ where $\Delta S'$ contains a little bit of information about the state. As a result the update will distinguish some states even if they yield the same reward and increase the number of occurrences only of states containing the encountered $\Delta S'$ while decreasing the number of occurrences for unseen ones. As in a single trajectory, we do not visit each possible $\Delta S'$, this adds variance. The less information an encoding $U$ has, the more underlying states it groups together, and hence the less rewarding outcomes are 'forgotten' in the gradient estimator, leading to lower variance.

### 3.4 Learning the contribution coefficients

In practice, we do not have access to the ground-truth contribution coefficients, but need to learn them from observations. Following Harutyunyan et al. [1], we can approximate the hindsight distribution $p^\pi(A_t = a \mid S_t = s, U' = u')$, now conditioned on rewarding outcome encodings instead of states, by training a model $h(a \mid s, u')$ on the supervised discriminative task of classifying the observed action $a_t$ given the current state $s_t$ and some future rewarding outcome $u'$. Note that if the model $h$ does not approximate the hindsight distribution perfectly, the COCOA gradient estimator (4) can be biased (c.f. Section 4). A central difficulty in approximating the hindsight distribution is that it is policy dependent, and hence changes during training. Proposition 5 shows that we can provide the policy logits as an extra input to the hindsight network without altering the learned hindsight

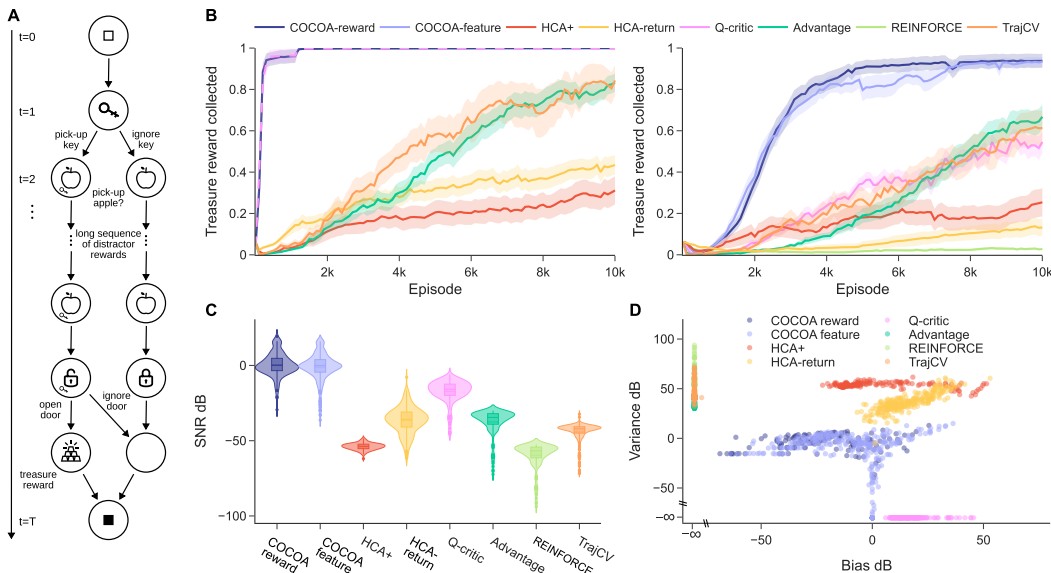

Figure 3: **COCOA enhances policy gradient estimates and sample efficiency whereas HCA fails to improve over baselines.** (A) Schematic representation of the linear key-to-door environment. (B) Performance of COCOA and baselines on the main task of picking up the treasure, measured as the average fraction of treasure rewards collected. (Left) ground-truth policy gradient estimators computed using dynamic programming, (right) learning the contribution coefficients or (action-)value function using neural networks. Shaded regions are the standard error (30 seeds). (C) Violin plot of the signal-to-noise ratio (SNR) in Decibels for the various policy gradient estimators with learned coefficients and (action-)value functions, computed on the same trajectories of a shared base policy. (D) Comparison of the bias-variance trade-off incurred by different policy gradient estimators, computed as in (C), normalized by the ground-truth policy gradient norm (scatter plot showing 30 seeds per method).

distribution, to make the parameters of the hindsight network less policy-dependent. This observation justifies and generalizes the strategy of adding the policy logits to the hindsight model output, as proposed by Alipov et al. [29].

**Proposition 5.** $p^\pi(a \mid s, u', l) = p^\pi(a \mid s, u')$, *with $l$ a deterministic function of $s$, representing the sufficient statistics of $\pi(a \mid s)$.*

As an alternative to learning the hindsight distribution, we can directly estimate the probability ratio $p^\pi(A_t = a \mid S_t = s, U' = u')/\pi(a \mid s)$ using a contrastive loss (c.f. App. D). Yet another path builds on the observation that the sums $\sum_{k \geq 1} p^\pi(U_{t+k} = u' \mid s, a)$ are akin to Successor Representations and can be learned via temporal difference updates [30, 31] (c.f. App. D). We experimented both with the hindsight classification and the contrastive loss and found the former to work best in our experiments. We leverage the Successor Representation to obtain ground truth contribution coefficients via dynamic programming for the purpose of analyzing our algorithms.

## 4 Experimental analysis

To systematically investigate long-term credit assignment performance of COCOA compared to standard baselines, we design an environment which pinpoints the core credit assignment problem and leverage dynamic programming to compute ground-truth policy gradients, contribution coefficients, and value functions (c.f. App E.2). This enables us to perform detailed bias-variance analyses and to disentangle the theoretical optimal performance of the various gradient estimators from the approximation quality of learned contribution coefficients and (action-)value functions.

We consider the *linear key-to-door* environment (c.f. Fig. 3A), a simplification of the key-to-door environment [3, 4, 32] to a one-dimensional track. Here, the agent needs to pick up a key in the first time step, after which it engages in a distractor task of picking up apples with varying reward values. Finally, it can open a door with the key and collect a treasure. This setting allows us to parametrically increase the difficulty of long-term credit assignment by increasing the distance between the key and door, making it harder to pick up the learning signal of the treasure reward

among a growing number of varying distractor rewards [4]. We use the signal-to-noise ratio, SNR= $\|\nabla_\theta V^\pi\|^2 / \mathbb{E}[\|\hat{\nabla}_\theta V^\pi - \nabla_\theta V^\pi\|^2]$, to quantify the quality of the different policy gradient estimators; a higher SNR indicates that we need fewer trajectories to estimate accurate policy gradients [33].

Previously, we showed that taking the reward as rewarding outcome encoding results in the lowest-variance policy gradients when using ground-truth contribution coefficients. In this section, we will argue that when *learning* the contribution coefficients, it is beneficial to use an encoding $u$ of the underlying *rewarding object* since this allows to distinguish different rewarding objects when they have the same scalar reward value and allows for quick adaptation when the reward function but not the environment dynamics changes.

We study two variants of COCOA, COCOA-reward which uses the reward identity for $U$, and COCOA-feature which acquires features of rewarding objects by learning a reward model $r(s, a)$ and taking the penultimate network layer as $U$. We learn the contribution coefficients by approximating the hindsight distribution with a neural network classifier $h(a \mid s, u', l)$ that takes as input the current state $s$, resulting policy logits $l$, and rewarding outcome $u'$, and predicts the current action $a$ (c.f. App. E for all experimental details). As HCA+ (c.f. Tab. 1) performs equally or better compared to HCA [1] in our experiments (c.f. App. F), we compare to HCA+ and several other baselines: (i) three classical policy gradient estimators, REINFORCE, Advantage and Q-critic, (ii) TrajCV [34], a state-of-the-art control variate method that uses hindsight information in its baseline, and (iii) HCA-return [1], a different HCA variant that uses the hindsight distribution conditioned on the return as an action-dependent baseline (c.f. Tab. 1).

## 4.1 COCOA improves sample-efficiency due to favorable bias-variance trade-off.

To investigate the quality of the policy gradient estimators of COCOA, we consider the linear key-to-door environment with a distance of 100 between key and door. Our dynamic programming setup allows us to disentangle the performance of the estimators independent of the approximation quality of learned models by using ground truth contribution coefficients and (action-)value functions. The left panel of figure 3B reveals that in this ground truth setting, COCOA-reward almost immediately solves the task performing as well as the theoretically optimal Q-critic with a perfect action-value function. This is in contrast to HCA and HCA-return which perform barely better than REINFORCE, all failing to learn to consistently pick up the key in the given number of episodes. This result translates to the setting of learning the underlying models using neural networks. COCOA-reward and -feature outperform competing policy gradient estimators in terms of sample efficiency while HCA only improves over REINFORCE. Notably, having to learn the full action-value function leads to a less sample-efficient policy gradient estimator for the Q-critic.

In Figure 3C and D we leverage dynamic programming to compare to the ground truth policy gradient. This analysis reveals that improved performance of COCOA is reflected in a higher SNR compared to other estimators due to its favorable bias-variance trade-off. Fig 12 in App. F indicates that COCOA maintains a superior SNR, even when using significantly biased contribution coefficients. As predicted by our theory in Section 3.3, HCA significantly underperforms compared to baselines due to its high variance caused by spurious contributions. In particular, the Markov state representation of the linear key-to-door environment contains the information of whether the key has been picked up. As a result, HCA always credits picking up the key or not, even for distractor rewards. These spurious contributions bury the useful learning signal of the treasure reward in noisy distractor rewards. HCA-return performs poorly as it is a biased gradient estimator, even when using the ground-truth hindsight distribution (c.f. Appendix F and L). Interestingly, the variance of COCOA is significantly lower compared to a state-of-the-art control variate method, TrajCV, pointing to a potential benefit of the multiplicative interactions between contribution coefficients and rewards in the COCOA estimators, compared to the additive interaction of the control variates: the value functions used in TrajCV need to approximate the full average returns, whereas COCOA can ignore rewards from the distractor subtask, by multiplying them with a contribution coefficient of zero.

## 4.2 COCOA enables long-term credit assignment by disentangling rewarding outcomes.

The linear key-to-door environment allows us to parametrically increase the difficulty of the long-term credit assignment problem by increasing the distance between the key and door and thereby increasing the variance due to the distractor task [4]. Figure 1B reveals that as this distance increases, performance measured as the average fraction of treasure collected over a fixed number of episodes

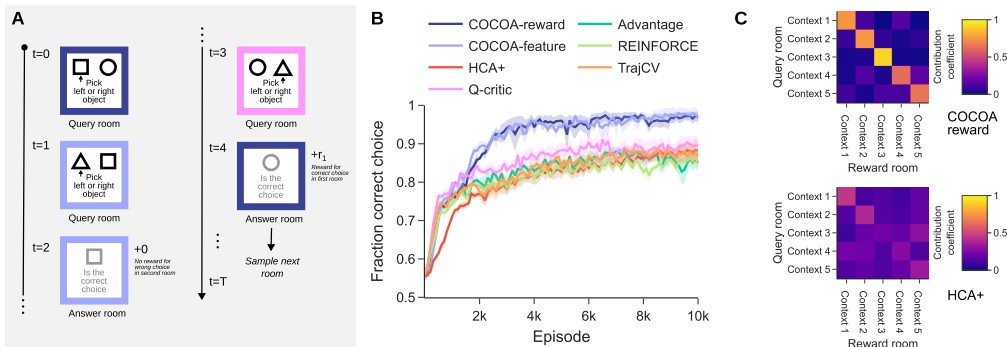

Figure 4: **COCOA improves performance by disentangling subtasks.** (A) Schematic representation of the task interleaving environment where colored borders indicate the context of a room. (B) Performance of COCOA and baselines with learned contribution coefficients or value functions, measured as the fraction of correct choices. (C) Visualization of the contribution coefficient magnitudes of each query room on reward rooms for COCOA (top) and HCA (bottom), c.f. App. E.7.

drops for all baselines including HCA but remains relatively stable for the COCOA estimators. Hung et al. [4] showed that the SNR of REINFORCE decreases inversely proportional to growing distance between key and door. Figure 1B shows that HCA and all baselines follow this trend, whereas the COCOA estimators maintain a robust SNR.

We can explain the qualitative difference between COCOA and other methods by observing that the key-to-door task consists of two distinct subtasks: picking up the key to get the treasure, and collecting apples. COCOA can quickly learn that actions relevant for one task, do not influence rewarding outcomes in the other task, and hence output a contribution coefficient equal to zero for those combinations. Value functions in contrast estimate the expected sum of future rewards, thereby mixing the rewards of both tasks. When increasing the variance of the return in the distractor task by increasing the number of stochastic distractor rewards, estimating the value functions becomes harder, whereas estimating the contribution coefficients between state-action pairs and distinct rewarding objects remains of equal difficulty, showcasing the power of disentangling rewarding outcomes.

To further showcase the power of disentangling subtasks, we consider a simplified version of the *task interleaving* environment of Mesnard et al. [3] (c.f. Fig. 4A, App. E.7). Here, the agent is faced with a sequence of contextual bandit tasks, where the reward for a correct decision is given at a later point in time, together with an observation of the relevant context. The main credit assignment difficulty is to relate the reward and contextual observation to the correct previous contextual bandit task. Note that the variance in the return is now caused by the stochastic policy, imperfectly solving future tasks, and by stochastic state transitions, in contrast to the linear key-to-door environment where the variance is caused by stochastic distractor rewards. Figure 4B shows that our COCOA algorithms outperform all baselines. The learned contribution coefficients of COCOA reward accurately capture that actions in one context only contribute to rewards in the same context as opposed to HCA that fails to disentangle the contributions (c.f. Figure 4C).

### 4.3 Learned credit assignment features allow for disentangling aliased rewards

For COCOA-reward we use the scalar reward value to identify rewarding outcomes in the hindsight distribution, i.e. $U = R$. In cases where multiple rewarding outcomes yield an identical scalar reward value, the hindsight distribution cannot distinguish between them and has to estimate a common hindsight

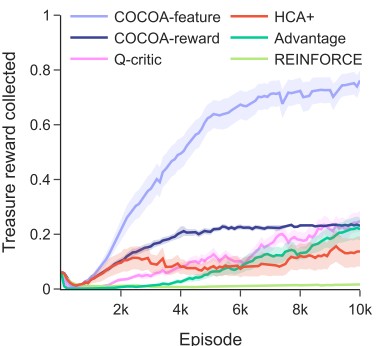

Figure 5: **COCOA-features is robust to reward aliasing**. On a version of the linear key-to-door environment where one of the distractor reward values has the same magnitude as the treasure reward, COCOA-reward can no longer distinguish between the distractor and treasure reward and as a result its performance decreases. COCOA-feature is robust to this manipulation since it relies on learned features to distinguish rewarding objects.

probability, making it impossible to disentangle the tasks and hence rendering learning of the contribution coefficients potentially more difficult. In contrast, COCOA-feature learns hindsight features of rewarding objects that are predictive of the reward. Even when multiple rewarding objects lead to an identical scalar reward value their corresponding features are likely different, allowing COCOA-feature to disentangle the rewarding outcomes.

In Fig. 5, we test this *reward aliasing* setting experimentally and slightly modify the linear key-to-door environment by giving the treasure reward the same value as one of the two possible values of the stochastic distractor rewards. As expected, COCOA-feature is robust to reward aliasing, continuing to perform well on the task of picking up the treasure while performance of COCOA-reward noticeably suffers. Note that the performance of all methods has slightly decreased compared to Fig. 3, as the magnitude of the treasure reward is now smaller relative to the variance of the distractor rewards, resulting in a worse SNR for all methods.

## 5  Discussion

We present a theory for model-based credit assignment compatible with discrete actions and show in a comprehensive theoretical and experimental analysis that this yields a powerful policy gradient estimator, enabling long-term credit assignment by disentangling rewarding outcomes.

Building upon HCA [1], we focus on amortizing the estimation of the contribution coefficients in an inverse dynamics model, $p^\pi(a \mid s, u')$. The quality of this model is crucial for obtaining low-bias gradient estimates, but it is restricted to learn from on-policy data, and rewarding observations in case of $u = r$. Scaling these inverse models to complex environments will potentially exacerbate this tension, especially in sparse reward settings. A promising avenue for future work is to leverage forward dynamics models and directly estimate contribution coefficients from synthetic trajectories. While learning a forward model is a difficult problem in its own, its policy independence increases the data available for learning it. This would result in an algorithm close in spirit to Stochastic Value Gradients [20] and Dreamer [21–23] with the crucial advance that it enables model-based credit assignment on discrete actions. Another possibility to enable learning from non-rewarding observations is to learn a generative model that can recombine inverse models based on state representations into reward contributions (c.f. App. H).

Related work has explored the credit assignment problem through the lens of transporting rewards or value estimates towards previous states to bridge long-term dependencies [4, 5, 32, 35–41]. This approach is compatible with existing and well-established policy gradient estimators but determining how to redistribute rewards has relied on heuristic contribution analyses, such as via the access of memory states [4], linear decompositions of rewards [32, 35–39] or learned sequence models [5, 40, 41]. Leveraging our unbiased contribution analysis framework to reach more optimal reward transport is a promising direction for future research.

While we have demonstrated that contribution coefficients with respect to states as employed by HCA suffer from spurious contributions, any reward feature encoding that is fully predictive of the reward can in principle suffer from a similar problem in the case where each environment state has a unique reward value. In practice, this issue might occur in environments with continuous rewards. A potential remedy in this situation is to assume that the underlying reward distribution is stochastic, smoothing the contribution coefficients as now multiple states could have led to the same reward. This lowers the variance of the gradient estimator as we elaborate in App. G.

Finally, we note that our contribution coefficients are closely connected to causality theory [42] where the contribution coefficients correspond to performing *Do-interventions* on the causal graph to estimate their effect on future rewards (c.f. App I). Within causality theory, counterfactual reasoning goes a step further by inferring the external, uncontrollable environment influences and considering the consequences of counterfactual actions *given that all external influences remain the same* [3, 20, 42–44]. Extending COCOA towards this more advanced counterfactual setting by building upon recent work [3, 43] is an exciting direction for future research (c.f. App. I).

**Concluding remarks.** By overcoming the failure mode of *spurious contributions* in HCA, we have presented here a comprehensive theory on how to leverage model information for credit assignment, compatible with discrete action spaces. COCOA-reward and COCOA-feature are promising first algorithms in this framework, opening the way towards sample-efficient reinforcement learning by model-based credit assignment.

## 6 Acknowledgements

We thank Angelika Steger, Yassir Akram, Ida Momennejad, Blake Richards, Matt Botvinick and Joel Veness for discussions and feedback. Simon Schug is supported by the Swiss National Science Foundation (PZ00P3_186027). Seijin Kobayashi is supported by the Swiss National Science Foundation (CRSII5_173721). Simon Schug would like to kindly thank the TPU Research Cloud (TRC) program for providing access to Cloud TPUs from Google.

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

# Supplementary Materials

## Table of Contents

# A   Related work

Our work builds upon Hindsight Credit Assignment (HCA) [1] which has sparked a number of follow-up studies. We generalize the theory of HCA towards estimating contributions upon rewarding outcomes instead of rewarding states and show through a detailed variance analysis that HCA suffers from spurious contributions leading to high variance, while using rewards or rewarding objects as rewarding outcome encodings leads to low-variance gradient estimators. Follow-up work on HCA discusses its potential high variance in constructed toy settings, and reduces the variance of the HCA advantage estimates by combining it with Monte Carlo estimates [45] or using temporal difference errors instead of rewards [46]. Alipov et al. [29] leverages the latter approach to scale up HCA towards more complex environments. However, all of the above approaches still suffer from spurious contributions, a significant source of variance in the HCA gradient estimator. In addition, recent studies have theoretically reinterpreted the original HCA formulation from different angles: Ma and Pierre-Luc [47] create the link to Conditional Monte Carlo methods [48, 49] and Arumugam et al. [50] provide an information theoretic perspective on credit assignment. Moreover, Young [51] applies HCA to the problem of estimating gradients in neural networks with stochastic, discrete units.

Long-term credit assignment in RL is hard due to the high variance of the sum of future rewards for long trajectories. A common technique to reduce the resulting variance in the policy gradients is to subtract a baseline from the sum of future rewards [6–9]. To further reduce the variance, a line of work introduced state-action-dependent baselines [52–55]. However, Tucker et al. [56] argues that these methods offer only a small benefit over the conventional state-dependent baselines, as the current action often only accounts for a minor fraction of the total variance. More recent work proposes improved baselines by incorporating hindsight information about the future trajectory into the baseline, accounting for a larger portion of the variance [1, 3, 34, 57–60].Cheng et al. [34] includes Q-value estimates of future state-action pairs into the baseline, and Huang and Jiang [60] goes one step further by also including cheap estimates of the future Q-value gradients.Mesnard et al. [3] learn a summary metric of the uncontrollable, external environment influences in the future trajectory, and provide this hindsight information as an extra input to the value baseline. Nota et al. [57] consider partially observable MDPs, and leverage the future trajectory to more accurately infer the current underlying Markov state, thereby providing a better value baseline. Harutyunyan et al. [1] propose return-HCA, a different variant of HCA that uses a return-conditioned hindsight distribution to construct a baseline, instead of using state-based hindsight distributions for estimating contributions. Finally, Guez et al. [58] and Venuto et al. [59] learn a summary representation of the full future trajectory and provide it as input to the value function, while imposing an information bottleneck to prevent the value function from overly relying on this hindsight information.

Environments with sparse rewards and long delays between actions and corresponding rewards put a high importance on long-term credit assignment. A popular strategy to circumvent the sparse and delayed reward setting is to introduce reward shaping [61–68]. These approaches add auxiliary rewards to the sparse reward function, aiming to guide the learning of the policy with dense rewards. A recent line of work introduces a reward-shaping strategy specifically designed for long-term credit assignment, where rewards are decomposed and distributed over previous state-action pairs that were instrumental in achieving that reward [4, 5, 32, 35–41]. To determine how to redistribute rewards,

these approaches rely on heuristic contribution analyses, such as via the access of memory states [4], linear decompositions of rewards [32, 35–39] or learned sequence models [5, 40, 41]. Leveraging our unbiased contribution analysis framework to reach more optimal reward transport is a promising direction for future research.

When we have access to a (learned) differentiable world model of the environment, we can achieve precise credit assignment by leveraging path-wise derivatives, i.e. backpropagating value gradients through the world model [20–24, 69]. For stochastic world models, we need access to the noise variables to compute the path-wise derivatives. The Dreamer algorithms [21–23] approach this by computing the value gradients on *simulated* trajectories, where the noise is known. The Stochastic Value Gradient (SVG) method [20] instead *infers* the noise variables on real *observed* trajectories. To enable backpropagating gradients over long time spans, Ma et al. [70] equip the learned recurrent world models of SVG with an attention mechanism, allowing the authors to leverage Sparse Attentive Backtracking [71] to transmit gradients through skip connections. Buesing et al. [43] leverages the insights from SVG in partially observable MDPs, using the inferred noise variables to estimate the effect of counterfactual policies on the expected return. Importantly, the path-wise derivatives leveraged by the above model-based credit assignment methods are not compatible with discrete action spaces, as sensitivities w.r.t. discrete actions are undefined. In contrast, COCOA can leverage model-based information for credit assignment, while being compatible with discrete actions.

Incorporating hindsight information has a wide variety of applications in RL. Goal-conditioned policies [72] use a goal state as additional input to the policy network or value function, thereby generalizing it to arbitrary goals. Hindsight Experience Replay [73] leverages hindsight reasoning to learn almost as much from undesired outcomes as from desired ones, as at hindsight, we can consider every final, possibly undesired state as a 'goal' state and update the value functions or policy network [74] accordingly. Goyal et al. [75] train an inverse environment model to simulate alternative past trajectories leading to the same rewarding state, hence leveraging hindsight reasoning to create a variety of highly rewarding trajectories. A recent line of work frames RL as a supervised sequence prediction problem, learning a policy conditioned on goal states or future returns [76–79]. These models are trained on past trajectories or offline data, where we have in hindsight access to states and returns, considering them as targets for the learned policy.

Finally, Temporal Difference (TD) learning [80] has a rich history of leveraging proximity in time as a proxy for credit assignment [2]. TD($\lambda$) [80] considers eligibility traces to trade off bias and variance, crediting past state-action pairs in the trajectory for the current reward proportional to how close they are in time. van Hasselt et al. [81] estimate expected eligibility traces, taking into account that the same rewarding state can be reached from various previous states. Hence, not only the past state-actions on the trajectory are credited and updated, but also counterfactual ones that lead to the same rewarding state. Extending the insights from COCOA towards temporal difference methods is an exciting direction for future research.

# B   Undiscounted infinite-horizon MDPs

In this work, we consider an undiscounted MDP with a finite state space $\mathcal{S}$, bounded rewards and an infinite horizon. To ensure that the expected return and value functions remain finite, we require some standard regularity conditions [2]. We assume the MDP contains an absorbing state $s_\infty$ that transitions only to itself and has zero reward. Moreover, we assume *proper* transition dynamics, meaning that an agent following any policy will eventually end up in the absorbing state $s_\infty$ with probability one as time goes to infinity.

The discounted infinite horizon MDP formulation is a special case of this setting, with a specific class of transition dynamics. An explicit discounting of future rewards $\sum_{k \geq o} \gamma^k R_k$ with discount factor $\gamma \in [0, 1]$, is equivalent to considering the above undiscounted MDP setting, but modifying the state transition probability function $p(S_{t+1} \mid S_t, A_t)$ such that each state-action pair has a fixed probability $(1 - \gamma)$ of transitioning to the absorbing state [2]. Hence, all the results considered in this work can be readily applied to the discounted MDP setting, by modifying the environment transitions as outlined above. We can also explicitly incorporate temporal discounting in the policy gradients and contribution coefficients, which we outline in App. J.

The undiscounted infinite-horizon MDP with an absorbing state can also model episodic RL with a fixed time horizon. We can include time as an extra feature in the states $S$, and then have a probability of 1 to transition to the absorbing state when the agent reaches the final time step.

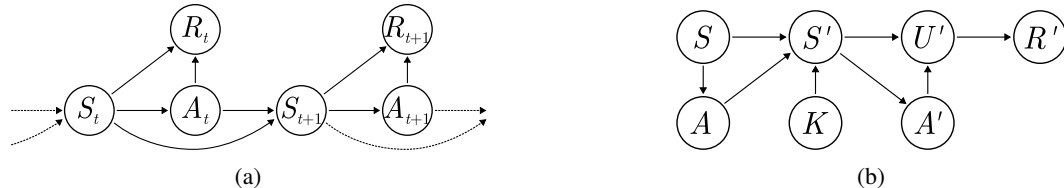

(a)                  (b)

Figure 6: (a) Graphical model of the MDP. (b) Graphical model of the MDP, where we abstracted time.

## C   Theorems, proofs and additional information for Section 3

### C.1   Contribution coefficients, hindsight distribution and graphical models

Here, we provide more information on the derivation of the contribution coefficients of Eq. 3.

**Abstracting time.** In an undiscounted environment, it does not matter at which point in time the agent achieves the rewarding outcome $u'$. Hence, the contribution coefficients (3) sum over all future time steps, to reflect that the rewarding outcome $u'$ can be encountered at any future time, and to incorporate the possibility of encountering $u'$ multiple times. Note that when using temporal discounting, we can adjust the contribution coefficients accordingly (c.f. App J).

**Hindsight distribution.** To obtain the hindsight distribution $p(A_t = a \mid S_t = s, U' = u')$, we convert the classical graphical model of an MDP (c.f. Fig. 6a) into a graphical model that incorporates the time $k$ into a separate node (c.f. Fig. 6b). Here, we rewrite $p(U_k = u' \mid S = s, A = a)$, the probability distribution of a rewarding outcome $U_k$ $k$ time steps later, as $p(U' = u' \mid S = s, A = a, K = k)$. By giving $K$ the geometric distribution $p(K = k) = (1 - \beta)\beta^{k-1}$ for some $\beta$, we can rewrite the infinite sums used in the time-independent contribution coefficients as a marginalization over $K$:

$$\sum_{k \geq 1} p(U_k = u' \mid S = s, A = a) = \lim_{\beta \to 1} \sum_{k \geq 1} p(U_k = u' \mid S = s, A = a)\beta^{k-1} \tag{5}$$

$$= \lim_{\beta \to 1} \frac{1}{1 - \beta} \sum_{k \geq 1} p(U' = u' \mid S = s, A = a, K = k)p(K = k) \tag{6}$$

$$= \lim_{\beta \to 1} \frac{1}{1 - \beta} p(U' = u' \mid S = s, A = a) \tag{7}$$

Note that this limit is finite, as for $k \to \infty$, the probability of reaching an absorbing state $s_\infty$ and corresponding rewarding outcome $u_\infty$ goes to 1 (and we take $u' \neq u_\infty$). Via Bayes rule, we then have that

$$w(s, a, u') = \frac{\sum_{k \geq 1} p(U_k = u' \mid S = s, A = a)}{\sum_{k \geq 1} p(U_k = u' \mid S = s)} - 1 \tag{8}$$

$$= \frac{p(U' = u' \mid S = s, A = a)}{p(U' = u \mid S = s)} - 1 = \frac{p(A = a \mid S = s, U' = u')}{\pi(a \mid s)} - 1 \tag{9}$$

where we assume $\pi(a \mid s) > 0$ and where we dropped the limit of $\beta \to 1$ in the notation, as we will always take this limit henceforth. Note that when $\pi(a \mid s) = 0$, the hindsight distribution $p^\pi(a \mid s, u')$ is also equal to zero, and hence the right-hand-side of the above equation is undefined. However, the middle term using $p^\pi(U = u' \mid S = s, A = a)$ is still well-defined, and hence we can use the contribution coefficients even for actions where $\pi(a \mid s) = 0$.

### C.2   Proof Theorem 1

We start with a lemma showing that the contribution coefficients can be used to estimate the advantage $A^\pi(s, a) = Q^\pi(s, a) - V^\pi(s)$, by generalizing Theorem 1 of Harutyunyan et al. [1] towards rewarding outcomes.

**Definition 3** (Fully predictive, repeated from main text). *A rewarding outcome encoding $U$ is fully predictive of the reward $R$, if the following conditional independence condition holds:* $p^\pi(R_k = r \mid$

$S_0 = s, A_0 = a, U_k = u) = p^\pi(R = r \mid U = u)$, where the right-hand side does not depend on the time $k$.

**Lemma 6.** *For each state-action pair $(s, a)$ with $\pi(a \mid s) > 0$, and assuming that $u = f(s, a, r)$ is fully predictive of the reward (c.f. Definition 3), we have that*

$$A^\pi(s, a) = r(s, a) - \sum_{a' \in \mathcal{A}} \pi(a \mid s) r(s, a) + \mathbb{E}_{T \sim \mathcal{T}(s, \pi)} \left[ \sum_{k \geq 1} w(s, a, U_k) R_k \right] \qquad (10)$$

*with the advantage function $A^\pi$, and the reward function $r(s, a) \triangleq \mathbb{E}[R \mid s, a]$.*

*Proof.* We rewrite the undiscounted state-action value function in the limit of a discounting factor $\beta \to 1_-$ (the minus sign indicating that we approach 1 from the left):

$$Q(s, a) = \lim_{\beta \to 1_-} \mathbb{E}_{T \sim \mathcal{T}(s, a, \pi)} \left[ \sum_{k \geq 1} \beta^k R_k \right] \qquad (11)$$

$$= r(s, a) + \lim_{\beta \to 1_-} \sum_{r \in \mathcal{R}} \sum_{k \geq 1} \beta^k p^\pi(R_k = r \mid s, a) r \qquad (12)$$

$$= r(s, a) + \lim_{\beta \to 1_-} \sum_{r \in \mathcal{R}} \sum_{u \in \mathcal{U}} \sum_{k \geq 1} \beta^k p^\pi(R_k = r, U_k = u \mid s, a) r \qquad (13)$$

$$= r(s, a) + \lim_{\beta \to 1_-} \sum_{r \in \mathcal{R}} \sum_{u \in \mathcal{U}} \sum_{k \geq 1} \beta^k p^\pi(R = r \mid U = u) r p^\pi(U_k = u \mid s, a) \qquad (14)$$

$$= r(s, a) + \lim_{\beta \to 1_-} \sum_{u \in \mathcal{U}} r(u) \sum_{k \geq 1} \beta^k p^\pi(U_k = u \mid s, a) \qquad (15)$$

$$= r(s, a) + \lim_{\beta \to 1_-} \sum_{u \in \mathcal{U}} r(u) \sum_{k \geq 1} \beta^k p^\pi(U_k = u \mid s) \frac{\sum_{k' \geq 1} \beta^{k'} p^\pi(U_{k'} = u \mid s, a)}{\sum_{k' \geq 1} \beta^{k'} p^\pi(U_{k'} = u \mid s)} \qquad (16)$$

where we use the property that $u$ is *fully predictive* of the reward (c.f. Definition 2), and where we define $r(s, a) \triangleq \mathbb{E}[R \mid s, a]$ and $r(u) \triangleq \mathbb{E}[R \mid u]$ Using the graphical model of Fig. 6b where we abstract time $k$ (c.f. App. C.1), we have that $\frac{1}{1-\beta} p_\beta^\pi(R = r' \mid s, a) = \sum_{k=0} \beta^k p^\pi(R_k = r \mid s, a)$. Leveraging Bayes rule, we get

$$p^\pi(a \mid s, U' = u) = \lim_{\beta \to 1_-} \frac{p_\beta^\pi(U' = u \mid s, a) \pi(a \mid s)}{p_\beta^\pi(U' = u \mid s)} = \lim_{\beta \to 1_-} \frac{\sum_{k \geq 1} \beta^k p^\pi(U_k = u \mid s, a)}{\sum_{k \geq 1} \beta^k p^\pi(U_k = u \mid s)} \pi(a \mid s) \qquad (17)$$

Where we dropped the limit of $\beta \to 1_-$ in the notation of $p^\pi(a \mid s, R = r)$, as we henceforth always consider this limit. Taking everything together, we have that

$$Q^\pi(s, a) = r(s, a) + \sum_{u \in \mathcal{U}} \sum_{k \geq 1} p^\pi(U_k = u \mid s) \frac{p^\pi(a \mid s, U' = u)}{\pi(a \mid s)} r(u) \qquad (18)$$

$$= r(s, a) + \sum_{u \in \mathcal{U}} \sum_{k \geq 1} p^\pi(U_k = u \mid s)(w(s, a, u) + 1) r(u) \qquad (19)$$

$$= r(s, a) + \sum_{u \in \mathcal{U}} \sum_{k \geq 1} p^\pi(U_k = u \mid s)(w(s, a, u) + 1) \sum_{r \in \mathcal{R}} p(R_k = r \mid U_k = u) r \qquad (20)$$

$$= r(s, a) + \mathbb{E}_{T \sim \mathcal{T}(s, \pi)} \left[ \sum_{k \geq 1} (w(s, a, U_k) + 1) R_k \right] \qquad (21)$$

Subtracting the value function, we get

$$A^\pi(s, a) = r(s, a) - \sum_{a' \in \mathcal{A}} \pi(a' \mid s) r(s, a') + \mathbb{E}_{T \sim \mathcal{T}(s, \pi)} \left[ \sum_{k \geq 1} w(s, a, U_k) R_k \right] \qquad (22)$$

$\square$

Now we are ready to prove Theorem 1.

*Proof.* Using the policy gradient theorem [26], we have

$$\nabla_\theta V^\pi(s_0) = \mathbb{E}_{T \sim \mathcal{T}(s_0, \pi)} \left[ \sum_{t \geq 0} \sum_{a \in \mathcal{A}} \nabla_\theta \pi(a \mid S_t) A^\pi(S_t, a) \right] \tag{23}$$

$$= \mathbb{E}_{T \sim \mathcal{T}(s_0, \pi)} \left[ \sum_{t \geq 0} \sum_{a \in \mathcal{A}} \nabla_\theta \pi(a \mid S_t) \left[ r(S_t, a) + \sum_{k \geq 1} w(S_t, a, U_{t+k}) R_{t+k} \right] \right] \tag{24}$$

$$= \mathbb{E}_{T \sim \mathcal{T}(s_0, \pi)} \left[ \sum_{t \geq 0} \nabla_\theta \log \pi(A_t \mid S_t) R_t + \sum_{a \in \mathcal{A}} \nabla_\theta \pi(a \mid S_t) \sum_{k \geq 1} w(S_t, a, U_{t+k}) R_{t+k} \right] \tag{25}$$

where we used that removing a baseline $\sum_{a' \in \mathcal{A}} \pi(a' \mid s) r(s, a')$ independent from the actions does not change the policy gradient, and replaced $\mathbb{E}_{T \sim \mathcal{T}(s_0, \pi)} [\sum_{a \in \mathcal{A}} \nabla_\theta \pi(a \mid S_t) r(S_t, a)]$ by its sampled version $\mathbb{E}_{T \sim \mathcal{T}(s_0, \pi)} [\nabla_\theta \log \pi(A_t \mid S_t) R_t]$. Hence the policy gradient estimator of Eq. 4 is unbiased. $\square$

### C.3 Different policy gradient estimators leveraging contribution coefficients.

In this work, we use the COCOA estimator of Eq. 4 to estimate policy gradients leveraging the contribution coefficients of Eq. 3, as this estimator does not need a separate reward model, and it works well for the small action spaces we considered in our experiments. However, one can design other unbiased policy gradient estimators compatible with the same contribution coefficients.

Harutyunyan et al. [1] introduced the HCA gradient estimator of Eq. 2, which can readily be extended to rewarding outcomes $U$:

$$\hat{\nabla}_\theta V^\pi = \sum_{t \geq 0} \sum_{a \in \mathcal{A}} \nabla_\theta \pi(a \mid S_t) \left( r(S_t, a) + \sum_{k \geq 1} w(S_t, a, U_{t+k}) R_{t+k} \right) \tag{26}$$

This gradient estimator uses a reward model $r(s, a)$ to obtain the rewards corresponding to counterfactual actions, whereas the COCOA estimator (4) only uses the observed rewards $R_k$.

For large action spaces, it might become computationally intractable to sum over all possible actions. In this case, we can sample independent actions from the policy, instead of summing over all actions, leading to the following policy gradient.

$$\hat{\nabla}_\theta V^\pi = \sum_{t \geq 0} \nabla_\theta \log \pi(A_t \mid S_t) R_t + \frac{1}{M} \sum_m \nabla_\theta \log \pi(a^m \mid S_t) \sum_{k=1}^\infty w(S_t, a^m, U_{t+k}) R_{t+k} \tag{27}$$

where we sample $M$ actions from $a^m \sim \pi(\cdot \mid S_t)$. Importantly, for obtaining an unbiased policy gradient estimate, the actions $a^m$ should be sampled independently from the actions used in the trajectory of the observed rewards $R_{t+k}$. Hence, we cannot take the observed $A_t$ as a sample $a^m$, but need to instead sample independently from the policy. This policy gradient estimator can be used for continuous action spaces (c.f. App. G, and can also be combined with the reward model as in Eq. 26.

One can show that the above policy gradient estimators are unbiased with a proof akin to the one of Theorem 1.

**Comparison of COCOA to using time as a credit assignment heuristic.** In Section 2 we discussed briefly the following discounted policy gradient:

$$\hat{\nabla}_\theta^{\text{REINFORCE}, \gamma} V^\pi(s_0) = \sum_{t \geq 0} \nabla_\theta \log \pi(A_t \mid S_t) \sum_{k \geq t} \gamma^{k-t} R_k \tag{28}$$

with discount factor $\gamma \in [0, 1]$. Note that we specifically put no discounting $\gamma^t$ in front of $\nabla_\theta \log \pi(A_t \mid S_t)$, which would be required for being an unbiased gradient estimate of the discounted expected total return, as the above formulation is most used in practice [12, 27]. Rewriting

the summations reveals that this policy gradient uses time as a heuristic for credit assignment:

$$\hat{\nabla}_\theta^{\text{REINFORCE},\gamma} V^\pi(s_0) = \sum_{t\geq 0} R_t \sum_{k\leq t} \gamma^{t-k}\nabla_\theta \log \pi(A_k \mid S_k). \tag{29}$$

We can rewrite the COCOA gradient estimator (4) with the same reordering of summation, showcasing that it leverages the contribution coefficient for providing precise credit to past actions, instead of using the time discounting heuristic:

$$\hat{\nabla}_\theta^U V^\pi(s_0) = \sum_{t\geq 0} R_t \left[ \nabla_\theta \log \pi(A_t \mid S_t) + \sum_{k<t}\sum_{a\in\mathcal{A}} w(S_k, a, U_t)\nabla_\theta \pi(a \mid S_k) \right] \tag{30}$$

## C.4    Proof of Proposition 2

*Proof.* Proposition 2 assumes that each action sequence $\{A_m\}_{m=t}^{t+k}$ leads to a unique state $s'$. Hence, all previous actions can be decoded perfectly from the state $s'$, leading to $p^\pi(A_t = a \mid S_t, S' = s') = \delta(a = a_t)$, with $\delta$ the indicator function and $a_t$ the action taken in the trajectory that led to $s'$. Filling this into the COCOA gradient estimator leads to

$$\hat{\nabla}_\theta^S V^\pi(s_0) = \sum_{t\geq 0} \nabla_\theta \log \pi(A_t \mid S_t)R_t + \sum_{a\in\mathcal{A}} \nabla_\theta \pi(a \mid S_t) \sum_{k\geq 1}\left(\frac{\delta(a = A_t)}{\pi(a \mid S_t)} - 1\right)R_{t+k} \tag{31}$$

$$= \sum_{t\geq 0}\frac{\nabla_\theta \pi(A_t \mid S_t)}{\pi(A_t \mid S_t)}\sum_{k\geq 0} R_{t+k} \tag{32}$$

$$= \sum_{t\geq 0}\nabla_\theta \log \pi(A_t \mid S_t)\sum_{k\geq 0} R_{t+k} \tag{33}$$

where we used that $\sum_a \nabla_\theta \pi(a \mid s) = 0$. $\qquad\square$

## C.5    Proof Theorem 3

*Proof.* Theorem 3 considers the case where the environment only contains a reward at the final time step $t = T$, and where we optimize the policy only on a single (initial) time step $t = 0$. Then, the policy gradients are given by

$$\hat{\nabla}_\theta^U V^\pi(U_T, R_T) = \sum_a \nabla_\theta \pi(a \mid s)w(s, a, U_T)R_T \tag{34}$$

$$\hat{\nabla}_\theta^{\text{REINFORCE}} V^\pi(A_0, R_T) = \nabla_\theta \log \pi(A_0 \mid s)R_T \tag{35}$$

With $U$ either $S$, $R$ or $Z$, and $s$ the state at $t = 0$. As only the last time step can have a reward by assumption, and the encoding $U$ needs to retain the predictive information of the reward, the contribution coefficients for $u$ corresponding to a nonzero reward are equal to

$$w(s, a, u) = \frac{p^\pi(A_0 = a \mid S_0 = s, U_T = u)}{\pi(a \mid s)} - 1 \tag{36}$$

The coefficients corresponding to zero-reward outcomes are multiplied with zero in the gradient estimator, and can hence be ignored. Now, we proceed by showing that $\mathbb{E}[\hat{\nabla}_\theta^{\text{REINFORCE}} V^\pi(A_0, R_T) \mid S_T, R_T] = \hat{\nabla}_\theta^S V^\pi(S_T, R_T)$, $\mathbb{E}[\hat{\nabla}_\theta^S V^\pi(S_T, R_T) \mid U_T', R_T] = \hat{\nabla}_\theta^{U'} V^\pi(U_T', R_T)$, $\mathbb{E}[\hat{\nabla}_\theta^{U'} V^\pi(U_T', R_T) \mid U_T, R_T] = \hat{\nabla}_\theta^U V^\pi(U_T, R_T)$ , and $\mathbb{E}[\hat{\nabla}_\theta^U V^\pi(U_T, R_T) \mid R_T] = \hat{\nabla}_\theta^R V^\pi(R_T)$, after which we can use the law of total variance to prove our theorem.

As $S_T$ is fully predictive of $R_T$, the following conditional independence holds $p^\pi(A_0 \mid S_0 = s, S_T, R_T) = p^\pi(A_0 \mid S_0 = s, S_T)$ (c.f. Definition 2). Hence, we have that

$$\mathbb{E}[\hat{\nabla}_\theta^{\text{REINFORCE}} V^\pi(A_0, R_T) \mid S_T, R_T] \tag{37}$$

$$= \sum_{a\in\mathcal{A}} p(A_0 = a \mid S_0 = s, S_T)\frac{\nabla_\theta \pi(a \mid s)}{\pi(a \mid s)}R_T = \hat{\nabla}_\theta^S V^\pi(S_T, R_T) \tag{38}$$

where we used that $\sum_a \nabla_\theta \pi(a \mid s) = \nabla_\theta \sum_a \pi(a \mid s) = 0$.

Similarly, as $S_T$ is fully predictive of $U'_T$ (c.f. Definition 2), we have that

$$p(A \mid S, U'_T) = \sum_{s_T \in \mathcal{S}} p(S_T = s_T \mid S, U'_T) p(A \mid S, S_T = s_T, U'_T) \tag{39}$$

$$= \sum_{s_T \in \mathcal{S}} p(S_T = s_T \mid S, U'_T) p(A \mid S, S_T = s_T) \tag{40}$$

Using the conditional independence relation $p^\pi(S_T \mid S_0, U'_T, R_T) = p^\pi(S_T \mid S_0, U'_T)$, following from d-separation in Fig. 6b, this leads us to

$$\mathbb{E}[\hat{\nabla}_\theta^S V^\pi(S_T, R_T) \mid U'_T, R_T] \tag{41}$$

$$= \sum_{a \in \mathcal{A}} \sum_{s_T \in \mathcal{S}} p(S_T = s_T \mid S_0 = s, U'_T) p(A_0 = a \mid S = s, S_T = s_T) \frac{\nabla_\theta \pi(a \mid s)}{\pi(a \mid s)} R_T \tag{42}$$

$$= \sum_{a \in \mathcal{A}} p(A_0 = a \mid S = s, U'_T) \frac{\nabla_\theta \pi(a \mid s)}{\pi(a \mid s)} R_T \tag{43}$$

$$= \hat{\nabla}_\theta^{U'} V^\pi(U'_T, R_T) \tag{44}$$

Using the same derivation leveraging the fully predictive properties (c.f. Definition 2), we get

$$\mathbb{E}[\hat{\nabla}_\theta^{U'} V^\pi(U'_T, R_T) \mid U_T, R_T] = \hat{\nabla}_\theta^U V^\pi(U_T, R_T) \tag{45}$$

$$\mathbb{E}[\hat{\nabla}_\theta^U V^\pi(U_T, R_T) \mid R_T] = \hat{\nabla}_\theta^R V^\pi(R_T) \tag{46}$$

Now we can use the law of total variance, which states that $\mathbb{V}[X] = \mathbb{E}[\mathbb{V}[X \mid Y]] + \mathbb{V}[\mathbb{E}[X \mid Y]]$. Hence, we have that

$$\mathbb{V}[\hat{\nabla}_\theta^{\text{REINFORCE}} V^\pi] = \mathbb{V}[\mathbb{E}[\hat{\nabla}_\theta^{\text{REINFORCE}} V^\pi \mid S_T]] + \mathbb{E}[\mathbb{V}[\hat{\nabla}_\theta^{\text{REINFORCE}} V^\pi \mid S_T]] \tag{47}$$

$$= \mathbb{V}[\hat{\nabla}_\theta^S V^\pi] + \mathbb{E}[\mathbb{V}[\hat{\nabla}_\theta^{\text{REINFORCE}} V^\pi \mid S_T]] \succcurlyeq \mathbb{V}[\hat{\nabla}_\theta^S V^\pi] \tag{48}$$

as $\mathbb{E}[\mathbb{V}[\hat{\nabla}_\theta^{\text{REINFORCE}} V^\pi \mid S_T]]$ is positive semi definite. Using the same construction for the other pairs, we arrive at

$$\mathbb{V}[\hat{\nabla}_\theta^R V^\pi(s_0)] \preccurlyeq \mathbb{V}[\hat{\nabla}_\theta^U V^\pi(s_0)] \preccurlyeq \mathbb{V}[\hat{\nabla}_\theta^{U'} V^\pi(s_0)] \preccurlyeq \mathbb{V}[\hat{\nabla}_\theta^S V^\pi(s_0)] \preccurlyeq \mathbb{V}[\hat{\nabla}_\theta^{\text{REINFORCE}} V^\pi(s_0)] \tag{49}$$

thereby concluding the proof. $\qquad\square$

**Additional empirical verification.** To get more insight into how the information content of the rewarding-outcome encoding $U$ relates to the variance of the COCOA gradient estimator, we repeat the experiment of Fig. 2, but now plot the variance as a function of the amount of information in the rewarding outcome encodings, for a fixed state overlap of 3. Fig. 7 shows that the variance of the resulting COCOA gradient estimators interpolate between COCOA-reward and HCA+, with more informative encodings leading to a higher variance. These empirical results show that the insights of Theorem 3 hold in this more general setting of estimating full policy gradients in an MDP with random rewards.

## C.6 Proof of Theorem 4

*Proof.* This proof follows a similar technique to the policy gradient theorem [26]. Let us first define the expected number of occurrences of $u$ starting from state $s$ as $O^\pi(u, s) = \sum_{k \geq 1} p^\pi(U_k = u \mid S_0 = s)$, and its action equivalent $O^\pi(u, s, a) = \sum_{k \geq 1} p^\pi(U_k = u \mid S_0 = s, A_0 = a)$. Expanding $O^\pi(u, s)$ leads to

$$\nabla_\theta O^\pi(u, s) = \nabla_\theta \left[ \sum_{a \in \mathcal{A}} \pi(a \mid s) \sum_{s' \in \mathcal{S}} p(s' \mid s, a)\big(\delta(s' = s) + O^\pi(u, s')\big) \right] \tag{50}$$

$$= \sum_{a \in \mathcal{A}} \nabla_\theta \pi(a \mid s) O^\pi(u, s, a) + \sum_{s' \in \mathcal{S}} p(s' \mid s) \nabla_\theta O^\pi(u, s') \tag{51}$$

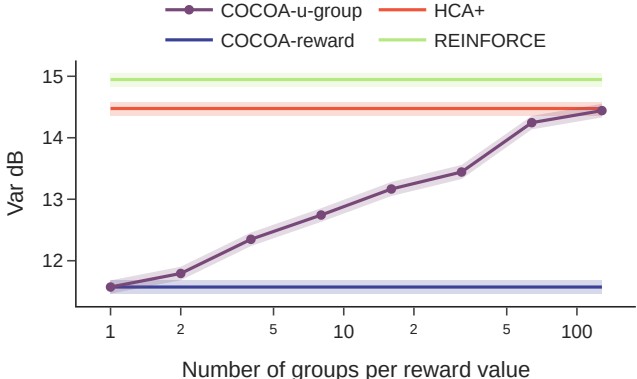

Figure 7: **Less informative rewarding outcome encodings lead to gradient estimators with lower variance.** Normalized variance in dB using ground-truth coefficients and a random uniform policy, for various gradient estimators on the tree envrionment (shaded region represents standard error over 10 random environments). To increase the information content of $U$, we increase the number of different encodings $u$ corresponding to the same reward value (c.f. $n_g$ in Section E.6), indicated by the x-axis. COCOA-u-group indicates the COCOA estimator with rewarding outcome encodings of increasing information content, whereas COCOA-reward and HCA+ have fixed rewarding outcome encodings of $U = R$ and $U = S$ respectively.

Now define $\phi(u, s) = \sum_{a \in \mathcal{A}} \nabla_\theta \pi(a \mid s) O^\pi(u, s, a)$, and $p^\pi(S_l = s' \mid S_0 = s)$ as the probability of reaching $s'$ starting from $s$ in $l$ steps. We have that $\sum_{s'' \in \mathcal{S}} p^\pi(S_l = s'' \mid S_0 = s) p^\pi(S_1 = s' \mid S_0 = s'') = p^\pi(S_{l+1} = s' \mid S_0 = s)$. Leveraging this relation and recursively applying the above equation leads to

$$\nabla_\theta O^\pi(u, s) = \sum_{s' \in \mathcal{S}} \sum_{l=0}^\infty p^\pi(S_l = s' \mid S_0 = s) \phi(u, s') \tag{52}$$

$$= \sum_{s' \in \mathcal{S}} \sum_{l=0}^\infty p^\pi(S_l = s' \mid S_0 = s) \sum_{a \in \mathcal{A}} \nabla_\theta \pi(a \mid s') O^\pi(u, s', a) \tag{53}$$

$$\propto \mathbb{E}_{S \sim \mathcal{T}(s, \pi)} \sum_{a \in \mathcal{A}} \nabla_\theta \pi(a \mid S) O^\pi(u, S, a) \tag{54}$$

where in the last step we normalized $\sum_{l=0}^\infty p^\pi(S_l = s' \mid S_0 = s)$ with $\sum_{s' \in \mathcal{S}} \sum_{l=0}^\infty p^\pi(S_l = s' \mid S_0 = s)$, resulting in the state distribution $S \sim \mathcal{T}(s, \pi)$ where $S$ is sampled from trajectories starting from $s$ and following policy $\pi$.

Finally, we can rewrite the contribution coefficients as

$$w(s, a, u) = \frac{O^\pi(u, s, a)}{O^\pi(u, s)} - 1 \tag{55}$$

which leads to

$$\nabla_\theta O^\pi(u, s) \propto \mathbb{E}_{S \sim \mathcal{T}(s, \pi)} \sum_{a \in \mathcal{A}} \nabla_\theta \pi(a \mid S) w(S, a, u) O^\pi(u, S) \tag{56}$$

where we used that $\sum_{a \in \mathcal{A}} \nabla_\theta \pi(a \mid s) = 0$, thereby concluding the proof. $\qquad \square$

## D   Learning the contribution coefficients

### D.1   Proof Proposition 5

*Proof.* In short, as the logits $l$ are a deterministic function of the state $s$, they do not provide any further information on the hindsight distribution and we have that $p^\pi(A_0 = a \mid S_0 = s, L_0 = l, U' = u') = p^\pi(A_0 = a \mid S_0 = s, U' = u')$.

We can arrive at this result by observing that $p(A_0 = a \mid S_0 = s, L_0 = l) = p(A_0 = a \mid S_0 = s) = \pi(a \mid s)$, as the policy logits encode the policy distribution. Using this, we have that

$$p^\pi(A_0 = a \mid S_0 = s, L_0 = l, U' = u') \tag{57}$$

$$= \frac{p^\pi(A_0 = a, U' = u' \mid S_0 = s, L_0 = l)}{p^\pi(U' = u' \mid S_0 = s, L_0 = l)} \tag{58}$$

$$= \frac{p^\pi(U' = u' \mid S_0 = s, A_0 = a, L_0 = l)p(A_0 = a \mid S_0 = s, L_0 = l)}{\sum_{a' \in \mathcal{A}} p^\pi(A_0 = a', U' = u' \mid S_0 = s, L_0 = l)} \tag{59}$$

$$= \frac{p^\pi(U' = u' \mid S_0 = s, A_0 = a)p(A_0 = a \mid S_0 = s)}{\sum_{a' \in \mathcal{A}} p^\pi(U' = u' \mid S_0 = s, A_0 = a',)p(A_0 = a' \mid S_0 = s)} \tag{60}$$

$$= \frac{p^\pi(A_0 = a, U' = u' \mid S_0 = s)}{p^\pi(U' = u' \mid S_0 = s)} \tag{61}$$

$$= p^\pi(A_0 = a \mid S_0 = s, U' = u') \tag{62}$$

$$\square$$

## D.2 Learning contribution coefficients via contrastive classification.

Instead of learning the hindsight distribution, we can estimate the probability ratio $p^\pi(A_t = a \mid S_t = s, U' = u')/\pi(a \mid s)$ directly, by leveraging contrastive classification. Proposition 7 shows that by training a binary classifier $D(a, s, u')$ to distinguish actions sampled from $p^\pi(A_t = a \mid S_t = s, R' = r')$ versus $\pi(a \mid s)$, we can directly estimate the probability ratio.

**Proposition 7.** *Consider the contrastive loss*

$$L = \mathbb{E}_{s,u' \sim \mathcal{T}(s_0, \pi)} \left[ \mathbb{E}_{a \sim p^\pi(a \mid s, u')} [\log D(a, s, u')] + \mathbb{E}_{a \sim \pi(a \mid s)} [\log (1 - D(a, s, u'))] \right], \tag{63}$$

*and $D^*(s, a, u')$ its minimizer. Then the following holds*

$$w(s, a, u') = \frac{D^*(a, s, u')}{1 - D^*(a, s, u')} - 1. \tag{64}$$

*Proof.* Consider a fixed pair $s, u'$. We can obtain the discriminator $D^*(a, s, u')$ that maximizes $\mathbb{E}_{a \sim p^\pi(a \mid s, u')} [\log D(a, s, u')] + \mathbb{E}_{a \sim \pi(a \mid s)} [\log (1 - D(a, s, u'))]$ by taking the point-wise derivative of this objective and equating it to zero:

$$p^\pi(a \mid s, u') \frac{1}{D^*(a, s, u')} - \pi(a \mid s) \frac{1}{1 - D^*(a, s, u')} = 0 \tag{65}$$

$$\Rightarrow \quad \frac{p^\pi(a \mid s, u')}{\pi(a \mid s)} = \frac{D^*(a, s, u')}{1 - D^*(a, s, u')} \tag{66}$$

As for any $(a, b) \in \mathbb{R}^2_{/0}$, the function $f(x) = a \log x + b \log(1 - x)$ achieves its global maximum on support $x \in [0, 1]$ at $\frac{a}{a+b}$, the above maximum is a global maximum. Repeating this argument for all $(s, u')$ pairs concludes the proof. $\square$

We can approximate the training objective of $D$ by sampling $a^{(m)}, s^{(m)}, u'^{(m)}$ along the observed trajectories, while leveraging that we have access to the policy $\pi$, leading to the following loss

$$L = \sum_{m=1}^{M} \left[ -\log D(a^{(m)}, s^{(m)}, u'^{(m)}) - \sum_{a' \in \mathcal{A}} \pi(a' \mid s^{(m)}) \log (1 - D(a', s^{(m)}, u'^{(m)})) \right] \tag{67}$$

**Numerical stability.** Assuming $D$ uses the sigmoid nonlinearity on its outputs, we can improve the numerical stability by computing the logarithm and sigmoid jointly. This results in

$$\log \sigma(x) = -\log(1 + \exp(-x)) \tag{68}$$

$$\log(1 - \sigma(x)) = -x - \log(1 + \exp(-x)) \tag{69}$$

The probability ratio $D/(1 - D)$ can then be computed with

$$\frac{\sigma(x)}{1 - \sigma(x)} = \exp \left[ -\log(1 + \exp(-x)) + x + \log(1 + \exp(-x)) \right] = \exp(x) \tag{70}$$

### D.3 Successor representations.

If we take $u'$ equal to the state $s'$, we can observe that the sum $\sum_{k\geq 1} p^\pi(S_{t+k} = s' \mid s,a)$ used in the contribution coefficients (3) is equal to the successor representation $M(s,a,s')$ introduced by Dayan [30]. Hence, we can leverage temporal differences to learn $M(s,a,s')$, either in a tabular setting [30], or in a deep feature learning setting [31]. Using the successor representations, we can construct the state-based contribution coefficients as $w(s,a,s') = M(s,a,s')/[\sum_{\tilde{a}\in\mathcal{A}} \pi(\tilde{a} \mid s)M(s,\tilde{a},s')] - 1$.

For a rewarding outcome encoding $U$ different from $S$, we can recombine the state-based successor representations to obtain the required contribution coefficients using the following derivation.

$$\sum_{k\geq 0} p^\pi(U_k = u' \mid S_0 = s, A_0 = a) = \sum_{k\geq 0}\sum_{s'\in\mathcal{S}} p^\pi(U_k = u', S_k = s' \mid S_0 = s, A_0 = a) \tag{71}$$

$$= \sum_{k\geq 0}\sum_{s'\in\mathcal{S}} p(U' = u' \mid S' = s')p^\pi(S_k = s' \mid S_0 = s, A_0 = a) \tag{72}$$

$$= \sum_{s'\in\mathcal{S}} p(U' = u' \mid S' = s')\sum_{k\geq 0} p^\pi(S_k = s' \mid S_0 = s, A_0 = a) \tag{73}$$

$$= \sum_{s'\in\mathcal{S}} p(U' = u' \mid S' = s')M(s,a,s') \tag{74}$$

where in the second line we used that $S$ is fully predictive of $U$ (c.f. Definition 2). Note that $p(U' \mid S')$ is policy independent, and can hence be approximated efficiently using offline data. We use this recombination of successor representations in our dynamic programming setup to compute the ground-truth contribution coefficients (c.f. App. E).

## E  Experimental details and additional results

Here, we provide additional details to all experiments performed in the manuscript and present additional results.

### E.1  Algorithms

Algorithm 1 shows pseudocode for the COCOA family of algorithms.

### E.2  Dynamic programming setup

In order to delineate the different policy gradient methods considered in this work, we develop a framework to compute ground-truth policy gradients and advantages as well as ground-truth contribution coefficients. We will first show how we can recursively compute a quantity closely related to the successor representation using dynamic programming which then allows us to obtain expected advantages and finally expected policy gradients.

### E.2.1  Computing ground truth quantities

To compute ground truth quantities, we assume that the environment reaches an absorbing, terminal state $s_\infty$ after $T$ steps for all states $s$, i.e. $p^\pi(S_T = s_\infty|s_0 = s) = 1$. This assumption is satisfied by the linear key-to-door and tree environment we consider. As a helper quantity, we define the successor representation:

$$M(s,a,s',T) = \sum_{k=1}^{T} p^\pi(S_k = s'|S_0 = s, A_0 = a) \tag{75}$$

which captures the cumulative probability over all time steps of reaching state $s'$ when starting from state $S_0 = s$, choosing initial action $A_0 = a$ and following the policy $\pi$ thereafter. We can compute $M(s,a,s',T)$ recursively as

$$M(s,a,s',t) = \sum_{s''\in\mathcal{S}} p(S_1 = s''|S_0 = s, A_0 = a) \sum_{a''\in\mathcal{A}} \pi(a''|s'')(\mathbb{1}_{s''=s'} + M(s'',a'',s',t-1)) \tag{76}$$

where $\mathbb{1}$ is the indicator function and $M(s,a,s',0)$ is initialized to 0 everywhere. Then, the various quantities can be computed exactly as follows.

**Algorithm 1** COCOA

---

**Require:** Initial $\pi$, $h$, episode length $T$, number of episodes $N$, number of pretraining episodes $M$, batch size $L$, $K$ number of pretraining update steps.

1: **for** $i = 1$ to $M$ **do**                                     ▷ Collect random trajectories
2:      Sample $L$ trajectories $\tau = \{\{S_t, A_t, R_t\}_{t=0}^T\}_{l=1}^L$ from a random policy
3:      Add trajectories to the buffer
4: **end for**
5: **for** $j = 1$ to $K$ **do**                                     ▷ Pretrain reward features
6:      Sample batch from buffer
7:      Train reward features $U = f(S, A)$ to predict rewards $R$ via mean squared error
8: **end for**
9: **for** $i = 1$ to $N - M$ **do**
10:     Sample $L$ trajectories $\tau = \{\{S_t, A_t, R_t\}_{t=0}^T\}_{l=1}^L$ from $\pi$
11:     **for** $t = 1$ to $T$ **do**                              ▷ Update the contribution module
12:         **for** $k > t$ **do**
13:             Train $h(A_t|S_t, U_k, \pi_t)$ via cross-entropy loss on all trajectories $\tau$
14:         **end for**
15:     **end for**
16:     **for** $t = 1$ to $T$ **do**                              ▷ Update the policy
17:         **for** $a \in \mathcal{A}$ **do**
18:             $X_a = \nabla_\theta \pi(a \mid S_t) \sum_{k \geq 1} w(S_t, a, U_{t+k}) R_{t+k}$
19:         **end for**
20:         $\hat{\nabla}_\theta^U V^\pi(s_0) = \sum_{t \geq 0} \nabla_\theta \log \pi(A_t \mid S_t) R_t + \sum_{a \in \mathcal{A}} X_a$
21:     **end for**
22: **end for**

---

**Contribution coefficients**

$$w(s, a, u') = \frac{M(s, a, u', T)}{\sum_{a' \in \mathcal{A}} \pi(a' \mid s) M(s, a', u', T)} - 1 \tag{77}$$

where

$$M(s, a, u', T) = \sum_{s' \in \mathcal{S}} M(s, a, s', T) \sum_{a' \in \mathcal{A}} \pi(a' \mid s') \sum_{r'} p(r' \mid s', a') \mathbb{1}_{f(s',a',r')=u'}, \tag{78}$$

similar to the successor representations detailed in Section D. To discover the full state space $\mathcal{S}$ for an arbitrary environment, we perform a depth-first search through the environment transitions (which are deterministic in our setting), and record the encountered states.

**Value function**

$$V(s) = \sum_{a' \in \mathcal{A}} \pi(a'|s) r(s, a') + \sum_{k=1}^T \sum_{s' \in \mathcal{S}} p^\pi(S_k = s'|S_0 = s, \pi) \sum_{a' \in \mathcal{A}} \pi(a'|s') r(s', a') \tag{79}$$

$$= \sum_{a' \in \mathcal{A}} \pi(a'|s) r(s, a') + \sum_{s' \in \mathcal{S}} \sum_{a \in \mathcal{A}} \pi(a \mid s) M(s, a, s', T) \sum_{a' \in \mathcal{A}} \pi(a'|s') r(s', a') \tag{80}$$

where $r(s', a') = \sum_{r'} r' p(r' \mid s', a')$

**Action-value function**

$$Q(s, a) = r(s, a) + \sum_{k=1}^T \sum_{s' \in \mathcal{S}} p^\pi(S_k = s'|S_0 = s, A_0 = a) \sum_{a' \in \mathcal{A}} \pi(a'|s') r(s', a') \tag{81}$$

$$= r(s, a) + \sum_{s' \in \mathcal{S}} M(s, a, s', T) \sum_{a' \in \mathcal{A}} \pi(a'|s') r(s', a') \tag{82}$$

where $r(s', a') = \sum_{r'} r' p(r' \mid s', a')$

### E.2.2 Computing expected advantages

As a next step we detail how to compute the true expected advantage given a potentially imperfect estimator of the contribution coefficient $\hat{w}$, the value function $\hat{V}$ or the action-value function $\hat{Q}$.

**COCOA**

$$\mathbb{E}_\pi\left[\hat{A}^\pi(s,a)\right] = r(s,a) - \sum_{a'\in\mathcal{A}} \pi(a'\mid s)r(s,a') + \mathbb{E}_{T\sim\mathcal{T}(s,\pi)}\left[\sum_{k=1}^{T} \hat{w}(s,a,U_k)R_k\right] \tag{83}$$

$$= r(s,a) - \sum_{a'\in\mathcal{A}} \pi(a'\mid s)r(s,a') + \sum_{u'\in\mathcal{U}}\sum_{k=1}^{T} p^\pi(U_k = u'\mid S_0 = s)\hat{w}(s,a,u')r(u') \tag{84}$$

$$= r(s,a) - \sum_{a'\in\mathcal{A}} \pi(a'\mid s)r(s,a') + \sum_{u'\in\mathcal{U}} \hat{w}(s,a,u')r(u')\sum_{a'\in\mathcal{A}} \pi(a'\mid s)M(s,a',u',T) \tag{85}$$

where $r(u') = \sum_{r'} r'p(r'\mid u')$ and $M(s,a',u',T)$ is defined as in E.2.1.

**Advantage**

$$\mathbb{E}_\pi\left[\hat{A}^\pi(s,a)\right] = Q(s,a) - \hat{V}(s) \tag{86}$$

where $Q$ is the ground truth action-value function obtained following E.2.1.

**Q-critic**

$$\mathbb{E}_\pi\left[\hat{A}^\pi(s,a)\right] = \hat{Q}(s,a) - \sum_{a'\in\mathcal{A}} \pi(a'\mid s)\hat{Q}(s,a') \tag{87}$$

### E.2.3 Computing expected policy gradients

Finally, we show how we can compute the expected policy gradient, $\mathbb{E}_\pi\left[\hat{\nabla}_\theta V^\pi\right]$ of a potentially biased gradient estimator, given the expected advantage $\mathbb{E}_\pi\left[\hat{A}^\pi(s,a)\right]$.

$$\mathbb{E}_\pi\left[\hat{\nabla}_\theta V^\pi\right] = \sum_{k=0}^{T} \mathbb{E}_{S_k\sim\mathcal{T}(s_0,\pi)}\sum_{a\in\mathcal{A}} \nabla_\theta\pi(a|s)\mathbb{E}_\pi\left[\hat{A}^\pi(S_k,a)\right] \tag{88}$$

$$= \sum_{k=0}^{T}\sum_{s\in\mathcal{S}} p^\pi(S_k = s|s_0)\sum_{a\in\mathcal{A}} \nabla_\theta\pi(a|s)\mathbb{E}_\pi\left[\hat{A}^\pi(s,a)\right] \tag{89}$$

Using automatic differentiation to obtain $\nabla_\theta\pi(a|s_0)$, and the quantity $M(s,a,s',T)$ we defined above, we can then compute the expected policy gradient.

$$\mathbb{E}_\pi\left[\hat{\nabla}_\theta V^\pi\right] = \sum_{a\in\mathcal{A}} \nabla_\theta\pi(a|s_0)\mathbb{E}_\pi\left[\hat{A}^\pi(s_0,a)\right] + \tag{90}$$

$$\sum_{s\in\mathcal{S}}\sum_{a_0\in\mathcal{A}} \pi(a_0\mid s_0)M(s_0,a_0,s,T)\sum_{a\in\mathcal{A}} \nabla_\theta\pi(a|s)\mathbb{E}_\pi\left[\hat{A}^\pi(s,a)\right] \tag{91}$$

**Computing the ground-truth policy gradient.** To compute the ground-truth policy gradient, we can apply the same strategy as for Eq. 90 and replace the expected (possibly biased) advantage $\mathbb{E}_\pi\left[\hat{A}^\pi(s_0,a)\right]$ by the ground-truth advantage function, computed with the ground-truth action-value function (c.f. Section E.2.1).

### E.3 Bias, variance and SNR metrics

To analyze the quality of the policy gradient estimators, we use the signal-to-noise ratio (SNR), which we further subdivide into variance and bias. A higher SNR indicates that we need fewer trajectories to estimate accurate policy gradients, hence reflecting better credit assignment. To obtain meaningful

scales, we normalize the bias and variance by the norm of the ground-truth policy gradient.

$$SNR = \frac{\|\nabla_\theta V^\pi\|^2}{\mathbb{E}_\pi \left[ \|\hat{\nabla}_\theta V^\pi - \nabla_\theta V^\pi\|^2 \right]} \tag{92}$$

$$\text{Variance} = \frac{\mathbb{E}_\pi \left[ \|\hat{\nabla}_\theta V^\pi - \mathbb{E}_\pi[\hat{\nabla}_\theta V^\pi]\|^2 \right]}{\|\nabla_\theta V^\pi\|^2} \tag{93}$$

$$\text{Bias} = \frac{\|\mathbb{E}_\pi[\hat{\nabla}_\theta V^\pi] - \nabla_\theta V^\pi\|^2}{\|\nabla_\theta V^\pi\|^2} \tag{94}$$

We compute the full expectations $\mathbb{E}_\pi[\hat{\nabla}_\theta V^\pi]$ and ground-truth policy gradient $\nabla_\theta V^\pi$ by leveraging our dynamic programming setup, while for the expectation of the squared differences $\mathbb{E}_\pi \left[ \|\hat{\nabla}_\theta V^\pi - \nabla_\theta V^\pi\|^2 \right]$ we use Monte Carlo sampling with a sample size of 512. We report the metrics in Decibels in all figures.

**Focusing on long-term credit assignment.** As we are primarily interested in assessing the long-term credit assignment capabilities of the gradient estimators, we report the statistics of the policy gradient estimator corresponding to learning to pick up the key or not. Hence, we compare the SNR, variance and bias of a partial policy gradient estimator considering only $t = 0$ in the outer sum (corresponding to the state with the key) for all considered estimators (c.f. Table 1).

**Shadow training.** Policy gradients evaluated during training depend on the specific learning trajectory of the agent. Since all methods' policy gradient estimators contain noise, these trajectories are likely different for the different methods. As a result, it is difficult to directly compare the quality of the policy gradient estimators, since it depends on the specific data generated by intermediate policies during training. In order to allow for a controlled comparison between methods independent of the noise introduced by different trajectories, we consider a *shadow training* setup in which the policy is trained with the Q-critic method using ground-truth action-value functions. We can then compute the policy gradients for the various estimators on the same shared data along this learning trajectory without using it to actually train the policy. We use this strategy to generate the results shown in Fig. 1B (right), Fig. 2B and Fig. 3C-D.

### E.4 Linear key-to-door environment setup

We simplify the key-to-door environment previously considered by various papers [e.g. 3, 4, 32], to a one-dimensional, linear track instead of the original two-dimensional grid world. This version still captures the difficulty of long-term credit assignment but reduces the computational burden allowing us to thoroughly analyze different policy gradient estimators with the aforementioned dynamic programming based setup. The environment is depicted in Fig. 3A. Here, the agent needs to pick up a key in the first time step, after which it engages in a distractor task of picking up apples which can either be to the left or the right of the agent and which can stochastically assume two different reward values. Finally, the agent reaches a door which it can open with the key to collect a treasure reward.

In our simulations we represent states using a nine-dimensional vector, encoding the relative position on the track as a floating number, a boolean for each item that could be present at the current location (empty, apple left, apple right, door, key, treasure) as well as boolean indicating whether the agent has the key and a boolean for whether the agent does not have the key.

There are four discrete actions the agent can pick at every time step: pick the key, pick to the left, pick to the right and open the door. Regardless of the chosen action, the agent will advance to the next position in the next time step. If it has correctly picked up the key in the first time step and opened the door in the penultimate time step, it will automatically pick up the treasure, not requiring an additional action.

Hung et al. [4] showed that the signal-to-noise ratio (SNR) of the REINFORCE policy gradient [6] for solving the main task of picking up the key can be approximated by

$$\text{SNR}^{\text{REINF}} \approx \frac{\|\mathbb{E}[\hat{\nabla}_\theta^{\text{REINF}} V^\pi]\|^2}{C(\theta)\mathbb{V}[\sum_{t \in T2} R_t] + \text{Tr}\left[\mathbb{V}[\hat{\nabla}_\theta^{\text{REINF}} V^\pi \mid \text{no T2}]\right]}. \tag{95}$$

with $\hat{\nabla}_\theta^{\text{REINF}} V^\pi$ the REINFORCE estimator (c.f. Table 1), $C(\theta)$ a reward-independent constant, $T2$ the set of time steps corresponding to the distractor task, and $\text{Tr}\left[\mathbb{V}[\hat{\nabla}_\theta^{\text{REINF}} V^\pi \mid \text{no T2}]\right]$ the trace of the covariance matrix of the REINFORCE estimator in an equivalent task setup without distractor rewards. Hence, we can adjust the difficulty of the task by increasing the number of distractor rewards and their variance. We perform experiments with environments of length $L \in \{20, 40, \dots, 100\}$ choosing the reward values such that the total distractor reward remains approximately constant. Concretely, distractor rewards are sampled as $r_{\text{distractor}} \sim \mathcal{U}(\{\frac{2}{L}, \frac{18}{L}\})$ and the treasure leads to a deterministic reward of $r_{\text{treasure}} = \frac{4}{L}$.

### E.5 Reward switching setup

While learning to get the treasure in the key-to-door environment requires long term credit assignment as the agent needs to learn to 1) pickup the key and 2) open the door, learning to stop picking up the treasure does not require long term credit assignment, since the agent can simply learn to stop opening the door.

We therefore reuse the linear key-to-door environment of length $L = 40$, with the single difference that we remove the requirement to open the door in order to get the treasure reward. The agent thus needs to perform similar credit assignment to both get the treasure and stop getting the treasure. When applying reward switching, we simply flip the sign of the treasure reward while keeping the distractor reward unchanged.

### E.6 Tree environment setup

We parameterize the tree environment by its depth $d$, the number of actions $n_a$ determining the branching factor, and the *state overlap* $o_s$, defined as the number of overlapping children from two neighbouring nodes. The states are represented by two integers: $i < d$ representing the current level in the tree, and $j$ the position of the state within that level. The root node has state $(i, j) = (0, 0)$, and the state transitions are deterministic and given by $i \leftarrow i + 1$ and $j \leftarrow j(n_a - o_s) + a$, where we represent the action by an integer $0 \le a < n_a$. We assign each state-action pair a reward $r \in \{-2, -1, 0, 1, 2\}$, computed as

$$r(s, a) = \big((\text{idx}(s) + ap + \text{seed}) \bmod n_r\big) - n_r//2 \tag{96}$$

with the modulo operator $\bmod$, the number of reward values $n_r = 5$, $\text{idx}(s)$ a unique integer index for the state $s = (i, j)$, a large prime number $p$, and an environment seed.

To introduce rewarding outcome encodings with varying information content, we group state-action pairs corresponding to the same reward value in $n_g$ groups:

$$u = (\text{idx}(s) + ap + \text{seed}) \bmod (n_r n_g) - n_r//2 \tag{97}$$

**Environment parameters for Figure 2.** For the experiment of Fig. 2, we use 6 actions and a depth of 4. We plotted the variance of the COCOA estimator for encodings $U$ corresponding to $n_g = 4$ and $n_g = 32$.

### E.7 Task interleaving environment setup

We simplify the *task interleaving* environment of Mesnard et al. [3] in the following way: the environment is parameterized by the number of contexts $C$, the number of objects per context $O$, the maximum number of open contexts $B$, and finally the dying probability $1 - \gamma$. In each context, a set of $2O$ objects is given, and a subset of $O$ objects is randomly chosen as the rewarding objects for that context. The task consists in learning, for all contexts, to chose the right objects when presented a pair of objects, one rewarding and one non rewarding. While this is reminiscent of a contextual bandit setting, crucially, the agent receives the reward associated to its choice only at a later time, after potentially making new choices for other contexts, and receiving rewards from other contexts, resulting in an interleaved bandit problem. The agent must thus learn to disentangle the reward associated to each context in order to perform proper credit assignment.

**State and action** At every timestep, there are 2 actions the agent can take: choose right, or choose left.

The state is defined by the following variables:

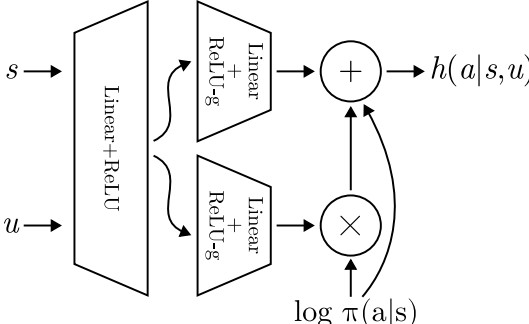

Figure 8: Schematic of the neural network architecture of the hindsight models

- A set of open contexts , i.e. contexts where choices have been made but no reward associated to those choices has been received

- For each open context, a bit (or key) indicating whether the correct choice has been made or not

- The context of the current task

- A pair of objects currently being presented to the agent, if the current context is a query room. Each element of the pair is assigned to either the left side or the right side.

**Transition and reward**    The transition dynamics work as follows. If the agent is in an answer room, then regardless of the action, it receives a context-dependent reward if it also has the key of the corresponding context (i.e. it has made the correct choice when last encountering query room corresponding to the context). Otherwise, it receives 0 reward. Then, the current context is removed from the set of open contexts.

If the agent is in a query room, then the agent is presented with 2 objects, one rewarding and one not, and must choose the side of the rewarding object. Regardless, the agent receives 0 reward at this timestep, and the current context is added to the set of open contexts. Furthermore, if it did make the correct choice, a key associated to the current context is given.

Finally, the next context is sampled. With a probability $\frac{C_o}{B}$ (where $C_o$ is the number of open context), the agent samples uniformly one of the open contexts. Otherwise, the agent samples uniformly one of the non-open contexts. The 2 objects are then also sampled uniformly out of the $O$ rewarding objects and $O$ unrewarding objects. The rewarding object is placed either on the right or left side with uniform probability, and the unrewarding object placed on the other side.

Crucially, there is a probability $1 - \gamma$ of dying, in which case the agent receives the reward but is put into a terminal state at the next timestep.

**Environment parameters**    For all our experiments, we choose $C = 5, B = 3, O = 2$ and $\gamma = 0.95$.

**Visualization of contribution coefficient magnitudes.**    For the heatmaps shown in Fig 4C, we computed each entry as $\sum_{a \in \mathcal{A}} \pi(a|s_t) \cdot |w(s_t, a, r_{t+k})|$ where states $s_t$ and rewards $r_{t+k}$ are grouped by the context of their corresponding room. We average over all query-answer pairs grouped by contexts excluding query-query or answer-answer pairs and only consider answer rooms with non-zero rewards.

### E.8    Training details

### E.8.1    Architecture

We use separate fully-connected ReLU networks to parameterize the policy, value function, and action-value function. For the policy we use two hidden layers of size 64, for the value function and action-value function respectively we use a single hidden layer of size 256. For the hindsight model of both HCA and COCOA we found a simple multilayer perceptron with the state $s$ and rewarding outcome encoding $u'$ as inputs to perform poorly. We hypothesize that this is due to the policy dependence of the hindsight distribution creating a moving target during learning as the

policy changes. Leveraging Proposition 5, we therefore add the policy logits as an extra input to the hindsight network to ease tracking the changing policy. We found good performance using a simple hypernetwork, depicted in Fig. 8 that combines the policy logits with the state and hindsight object inputs through a multiplicative interaction, outputting a logit for each possible action. The multiplicative interaction denoted by $\otimes$ consists of a matrix multiplication of a matrix output by the network with the policy logits, and can be interpreted as *selecting* a combination of policy logits to add to the output channel. In order to allow gating with both positive and negative values, we use a gated version of the ReLU nonlinearity in the second layer which computes the difference between the split, rectified input effectively halving the output dimension:

$$\text{ReLU-g}(x) = \text{ReLU}(x_{0:n/2}) - \text{ReLU}(x_{n/2:n}) \tag{98}$$

with $n$ the dimension of $x$. Gating in combination with the multiplicative interaction is a useful inductive bias for the hindsight model, since for actions which have zero contribution towards the rewarding outcome $u$ the hindsight distribution is equal to the policy. To increase the performance of our HCA+ baseline, we provide both the policy logits and one minus the policy logits to the multiplicative interaction.

### E.8.2 Optimization

For training of all models we use the AdamW optimizer with default parameters only adjusting the learning rates and clipping the global norm of the policy gradient. We use entropy regularization in combination with epsilon greedy to ensure sufficient exploration to discover the optimal policy. To estimate (action-) value functions we use TD($\lambda$) treating each $\lambda$ as a hyperparemeter. For all linear layers we use the default initialization of Haiku [82] where biases are initialized as zero and weights are sample from a truncated Gaussian with standard deviation $\frac{1}{\sqrt{n_{\text{input}}}}$ where $n_{\text{input}}$ is the input dimension of the respective layer and.

### E.8.3 Reward features

Learned reward features should both be fully predictive of the reward (c.f. Theorem 1), and contain as little additional information about the underlying state-action pair as possible (c.f. Theorem 3). We can achieve the former by training a network to predict the rewards given a state-action pair, and take the penultimate layer as the feature $u = f(s, a)$. For the latter, there exist multiple approaches. When using a deterministic encoding $u = f(s, a)$, we can bin the features such that similar features predicting the same reward are grouped together. When using a stochastic encoding $p(U \mid S, A)$ we can impose an information bottleneck on the reward features $U$, enforcing the encoding to discard as much information about $U$ as possible [83, 84]. We choose the deterministic encoding approach, as our Dynamic Programming routines require a deterministic encoding.[4] We group the deterministic rewarding outcome encodings by discretizing the reward prediction network up to the penultimate layer.

**Architecture.** For the neural architecture of the reward prediction network we choose a linear model parameterized in the following way. The input is first multiplicatively transformed by an action-specific mask. The mask is parameterized by a vector of the same dimensionality as the input, and which is squared in order to ensure positivity. A ReLU nonlinearity with straight through gradient estimator is then applied to the transformation. Finally, an action-independent readout weight transforms the activations to the prediction of the reward. The mask parameters are initialized to $1$. Weights of the readout layer are initialized with a Gaussian distribution of mean $0$ and std $\frac{1}{\sqrt{d}}$ where $d$ is the dimension of the input.

**Loss function.** We train the reward network on the mean squared error loss against the reward. To avoid spurious contributions (c.f. 3.3), we encourage the network to learn sparse features that discard information irrelevant to the prediction of the reward, by adding a L1 regularization term to all weights up to the penultimate layer with a strength of $\eta_{L_1} = 0.001$. The readout weights are trained with standard L2 regularization with strength $\eta_{L_2} = 0.03$. For the linear key-to-door experiments, we choose $(\eta_{L_1}, \eta_{L_2}) = (0.001, 0.03)$. For the task interleaving environment, we choose $(\eta_{L_1}, \eta_{L_2}) = (0.05, 0.0003)$.

All weights regularization are treated as weight decay, with the decay applied after the gradient update.

---

[4]In principle, our dynamic programming routines can be extended to allow for probabilistic rewarding outcome encodings and probabilistic environment transitions, which we leave to future work.

Table 2: The range of values swept over for each hyperparameter in a grid search for the linear key-to-door environment and task interleaving environment.

| Hyperparameter | Range |
|---|---|
| lr_agent | $\{0.003, 0.001, 0.0003, 0.0001\}$ |
| lr_hindsight | $\{0.01, 0.003, 0.001\}$ |
| lr_value | $\{0.01, 0.003, 0.001\}$ |
| lr_qvalue | $\{0.01, 0.003, 0.001\}$ |
| lr_features | $\{0.01, 0.003, 0.001\}$ |
| td_lambda_value | $\{0., 0.5, 0.9, 1.\}$ |
| td_lambda_qvalue | $\{0., 0.5, 0.9, 1.\}$ |
| entropy_reg | $\{0.3, 0.1, 0.03, 0.01\}$ |

Table 3: Hyperparameter values on the linear key-to-door environment. The best performing hyperparameters were identical accross all environment lengths. COCOA-r stands for COCOA-return, COCOA-f for COCOA-feature.

| Hyperparameter | COCOA-r | COCOA-f | HCA+ | Q-critic | Advantage | REINFORCE | TrajCV | HCA-return |
|---|---|---|---|---|---|---|---|---|
| lr_agent | 0.0003 | 0.0003 | 0.0003 | 0.0003 | 0.001 | 0.0003 | 0.003 | 0.0001 |
| lr_hindsight | 0.003 | 0.003 | 0.003 | - | - | - | - | 0.001 |
| lr_value | - | - | - | - | 0.001 | - | - | - |
| lr_qvalue | - | - | - | 0.003 | - | - | 0.01 | - |
| lr_features | - | 0.003 | - | - | - | - | - | - |
| td_lambda_value | - | - | - | - | 1. | - | - | - |
| td_lambda_qvalue | - | - | - | 0.9 | - | - | 0.9 | - |

**Pretraining.** To learn the reward features, we collect the first $B_{\text{feature}}$ mini-batches of episodes in a buffer using a frozen random policy. We then sample triplets $(s, a, r)$ from the buffer and train with full-batch gradient descent using the Adam optimizer over $N_{\text{feature}}$ steps. Once trained, the reward network is frozen, and the masked inputs, which are discretized using the element-wise threshold function $\mathbb{1}_{x>0.05}$, are used to train the contribution coefficient as in other COCOA methods. To ensure a fair comparison, other methods are already allowed to train the policy on the first $B_{\text{feature}}$ batches of episodes. For the linear key-to-door experiments, we chose $(B_{\text{feature}}, N_{\text{feature}}) = (30, 20000)$. For the task interleaving environment, we chose $(B_{\text{feature}}, N_{\text{feature}}) = (90, 30000)$.

### E.8.4 Hyperparameters

**Linear key-to-door setup (performance)** For all our experiments on the linear key-to-door environment, we chose a batch size of 8, while using a batch size of 512 to compute the average performance, SNR, bias and variance metrics. We followed a 2-step selection procedure for selecting the hyperparameters: first, we retain the set of hyperparameters for which the environment can be solved for at least 90% of all seeds, given a large amount of training budget. An environment is considered to be solved for a given seed when the probability of picking up the treasure is above 90%. Then, out of all those hyperparameters, we select the one which maximizes the cumulative amount of treasures picked over 10000 training batches. We used 30 seeds for each set of hyperparameters to identify the best performing ones, then drew 30 fresh seeds for our evaluation. The range considered

Table 4: Hyperparameter values on the task interleaving environment.

| Hyperparameter | COCOA-reward | COCOA-feature | HCA+ | Q-critic | Advantage | REINFORCE | TrajCV |
|---|---|---|---|---|---|---|---|
| lr_agent | 0.0003 | 0.0003 | 0.0001 | 0.0003 | 0.001 | 0.001 | 0.001 |
| lr_hindsight | 0.001 | 0.001 | 0.001 | - | - | - | - |
| lr_value | - | - | - | - | 0.001 | - | - |
| lr_qvalue | - | - | - | 0.003 | - | - | 0.001 |
| lr_features | - | 0.001 | - | - | - | - | - |
| entropy_reg | 0.01 | 0.01 | 0.01 | 0.01 | 0.01 | 0.01 | 0.01 |
| td_lambda_value | - | - | - | - | 1. | - | - |
| td_lambda_qvalue | - | - | - | 0.9 | - | - | 0.9 |

Table 5: The entropy regularization value selected for each environment length of the linear key-to-door environment. The values were obtained by linearly interpolating in log-log space between the best performing entropy regularization strength between environment length 20 and 100.

| Environment length | 20 | 40 | 60 | 80 | 100 | 100 (reward-aliasing) |
|---|---|---|---|---|---|---|
| `entropy_reg` | 0.03 | 0.0187 | 0.0142 | 0.0116 | 0.01 | 0.0062 |

for our hyperparameter search can be found in Table 2, and the selected hyperparameters in Table 3. Surprisingly, we found that for the environment length considered, the same hyperparameters were performing best, with the exception of `entropy_reg`. For the final values of `entropy_reg` for each environment length, we linearly interpolated in log-log space between the best performing values, 0.03 for length 20 and 0.01 for 100. The values can be found in Table 5.

**Linear key-to-door setup (shadow training)**    For measuring the bias and variances of different methods in the shadow training setting, we used the best performing hyperparameters found in the performance setting. We kept a batch size of 8 for the behavior policy and shadow training, while using a batch size of 512 during evaluation.

**Reward switching setup**    For the reward switching experiment, we chose hyperparameters following a similar selection procedure as in the linear key-to-door setup, but in the simplified door-less environment of length 40, without any reward switching. We found very similar hyperparameters to work well despite the absence of a door compared to the linear key-to-door setup. However, in order to ensure that noisy methods such as REINFORCE fully converged before the moment of switching the reward, we needed to train the models for 60000 training batches before the switch. To stabilize the hindsight model during this long training period, we added coarse gradient norm clipping by 1.. Furthermore, we found that a slightly decreased learning rate of 0.001 for the Q-critic performed best. Once the best hyperparameters were found, we applied the reward switching to record the speed of adaptation for each algorithm. We kept a batch size of 8 for the behavior policy and shadow training, while using a batch size of 512 during evaluation.

**Task interleaving experiment**    We choose a batch size of 8 and train on 10000 batches, while using a batch size of 512 to compute the average performance. We used 5 seeds for each set of hyperparameters to identify the best performing ones, then drew 5 fresh seeds for our evaluation. The selected hyperparameters can be found in Table 4.

## F    Additional results

We perform additional experiments to corroborate the findings of the main text. Specifically, we investigate how *reward aliasing* affects COCOA-reward and COCOA-feature and show that there is no significant difference in performance between HCA and our simplified version, HCA+.

### F.1    HCA vs HCA+

Our policy gradient estimator for $U = S$ presented in Eq. 4 differs slightly from Eq. 2, the version originally introduced by Harutyunyan et al. [1], as we remove the need for a learned reward model $r(s, a)$. We empirically verify that this simplification does not lead to a decrease in performance in Fig. 9. We run the longest and hence most difficult version of the linear key-to-door environment considered in our experiments and find no significant difference in performance between our simplified version (HCA+) and the original variant (HCA).

### F.2    Learned credit assignment features allow for quick adaptation to a change of the reward function.

Disentangling rewarding outcomes comes with another benefit: When the reward value corresponding to a rewarding outcome changes, e.g. the treasure that was once considered a reward turns out to be poisonous, we only have to relearn the contribution coefficients corresponding to this rewarding outcome. This is in contrast to value-based methods for which potentially many state values are altered. Moreover, once we have access to credit assignment features $u$ that encode rewarding outcomes which generalize to the new setting, the contribution coefficients remain invariant to changes in reward contingencies. For example, if we remember that we need to open the door with

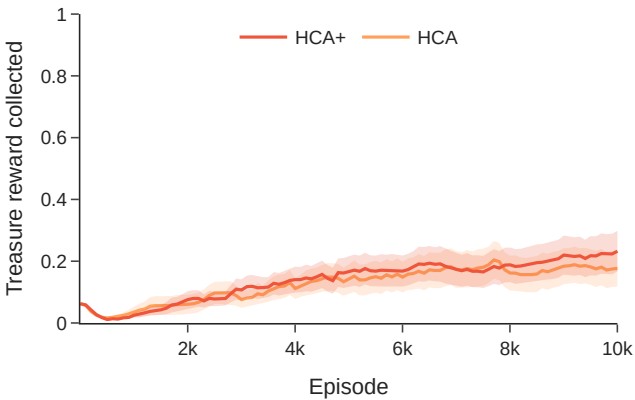

Figure 9: **No performance difference between the original HCA method and our modified variant HCA+.** In the linear key-to-door environment of length 103 both the original HCA method with an additional learned reward model and our simplified version HCA+ perform similarly in terms of performance measured as the percentage of treasure reward collected.

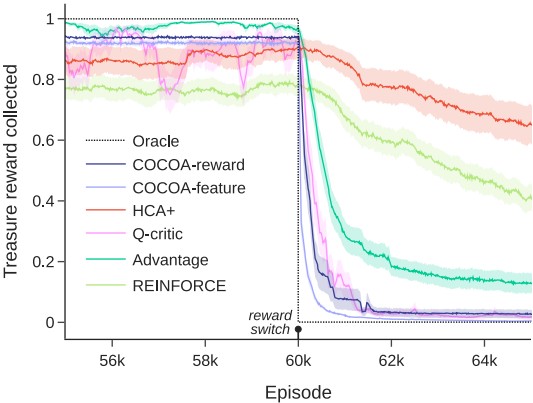

Figure 10: **COCOA-feature quickly adapts the policy to a change of the reward function .** Percentage of treasure reward collected before and after the change in reward contingencies. COCOA-feature quickly adapts to the new setting, as its credit assignment mechanisms generalize to the new setting, whereas COCOA-reward, Q-critic and Advantage need to adjust their models before appropriate credit assignment can take place.

the key to get to the treasure, we can use this knowledge to avoid picking up the key and opening the door when the treasure becomes poisonous.

To illustrate these benefits, we consider a variant of the linear key-to-door environment, where picking up the key always leads to obtaining the treasure reward, and where the treasure reward abruptly changes to be negative after 60k episodes. In this case, COCOA-feature learns to encode the treasure as the same underlying rewarding object before and after the switch, and hence can readily reuse the learned contribution coefficients adapting almost instantly (c.f. Fig. 10). COCOA-reward in contrast needs to encode the poisonous treasure as a new, previously unknown rewarding outcome. Since it only needs to relearn the contribution coefficients for the disentangled subtask of avoiding poison it nevertheless adapts quickly (c.f. Fig. 10). Advantage and Q-critic are both faced with relearning their value functions following the reward switch. In particular, they lack the property of disentangling different rewarding outcomes and all states that eventually lead to the treasure need to be updated.

### F.3 Using returns instead of rewards as hindsight information

In Appendix L, we discussed that HCA-return is a biased estimator in many relevant environments, including our key-to-door and task-interleaving environments. This explains its worse performance compared to our COCOA algorithms. To isolate the difference of using returns instead of rewards as

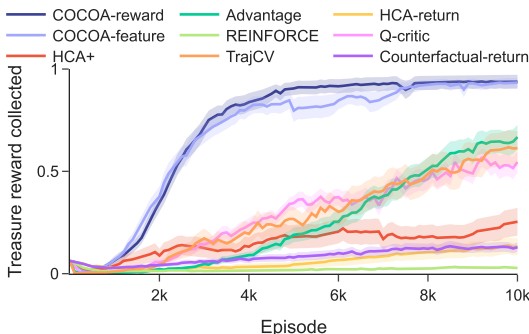

Figure 11: Performance of COCOA and baselines on the main task of picking up the treasure, measured as the average fraction of treasure rewards collected including the *Counterfactual return* method.

hindsight information for constructing the contribution coefficients, we introduce the following new baseline:

$$\sum_{t \geq 0} \sum_a \nabla_\theta \pi(a \mid S_t) \Big( \frac{p^\pi(a \mid S_t, Z_t)}{\pi(a \mid S_t)} - 1 \Big) Z_t \tag{99}$$

Similar to HCA-return, this *Counterfactual Return* variant leverages a hindsight distribution conditioned on the return. Different from HCA-return, we use this hindsight distribution to compute contribution coefficients that can evaluate all counterfactual actions. We prove that this estimator is unbiased.

Figure 11 shows that the performance of HCA-return and Counterfactual return lags far behind the performance of COCOA-reward on the key-to-door environment. This is due to the high variance of HCA-return and Counterfactual Return, and the biasedness of the former. As the return is a combination of all the rewards of a trajectory, it cannot be used to disentangle rewarding outcomes, causing the variance of the distractor subtask to spill over to the subtask of picking up the treasure.

### F.4   Investigation into the required accuracy of the hindsight models

To quantify the relation between the approximation quality of the hindsight models and value functions upon the SNR of the resulting policy gradient estimate, we introduce the following experiment. We start from the ground-truth hindsight models and value functions computed with our dynamic programming setup (c.f. E.2), and introduce a persistent bias into the output of the model by elementwise multiplying the output with $(1+\sigma\epsilon)$, with $\epsilon$ zero-mean univariate Gaussian noise and $\sigma \in \{0.001, 0.003, 0.01, 0.03, 0.1, 0.3\}$ a scalar of which we vary the magnitude in the experiment.

Figure 12 shows the SNR of the COCOA-reward estimator and the baselines, as a function of the perturbance magnitude $\log \sigma$, where we average the results over 30 random seeds. We see that the sensitivity of the COCOA-reward estimator to its model quality is similar to that of Q-critic. Furthermore, the SNR of COCOA-reward remains of better quality compared to Advantage and HCA-state for a wide range of perturbations.

## G   Contribution analysis in continuous spaces and POMDPs

In Section 3.3 we showed that HCA can suffer from spurious contributions, as state representations need to contain detailed features to allow for a capable policy. The same level of detail however is detrimental when assigning credit to actions for reaching a particular state, since at some resolution almost every action will lead to a slightly different outcome. Measuring the contribution towards a specific state ignores that often the same reward could be obtained in a slightly different state, hence overvaluing the importance of past actions. Many commonly used environments, such as pixel-based environments, continuous environments, and partially observable MDPs exhibit this property to a large extent due to their fine-grained state representations. Here, we take a closer look at how spurious contributions arise in continuous environments and Partially Observable MDPs (POMDPs).

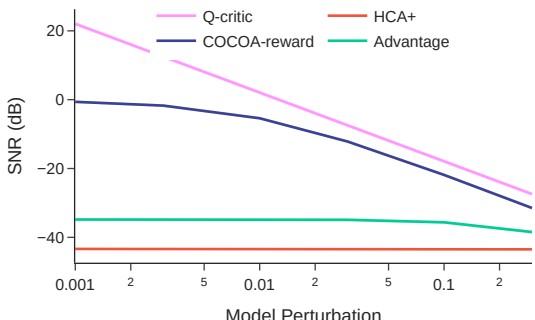

Figure 12: **The improved SNR of the COCOA estimator is robust to imperfect contribution coefficients.** The SNR of the COCOA-reward estimator and the baselines, as a function of the perturbance magnitude $\log \sigma$, where we average the results over 30 random seeds.

### G.1 Spurious contributions in continuous state spaces

When using continuous state spaces, $p^\pi(S_k = s' \mid s, a)$ represents a probability density function (PDF) instead of a probability. A ratio of PDFs $p(X = x)/p(X = x')$ can be interpreted as a likelihood ratio of how likely a sample $X$ will be close to $x$ versus $x'$. Using PDFs, the contribution coefficients with $U = S$ used by HCA result in

$$w(s, a, s') = \frac{\sum_{k \geq 1} p^\pi(S_{t+k} = s' \mid S_t = s, A_t = a)}{\sum_{k \geq 1} p^\pi(S_{t+k} = s' \mid S_t = s)} - 1 = \frac{p^\pi(A_t = a \mid S_t = s, S' = s')}{\pi(a \mid s)} - 1 \tag{100}$$

and can be interpreted as a likelihood ratio of encountering a state 'close' to $s'$, starting from $(s, a)$ versus starting from $s$ and following the policy. The contribution coefficient will be high if $p^\pi(S' = s' \mid s, a)$ has a high probability density at $s'$ conditioned on action $a$, compared to the other actions $a' \neq a$. Hence, the less $p^\pi(S_k = s' \mid s, a)$ and $p^\pi(S_k = s' \mid s, a')$ overlap around $s'$, the higher the contribution coefficient for action $a$. The variance of the distribution $p^\pi(S_k = s' \mid s, a)$, and hence its room for overlap, is determined by how diffuse the environment transitions and policy are. For example, a very peaked policy and nearly deterministic environment transitions leads to a sharp distribution $p^\pi(S_k = s' \mid s, a)$.

The randomness of the policy and environment is a poor measure for the contribution of an action in obtaining a reward. For example, consider the musician's motor system, that learns to both: (i) play the violin and (ii) play the keyboard. We assume a precise control policy and near deterministic environment dynamics, resulting in peaked distributions $p^\pi(S_k = s' \mid s, a)$. For playing the violin, a slight change in finger position significantly influences the pitch, hence the reward function sharply declines around the target state with perfect pitch. For playing the keyboard, the specific finger position matters to a lesser extent, as long as the correct key is pressed, resulting in a relatively flat reward function w.r.t. the precise finger positioning. The state-based contribution coefficients of Eq. 100 result in a high contribution for the action taken in the trajectory for both tasks. For playing violin, this could be a good approximation of what we intuitively think of as a 'high contribution', whereas for the keyboard, this overvalues the importance of the past action in many cases. From this example, it is clear that measuring contributions towards rewarding *states* in continuous state spaces can lead to spurious contributions, as the contributions mainly depend on how diffuse the policy and environment dynamics are, and ignore that different reward structures can require vastly different contribution measures.

### G.2 Deterministic continuous reward functions can lead to excessive variance

Proposition 2 shows that HCA degrades to REINFORCE in environments where each action sequence leads to distinct states. In Section G.1, we discussed that continuous state spaces exhibit this property to a large extend due to their fine-grained state representation. If each environment state has a unique reward value, COCOA can suffer from a similar degradation to HCA. As the rewarding outcome encoding $U$ needs to be fully predictive of the reward (c.f. Theorem 1), it needs to have a distinct encoding for each unique reward, and hence for each state.

With a continuous reward function this can become a significant issue, if the reward function is deterministic. Due to the fine-grained state representation in continuous state spaces, almost every action will lead to a (slightly) different state. When we have a deterministic, continuous reward function, two nearby but distinct states will often lead to nearby but distinct rewards. Hence, to a large extent, the reward contains the information of the underlying state, encoded in its infinite precision of a real value, resulting in COCOA degrading towards the high-variance HCA estimator.

The above problem does not occur when the reward function $p(R \mid S, A)$ is probabilistic, even for continuous rewards. If the variance of $p(R \mid S, A)$ is bigger than zero, the variance of $p(S, A \mid R)$ is also bigger than zero under mild assumptions. This means that it is not possible to perfectly decode the underlying state using a specific reward value $r$, and hence COCOA does not degrade towards HCA in this case. Intuitively, different nearby states can lead to the same sampled reward, removing the spurious contributions. In the following section, we will use this insight to alleviate spurious contributions, even for deterministic reward functions.

### G.3 Smoothing can alleviate excess variance by trading variance for bias

When each state has a unique reward, an unbiased COCOA estimator degrades to the high-variance HCA estimator, since $U$ needs to be fully predictive of the reward, and hence each state needs a unique rewarding outcome encoding. Here, we propose two ways forward to overcome the excess variance of the COCOA estimator in this extreme setting that trade variance for bias.

**Rewarding outcome binning.** One intuitive strategy to avoid that each state has a unique rewarding outcome encoding $U$, is to group rewarding outcome encodings corresponding to nearby rewards together, resulting in discrete bins. As the rewarding outcome encodings now contain less details, the resulting COCOA estimator will have lower variance. However, as now the rewarding outcome encoding is not anymore fully predictive of the reward, the COCOA estimator will be biased. An intimately connected strategy is to change the reward function in the environment to a discretized reward function with several bins. Policy gradients in the new discretized environments will not be exactly equal to the policy gradients in the original environment, however for fine discretizations, we would not expect much bias. Similarly for binning the rewarding outcome encodings, when we group few nearby rewarding outcome encodings together, we expect a low bias, but also a low variance reduction. Increasing the amount of rewarding outcomes we group together we further lower the bias, at a cost of an increasing bias, hence creating a bias-variance trade-off.

**Stochastic rewarding outcomes.** We can generalize the above binning technique towards rewarding outcome encodings that bin rewards stochastically. In Section G.2, we discussed that when the reward function is probabilistic, the excessive variance problem is less pronounced, as different states can lead to the same reward. When dealing with a deterministic reward function, we can introduce stochasticity in the rewarding outcome encoding to leverage the same principle and reduce variance at the cost of increasing bias. For example, we can introduce the rewarding outcome encoding $U \sim \mathcal{N}(R, \sigma)$, with $\mathcal{N}$ corresponding to a Gaussian distribution. As this rewarding outcome encoding is not fully predictive of the reward, it will introduce bias. We can control this bias-variance trade-off with the variance $\sigma$: a small sigma corresponds to a sharp distribution, akin to a fine discretization in the above binning strategy, and hence a low bias. Increasing $\sigma$ leads to more states that could lead to the same rewarding outcome encoding, hence lowering the variance at the cost of increasing the bias.

**Implicit smoothing by noise perturbations or limited capacity networks.** Interestingly, the above strategy of defining stochastic rewarding outcomes is equivalent to adjusting the training scheme of the hindsight model $h(a \mid s, u')$ by adding noise to the input $u'$. Here, we take $U$ equal to the (deterministic) reward $R$, but add noise to it while training the hindsight model. Due to the noise, the hindsight model cannot perfectly decode the action $a$ from its input, resulting in the same effects as explicitly using stochastic rewarding outcomes. Adding noise to the input of a neural network is a frequently used regularization technique. Hence, an interesting route to investigate is whether other regularization techniques on the hindsight model, such as limiting its capacity, can result in a bias-variance trade-off for HCA and COCOA.

**Smoothing hindsight states for HCA.** We can apply the same (stochastic) binning technique to hindsight states for HCA, creating a more coarse-grained state representation for backward credit assignment. However, the bias-variance trade-off is more difficult to control for states compared to rewards. The sensitivity of the reward function to the underlying state can be big in some regions of state-space, whereas small in others. Hence, a uniform (stochastic) binning of the state-space

will likely result in sub-optimal bias-variance trade-offs, as it will be too coarse-grained in some regions causing large bias, whereas too fine-grained in other regions causing a low variance-reduction. Binning rewards in contrast does not suffer from this issue, as binning is directly performed in a space relevant to the task.

Proposition 8 provides further insight on what the optimal smoothing or binning for HCA looks like. Consider the case where we have a discrete reward function with not too many distinct values compared to the state space, such that COCOA-reward is a low-variance gradient estimator. Proposition 8 shows that we can recombine the state-based hindsight distribution $p^\pi(A_0 = a \mid S_0 = s, S' = s')$ into the reward-based hindsight distribution $p^\pi(A_0 = a \mid S_0 = s, R' = r')$, by leveraging a smoothing distribution $\sum_{s' \in \mathcal{S}} p^\pi(S' = s' \mid S_0 = s, R' = r')$. When we consider a specific hindsight state $s''$, this means that we can obtain the low-variance COCOA-reward estimator, by considering all states $S'$ that could have lead to the same reward $r(s'')$, instead of only $s''$. Hence, instead of using a uniform stochastic binning strategy with e.g. $S' \sim \mathcal{N}(s'', \sigma)$, a more optimal binning strategy is to take the reward structure into account through $p^\pi(S' \mid S_0 = s, R' = r(s''))$

**Proposition 8.**

$$p^\pi(A_0 = a \mid S_0 = s, R' = r') = \sum_{s' \in \mathcal{S}} p^\pi(S' = s' \mid S_0 = s, R' = r')p^\pi(A_0 = a \mid S_0 = s, S' = s') \tag{101}$$

*Proof.* As $A_0$ is d-separated from $R'$ conditioned on $S'$ and $S_0$ (c.f. Fig. 6b) we can use the implied conditional independence to prove the proposition:

$$p^\pi(A_0 = a \mid S_0 = s, R' = r') \tag{102}$$

$$= \sum_{s' \in \mathcal{S}} p^\pi(S' = s', A_0 = a \mid S_0 = s, R' = r') \tag{103}$$

$$= \sum_{s' \in \mathcal{S}} p^\pi(S' = s' \mid S_0 = s, R' = r')p^\pi(A_0 = a \mid S_0 = s, S' = s', R' = r') \tag{104}$$

$$= \sum_{s' \in \mathcal{S}} p^\pi(S' = s' \mid S_0 = s, R' = r')p^\pi(A_0 = a \mid S_0 = s, S' = s') \tag{105}$$

$\square$

### G.4 Continuous action spaces

In the previous section, we discussed how continuous state spaces can lead to spurious contributions resulting high variance for the HCA estimator. Here, we briefly discuss how COCOA can be applied to continuous action spaces, and how Proposition 2 translates to this setting.

**Gradient estimator.** We can adjust the COCOA gradient estimator of Eq. 4 towards continuous action spaces by replacing the sum over $a'\mathcal{A}$ by an integral over the action space $\mathcal{A}'$.

$$\hat{\nabla}_\theta^U V^\pi(s_0) = \sum_{t \geq 0} \nabla_\theta \log \pi(A_t \mid S_t)R_t + \int_\mathcal{A} \mathrm{d}a\, \nabla_\theta \pi(a \mid S_t) \sum_{k \geq 1} w(S_t, a, U_{t+k})R_{t+k} \tag{106}$$

In general, computing this integral is intractable. We can approximate the integral by standard numerical integration methods, introducing a bias due to the approximation. Alternatively, we introduced in App. C.3 another variant of the COCOA gradient estimator that samples independent actions $A'$ from the policy instead of summing over the whole action space. This variant can readily be applied to continuous action spaces, resulting in

$$\hat{\nabla}_\theta V^\pi = \sum_{t \geq 0} \nabla_\theta \log \pi(A_t \mid S_t)R_t + \frac{1}{M}\sum_m \nabla_\theta \log \pi(a^m \mid S_t) \sum_{k=1}^\infty w(S_t, a^m, U_{t+k})R_{t+k} \tag{107}$$

where we sample $M$ actions independently from $a^m \sim \pi(\cdot \mid S_t)$. This gradient estimator is unbiased, but introduces extra variance through the sampling of actions.

**Spurious contributions.** Akin to discrete action spaces, HCA in continuous action spaces can suffer from spurious contributions when distinct action sequences lead to unique states. In this case, previous actions can be perfectly decoded from the hindsight state, and we have that the probability density function $p^\pi(A_0 = a \mid S_0 = s, S' = s')$ is equal to the Dirac delta function $\delta(a = a_0)$, with $a_0$ the action taken in the trajectory that led to $s'$. Substituting this Dirac delta function into the policy gradient estimator of Eq. 106 results in the high-variance REINFORCE estimator. When we use the COCOA estimator of Eq. 107 using action sampling, HCA will even have a higher variance compared to REINFORCE.

### G.5 Partially Observable MDPs

In many environments, agents do not have access to the complete state information $s$, but instead get observations with incomplete information. Partially observable Markov decision processes (POMDPs) formalize this case by augmenting MDPs with an observation space $\mathcal{O}$. Instead of directly observing Markov states, the agent now acquires an observation $o_t$ at each time step. The probability of observing $o$ in state $s$ after performing action $a$ is given by $p_o(o \mid s, a)$.

A popular strategy in deep RL methods to handle partial observability is learning an internal state $x_t$ that summarizes the observation history $h_t = \{o_{t'+1}, a_{t'}, r_{t'}\}_{t'=0}^{t-1}$, typically by leveraging recurrent neural networks [85–88]. This internal state is then used as input to a policy or value function. This strategy is intimately connected to estimating *belief states* [89]. The observation history $h_t$ provides us with information on what the current underlying Markov state $s_t$ is. We can formalize this by introducing the *belief state* $b_t$, which captures the sufficient statistics for the probability distribution over the Markov states $s$ conditioned on the observation history $h_t$. Theoretically, such belief states can be computed by doing Bayesian probability calculus using the environment transition model, reward model and observation model:

$$p_B(s \mid b_t) = p(S_t = s \mid H_t = h_t) \tag{108}$$

Seminal work has shown that a POMDP can be converted to an MDP, using the belief states $b$ as new Markov states instead of the original Markov states $s$ [90]. Now the conventional optimal control techniques can be used to solve the belief-state MDP, motivating the use of standard RL methods designed for MDPs, combined with learning an internal state $x$.

**HCA suffers from spurious contributions in POMDPs.** As the internal state $x$ summarizes the complete observation history $h$, past actions can be accurately decoded based on $h$, causing HCA to degrade to the high-variance REINFORCE estimator (c.f. Proposition 2). Here, the tension between forming good state representations for enabling capable policies and good representations for backward credit assignment is pronounced clearly. To enable optimal decision-making, a good internal state $x$ needs to encode which underlying Markov states the agent most likely occupies, as well as the corresponding uncertainty. To this end, the internal state needs to incorporate information about the full history. However, when using the same internal state for backward credit assignment, this leads to spurious contributions, as previous actions are encoded directly into the internal state.

To make the spurious contributions more tangible, let us consider a toy example where the state space $\mathcal{S}$ consists of three states $x$, $y$ and $z$. We assume the internal state represents a belief state $b = \{b^1, b^2\}$, which is a sufficient statistic for the belief distribution:

$$p(S = x \mid b) = b^1, \quad p(S = y \mid b) = b^2, \quad p(S = z \mid b) = 1 - b^1 - b^2 \tag{109}$$

We assume that $b_t$ is deterministically computed from the history $h_t$, e.g. by an RNN. Now consider that at time step $k$, our belief state is $b_k = \{0.5, 0.25\}$ and we get a reward that resulted from the Markov state $x$. As the belief states are deterministically computed, the distribution $p^\pi(B' = b' \mid b, a)$ is a Dirac delta distribution. Now consider that the action $a$ does not influence the belief distribution over the rewarding state $x$, but only changes the belief distribution over the non-rewarding states (e.g. $B_k = \{0.5, 0.23\}$ instead of $B_k = \{0.5, 0.25\}$ when taking action $a'$ instead of $a$). As the Dirac delta distributions $p^\pi(B' = b' \mid b, a)$ for different $a$ do not overlap, we get a high contribution coefficient for the action $a_0$ that was taken in the actual trajectory that led to the belief state $b'$, and a low contribution coefficient for all other actions, even though the actions did not influence the distribution over the rewarding state.

The spurious contributions originate from measuring contributions towards reaching a certain internal belief state, while ignoring that the same reward could be obtained in different belief states as well.

Adapting Proposition 8 towards these internal belief states provides further insight on the difference between HCA and COCOA-reward:

$$p^\pi(A_0 = a \mid X_0 = x, R' = r') = \sum_{X' \in \mathcal{X}} p^\pi(X' = x' \mid X_0 = x, R' = r')p^\pi(A_0 = a \mid X_0 = x, X' = x')$$

(110)

Here, we see that the contribution coefficients of COCOA-reward take into account that the same reward can be obtained while being in different internal belief states $x'$ by averaging over them, while HCA only considers a single internal state.

## H  Learning contribution coefficients from non-rewarding observations

### H.1  Latent learning

Building upon HCA [1], we learn the contribution coefficients (3) by approximating the hindsight distribution $p^\pi(a \mid s, u')$. The quality of this model is crucial for obtaining low-bias gradient estimates. However, its training data is scarce, as it is restricted to learn from on-policy data, and rewarding observations in case the reward or rewarding object is used as encoding $U$. We can make the model that approximates the hindsight distribution less dependent on the policy by providing the policy logits as input (c.f. Proposition 5). Enabling COCOA-reward or COCOA-feature to learn from non-rewarding states is a more fundamental issue, as in general, the rewarding outcome encodings corresponding to zero rewards do not share any features with those corresponding to non-zero rewards.

Empowering COCOA with *latent learning* is an interesting direction to make the learning of contribution coefficients more sample efficient. We refer to *latent learning* as learning useful representations of the environment structure without requiring task information, which we can then leverage for learning new tasks more quickly [91]. In our setting, this implies learning useful representations in the hindsight model without requiring rewarding observations, such that when new rewards are encountered, we can quickly learn the corresponding contribution coefficients, leveraging the existing representations.

### H.2  Optimal rewarding outcome encodings for credit assignment.

Theorem 3 shows that the less information a rewarding outcome encoding $U$ contains, the lower the variance of the corresponding COCOA gradient estimator (4). Latent learning on the other hand considers the sample-efficiency and corresponding bias of the learned contribution coefficients: when the hindsight model can learn useful representations with encodings $U$ corresponding to zero rewards, it can leverage those representations to quickly learn the contribution coefficients for rewarding outcome encodings with non-zero rewards, requiring less training data to achieve a low bias.

These two requirements on the rewarding outcome encoding $U$ are often in conflict. To obtain a low-variance gradient estimator, $U$ should retain as little information as possible while still being predictive of the reward. To enable latent learning for sample-efficient learning of the contribution coefficients, the hindsight model needs to pick up on recurring structure in the environment, requiring keeping as much information as possible in $U$ to uncover the structural regularities. Using the state as rewarding outcome encoding is beneficial for enabling latent learning, as it contains rich information on the environment structure, but results in spurious contributions and a resulting high variance. Using the rewards or rewarding object as rewarding outcome encoding removes spurious contributions resulting in low variance, but renders latent learning difficult.

A way out of these conflicting pressures for an optimal rewarding outcome encoding is to use rewards or rewarding objects as the optimal encoding for low variance, but extract these contribution coefficients from models that allow for sample-efficient, latent learning. One possible strategy is to learn probabilistic world models [20, 21, 88] which can be done using both non-rewarding and rewarding observations, and use those to approximate the contribution coefficients of Eq. 3. Another strategy that we will explore in more depth is to learn hindsight models based on the state as rewarding outcome encoding to enable latent learning, but then recombine those learned hindsight models to obtain contribution coefficients using the reward or rewarding object as $U$.

### H.3  Counterfactual reasoning on rewarding states

HCA results in spurious contributions because it computes contributions towards reaching a precise rewarding state, while ignoring that the same reward could be obtained in other (nearby) states.

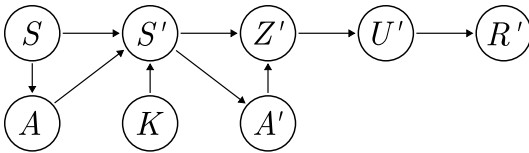

Figure 13: Schematic of the graphical model used in our variational information bottleneck approach.

Proposition 9 (a generalization of Proposition 8), shows that we can reduce the spurious contributions of the state-based contribution coefficients by leveraging counterfactual reasoning on rewarding states. Here, we obtain contribution coefficients for a certain rewarding outcome encoding (e.g. the rewarding object) by considering which other states $s'$ could lead to the same rewarding outcome, and averaging over the corresponding coefficients $w(s, a, s')$.

**Proposition 9.** *Assuming $S'$ is fully predictive of $U'$, we have that*

$$w(s, a, u') = \sum_{s' \in \mathcal{S}} p^\pi(S' = s' \mid S_0 = s, U' = u') w(s, a, s') \tag{111}$$

The proof follows the same technique as that of Proposition 8. Proposition 8) shows that it is possible to learn state-based contribution coefficients, enabling latent learning, and obtain a low-variance COCOA estimator by recombining the state-based contribution coefficients into coefficients with less spurious contributions, if we have access to the generative model $p^\pi(S' = s' \mid S_0 = s, U' = u')$. This model embodies the counterfactual reasoning on rewarding states: 'which other states $s'$ are likely considering I am currently in $u'$ and visited state $s$ somewhere in the past'.

In general, learning this generative model is as difficult or more difficult than approximating the hindsight distribution $p^\pi(a \mid s, u')$, as it requires rewarding observations as training data. Hence, it is not possible to directly use Proposition 9 to combine latent learning with low-variance estimators. In the following section, we propose a possible way forward to circumvent this issue.

### H.4 Learning credit assignment representations with an information bottleneck

Here, we outline a strategy where we learn a latent representation $Z$ that retains useful information on the underlying states $S$, and crucially has a latent space structure such that $p^\pi(Z' = z' \mid S = s, U' = u')$ is easy to approximate. Then, leveraging Proposition 9 (and replacing $S'$ by $Z'$) allows us to learn a hindsight representation based on $Z$, enabling latent learning, while reducing the spurious contributions by counterfactual reasoning with $p^\pi(z' \mid s, u')$.

To achieve this, we use the Variational Information Bottleneck approach [83, 92], closely related to the $\beta$ Variational Autoencoder [93]. Fig. 13 shows the graphical model with the relations between the various variables and encoding: we learn a probabilistic encoding $p(Z \mid S, A; \theta)$ parameterized by $\theta$, and we assume that the latent variable $Z$ is fully predictive of the rewarding outcome encoding $U$ and reward $R$. We aim to maximize the mutual information $\mathcal{I}(Z'; S', A' \mid S, U')$ under some information bottleneck. We condition on $S$ and $U'$, to end up later with a decoder and prior variational model that we can combine with Proposition 9.

Following rate-distortion theory, Alemi et al. [92] consider the following tractable variational bounds on the mutual information

$$H - D \leq \mathcal{I}(Z'; S', A' \mid S, U') \leq \text{Rate} \tag{112}$$

with entropy $H$, distortion $D$ and rate defined as follows:

$$H = -\mathbb{E}_{S', A', S, U'}[\log p^\pi(S', A' \mid S, U')] \tag{113}$$

$$D = -\mathbb{E}_{S, U'} \left[ \mathbb{E}_{S', A' \mid S, U'} \left[ \int \mathrm{d}z' p(z' \mid s', a'; \theta) \log q(s', a' \mid s, z'; \psi) \right] \right] \tag{114}$$

$$\text{Rate} = \mathbb{E}_{S, U'} \left[ \mathbb{E}_{S', A' \mid S, U'} \left[ D_{\mathrm{KL}}\big(p(z' \mid s', a'; \theta) \| q(z' \mid s, u'; \phi)\big) \right] \right] \tag{115}$$

where the *decoder* $q(s', a' \mid s, z'; \psi)$ is a variational approximation to $p(s', a' \mid s, z')$, and the *marginal* $q(z' \mid s, u'; \phi)$ is a variational approximation to the true marginal $p(z' \mid s, u')$. The distortion $D$ quantifies how well we can decode the state-action pair $(s', a')$ from the encoding

$z'$, by using a decoder $q(s', a' \mid s, z'; \psi)$ parameterized by $\psi$. The distortion is reminiscent of an autoencoder loss, and hence encourages the encoding $p(z' \mid s', a')$ to retain as much information as possible on the state-action pair $(s', a')$. The rate measures the average KL-divergence between the encoder and variational marginal. In information theory, this rate measures the extra number of bits (or nats) required to encode samples from $Z'$ with an optimal code designed for the variational marginal $q(z' \mid s, u'; \phi)$.

We can use the rate to impose an information bottleneck on $Z'$. If we constrain the rate to Rate $\leq a$ with $a$ some positive constant, we restrict the amount of information $Z'$ can encode about $(S', A')$, as $p(z' \mid s', a'; \theta)$ needs to remain close to the marginal $q(z' \mid s, u'; \phi)$, quantified by the KL-divergence. We can maximize the mutual information $\mathcal{I}(Z'; S', A' \mid S, U')$ under the information bottleneck by minimizing the distortion under this rate constraint. Instead of imposing a fixed constraint on the rate, we can combine the rate and distortion into an unconstrained optimization problem by leveraging the Lagrange formulation

$$\min_{\theta, \phi, \psi} D + \beta \text{Rate} \tag{116}$$

Here, the $\beta$ parameter determines the strength of the information bottleneck. This formulation is equivalent to the $\beta$ Variational Autoencoder [93], and for $\beta = 1$ we recover the Variational Autoencoder [94].

To understand why this information bottleneck approach is useful to learn credit assignment representations $Z$, we examine the rate in more detail. We can rewrite the rate as

$$\text{Rate} = \mathbb{E}_{S, U'} \left[ D_{\text{KL}} \big( p(z' \mid s, u') \| q(z' \mid s, u'; \phi) \big) \right] + \tag{117}$$

$$\mathbb{E}_{S, U', Z'} \left[ D_{\text{KL}} \big( p(s', a' \mid s, z') \| p(s', a' \mid s, u') \big) \right] \tag{118}$$

Hence, optimizing the information bottleneck objective (116) w.r.t. $\phi$ fits the variational marginal $q(z' \mid s, u'; \phi)$ to the true marginal $p(z' \mid s, u')$ induced by the encoder $p(z' \mid s', a'; \theta)$. Proposition 9 uses this true marginal to recombine coefficients based on $Z$ into coefficients based on $U$. Hence, by optimizing the information bottleneck objective, we learn a model $q(z' \mid s, u'; \phi)$ that approximates the true marginal, which we then can use to obtain contribution coefficients with less spurious contributions by leveraging Proposition 9. Furthermore, minimizing the rate w.r.t. $\theta$ shapes the latent space of $Z'$ such that the true marginal $p(z' \mid s, u')$ moves closer towards the variational marginal $q(z' \mid s, u'; \phi)$.

In summary, the outlined information bottleneck approach for learning a credit assignment representation $Z$ is a promising way forward to merge the powers of latent learning with a low-variance COCOA gradient estimator. The distortion and rate terms in the information bottleneck objective of Eq. 116 represent a trade-off parameterized by $\beta$. Minimizing the distortion results in a detailed encoding $Z'$ with high mutual information with the state-action pair $(S', A')$, which can be leveraged for latent learning. Minimizing the rate shapes the latent space of $Z'$ in such way that the true marginal $p(z' \mid s, u')$ can be accurately approximated within the variational family of $q(z' \mid s, u'; \phi)$, and fits the parameters $\phi$ resulting in an accurate marginal model. We can then leverage the variational marginal $q(z' \mid s, u'; \phi)$ to perform counterfactual reasoning on the rewarding state encodings $Z'$ according to Proposition 9, resulting in a low-variance COCOA-estimator.

# I  Contribution analysis and causality

## I.1  Causal interpretation of COCOA

COCOA is closely connected to causality theory [42], where the contribution coefficients (3) correspond to performing $Do - interventions$ on the causal graph to estimate their effect on future rewards. To formalize this connection with causality, we need to use a new set of tools beyond conditional probabilities, as causation is in general not the same as correlation. We start with representing the MDP combined with the policy as a directed acyclic graphical model (c.f. Fig. 6a in App. C.1). In causal reasoning, we have two different 'modes of operation'. On the one hand, we can use observational data, corresponding to 'observing' the states of the nodes in the graphical model, which is compatible with conditional probabilities measuring correlations. On the other hand, we can perform *interventions* on the graphical model, where we manually set a node, e.g. $A_t$ to a specific value $a$ independent from its parents $S_t$, and see what the influence is on the probability distributions of other nodes in the graph, e.g. $R_{t+1}$. These interventions are formalized with *do-calculus* [42],

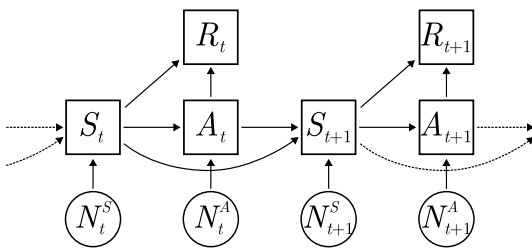

Figure 14: Structural causal model (SCM) of the MDP. Squares represent deterministic functions and circles random variables.

where we denote an intervention of putting a node $V_i$ equal to $v$ as $\mathrm{Do}(V_i = v)$, and can be used to investigate causal relations.

Using the graphical model of Fig. 6b that abstracts time, we can use do-interventions to quantify the causal contribution of an action $A_t = a$ taken in state $S_t = s$ upon reaching the rewarding outcome $U' = u'$ in the future as,

$$w_{\mathrm{Do}}(s, a, u') = \frac{p^\pi\big(U' = u' \mid S_t = s, \mathrm{Do}(A_t = a)\big)}{\sum_{\tilde{a} \in \mathcal{A}} \pi(\tilde{a} \mid s) p^\pi\big(U' = u' \mid S_t = s, \mathrm{Do}(A_t = \tilde{a})\big)} - 1. \tag{119}$$

As conditioning on $S$ satisfies the *backdoor criterion* [42] for $U'$ w.r.t. $A$, the interventional distribution $p^\pi\big(U' = u' \mid S_t = s, \mathrm{Do}(A_t = a)\big)$ is equal to the observational distribution $p^\pi\big(U' = u' \mid S_t = s, A_t = a\big)$. Hence, the causal contribution coefficients of Eq. 119 are equal to the contribution coefficients of Eq. 3 used by COCOA.

### I.2 Extending COCOA to counterfactual interventions

Within causality theory, *counterfactual reasoning* goes one step further than causal reasoning, by incorporating the hindsight knowledge of the external state of the world in its reasoning. Applied to COCOA, this more advanced counterfactual reasoning would evaluate the query: 'How does taking action $a$ influence the probability of reaching a rewarding outcome, compared to taking alternative actions $a'$, *given everything else remains the same*'. To formalize the difference between causal and counterfactual reasoning, we need to convert the causal DAG of Figure 6a into a structural causal model (SCM), as shown in Figure 14. The SCM expresses all conditional distributions as deterministic functions with independent noise variables $N$, akin to the reparameterization trick [94]. In causal reasoning, we perform do-interventions on nodes, which is equivalent to cutting all incoming edges to a node. To compute the resulting probabilities, we still use the prior distribution over the noise variables $N$. Counterfactual reasoning goes one step further. First, it infers the posterior probability $p^\pi(\{N\} \mid T = \tau)$, with $\{N\}$ the set of all noise variables, given the observed trajectory $\tau$. Then it performs a Do-intervention as in causal reasoning. However now, to compute the resulting probabilities on the nodes, it uses the posterior noise distribution combined with the modified graph.

One possible strategy to estimate contributions using the above counterfactual reasoning is to explicitly estimate the posterior noise distribution and combine it with forward dynamics models to obtain the counterfactual probability of reaching specific rewarding outcomes. Leveraging the work of Buesing et al. [43] is a promising starting point for this direction of future work. Alternatively, we can avoid explicitly modeling the posterior noise distribution, by leveraging the hindsight distribution combined with the work of Mesnard et al. [3]. Here, we aim to learn a parameterized approximation $h$ to the counterfactual hindsight distribution $p^\pi_\tau\big(A_t = a \mid S_t = s, U' = u'\big)$, where the $\tau$-subscript indicates the counterfactual distribution incorporating the noise posterior. Building upon the approach of Mesnard et al. [3], we can learn $h(a \mid s', s, \Phi_t(\tau))$ to approximate the counterfactual hindsight distribution, with $\Phi_t(\tau)$ a summary statistic trained to encapsulate the information of the posterior noise distribution $p(\{N\} \mid T = \tau)$. Mesnard et al. [3] show that such a summary statistic $\Phi_t(\tau)$ can be used to amortize the posterior noise estimation if it satisfies the following two conditions: (i) it needs to provide useful information for predicting the counterfactual hindsight distribution and (ii) it needs to be independent from the action $A_t$. We can achieve both characteristics by (i) training $h(a \mid s', s, \Phi_t(\tau))$ on the hindsight action classification task and backpropagating the gradients to $\Phi_t(\tau)$, and (ii) training $\Phi_t$ on an *independence maximization* loss $\mathcal{L}_{\mathrm{IM}}(s)$, which is minimized iff

Table 6: Comparison of discounted policy gradient estimators

| Method | Policy gradient estimator ($\hat{\nabla}_\theta V^\pi(s_0)$) |
|---|---|
| REINFORCE | $\sum_{t\geq 0} \gamma^t \nabla_\theta \log \pi(A_t \mid S_t) \sum_{k\geq 0} \gamma^k R_{t+k}$ |
| Advantage | $\sum_{t\geq 0} \gamma^t \nabla_\theta \log \pi(A_t \mid S_t) \left( \sum_{k\geq 0} \gamma^k R_{t+k} - V_\gamma^\pi(S_t) \right)$ |
| Q-critic | $\sum_{t\geq 0} \gamma^t \sum_{a\in\mathcal{A}} \nabla_\theta \pi(a \mid S_t) Q_\gamma^\pi(S_t, a)$ |
| COCOA | $\sum_{t\geq 0} \gamma^t \nabla_\theta \log \pi(A_t \mid S_t) R_t + \sum_{a\in\mathcal{A}} \nabla_\theta \pi(a \mid S_t) \sum_{k\geq 1} \gamma^k w_\gamma(S_t, a, U_{t+k}) R_{t+k}$ |
| HCA+ | $\sum_{t\geq 0} \gamma^t \nabla_\theta \log \pi(A_t \mid S_t) R_t + \sum_{a\in\mathcal{A}} \nabla_\theta \pi(a \mid S_t) \sum_{k\geq 1} \gamma^k w_\gamma(S_t, a, S_{t+k}) R_{t+k}$ |

$A_t$ and $\Phi_t$ are conditionally independent given $S_t$. An example is to minimize the KL divergence between $\pi(a_t \mid s_t)$ and $p(a_t \mid s_t, \Phi_t(\tau))$ where the latter can be approximated by training a classifier $q(a_t \mid s_t, \Phi_t(\tau))$. Leveraging this approach to extend COCOA towards counterfactual interventions is an exciting direction for future research.

## J   Contribution analysis with temporal discounting

As discussed in App. B, we can implicitly incorporate discounting into the COCOA framework by adjusting the transition probabilities to have a fixed probability of $(1 - \gamma)$ of transitioning to the absorbing state $s_\infty$ at each time step.

We can also readily incorporate explicit time discounting into the COCOA framework, which we discuss here. We consider now a discounted MDP defined as the tuple $(\mathcal{S}, \mathcal{A}, p, p_r, \gamma)$, with discount factor $\gamma \in [0, 1]$, and $(\mathcal{S}, \mathcal{A}, p, p_r)$ as defined in Section 2. The discounted value function $V_\gamma^\pi(s) = \mathbb{E}_{T\sim\mathcal{T}(s,\pi)}[\sum_{t=0}^\infty \gamma^t R_t]$ and action value function $Q_\gamma^\pi(s, a) = \mathbb{E}_{T\sim\mathcal{T}(s,a,\pi)}[\sum_{t=0}^\infty \gamma^t R_t]$ are the expected discounted return when starting from state $s$, or state $s$ and action $a$ respectively. Table 6 shows the policy gradient estimators of $V_\gamma^\pi(s)$ for REINFORCE, Advantage and Q-critic.

In a discounted MDP, it matters at which point in time we reach a rewarding outcome $u$, as the corresponding rewards are discounted. Hence, we adjust the contribution coefficients to

$$w_\gamma(s, a, u') = \frac{\sum_{k\geq 1} \gamma^k p^\pi(U_{t+k} = u' \mid S_t = s, A_t = a)}{\sum_{k\geq 1} \gamma^k p^\pi(U_{t+k} = u' \mid S_t = s)} - 1 \tag{120}$$

$$= \frac{p_\gamma^\pi(A_t = a \mid S_t = s, U' = u')}{\pi(a \mid s)} - 1 \tag{121}$$

Here, we define the discounted hindsight distribution $p_\gamma^\pi(A_t = a \mid S_t = s, U' = u')$ similarly to the undiscounted hindsight distribution explained in App. C.1, but now using a different probability distribution on the time steps $k$: $p_\beta(K = k) = (1 - \beta)\beta^{k-1}$, where we take $\beta = \gamma$. We can readily extend Theorems 1-4 to the explicit discounted setting, by taking $\beta = \gamma$ instead of the limit of $\beta \to 1$, and using the discounted COCOA policy gradient estimator shown in Table 6.

To approximate the discounted hindsight distribution $p_\gamma^\pi(A_t = a \mid S_t = s, U' = u')$, we need to incorporate the temporal discounting into the classification cross-entropy loss:

$$L_\gamma = \mathbb{E}_\pi \left[ \sum_{t\geq 0} \sum_{k\geq 1} \gamma^k CE\big(h_\gamma(\cdot \mid S_t, U_{t+k}), \delta(a = A_t)\big) \right], \tag{122}$$

with $CE$ the cross-entropy loss, $h_\gamma(\cdot \mid S_t, U_{t+k})$ the classification model that approximates the discounted hindsight distribution, and $\delta(a = A_t)$ a one-hot encoding of $A_t$.

## K   Bootstrapping with COCOA

Here, we show how COCOA can be combined with n-step returns, and we make a correction to Theorem 7 of Harutyunyan et al. [1] which considers n-step returns for HCA.

Consider the graphical model of Fig. 15a where we model time, $K$, as a separate node in the graphical model (c.f. App. C.1). To model $n$-step returns, we now define the following prior probability on $K$,

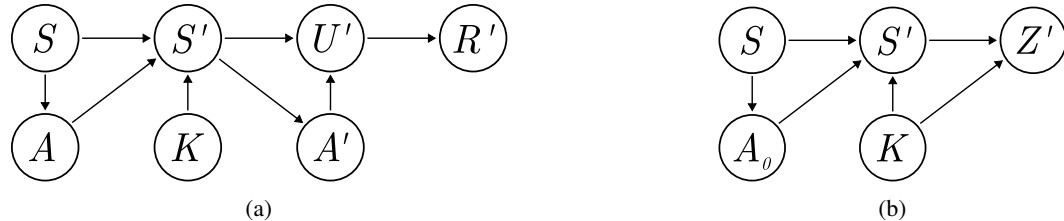

(a)  (b)

Figure 15: (a) Graphical model where we abstract time. (b) Graphical model implicitly used in the proof of Theorem 7 in Harutyunyan et al. [1].

parameterized by $\beta$:

$$p_{n,\beta}(K = k) = \begin{cases} \frac{\beta^k}{z} & \text{if } 1 \le k \le n-1 \\ 0 & \text{else} \end{cases} \qquad z = \beta \frac{1 - \beta^{n-1}}{1 - \beta} \tag{123}$$

Using $p^\pi_{n,\beta}$ as the probability distribution induced by this graphical model, we have that

$$p^\pi_{n,\beta}(U' = u \mid S = s, A = a) = \sum_k p^\pi_{n,\beta}(U' = u, K = k \mid S = s, A = a) \tag{124}$$

$$= \sum_k p_{n,\beta}(K = k) p^\pi_{n,\beta}(U' = u \mid S = s, A = a, K = k) \tag{125}$$

$$= \frac{1 - \beta}{\beta(1 - \beta^{n-1})} \sum_{k=1}^{n-1} \beta^k p^\pi(U_k = u \mid S_0 = s, A_0 = a) \tag{126}$$

We introduce the following contribution coefficients that we will use for n-step returns

$$w_{n,\beta}(s, a, u') = \frac{\sum_{k=1}^{n-1} \beta^k p^\pi(U_k = u' \mid S_0 = s, A_0 = a)}{\sum_{k=1}^{n-1} \beta^k p^\pi(U_k = u' \mid S_0 = s)} \tag{127}$$

$$= \frac{p^\pi_{n,\beta}(U' = u' \mid S = s, A = a)}{p^\pi_{n,\beta}(U' = u' \mid S = s)} - 1 = \frac{p^\pi_{n,\beta}(A = a \mid S = s, U' = u')}{\pi(a \mid s)} - 1 \tag{128}$$

$$w_n(s, a, s') = \frac{p^\pi(S_n = s' \mid S_0 = s, A_0 = a)}{p^\pi(S_n = s' \mid S_0 = s)} - 1 = \frac{p^\pi(A = a \mid S = s, S_n = s')}{\pi(a \mid s)} - 1 \tag{129}$$

Now we are ready to prove the n-step return theorem for the discounted MDP setting. We can recover the undiscounted setting by taking the limit of $\gamma \to 1_-$.

**Theorem 10.** *Consider state $s$ and action $a$ for which it holds that $\pi(a \mid s) > 0$ and take $\beta$ equal to the discount factor $\gamma \in [0, 1]$. Furthermore, assume that the rewarding outcome encoding $u = f(s, a, r)$ is fully predictive of the reward (c.f. Definition 2). Then the advantage $A^\pi(s, a) = Q^\pi(s, a) - V^\pi(s)$ is equal to*

$$A^\pi(s, a) = r(s, a) - r^\pi(s) + \mathbb{E}_{\mathcal{T}(s, \pi)} \left[ \sum_{k=1}^{n-1} \gamma^k w_{n,\beta}(s, a, U_k) R_k + \gamma^n w_n(s, a, S_n) V^\pi(S_n) \right]$$

*with $r(s, a)$ the reward model and $r^\pi(s) = \sum_a \pi(a \mid s) r(s, a)$.*

*Proof.* We start with the action-value function $Q^\pi$, and will subtract the value $V^\pi$ to obtain the result on the advantage function.

$$Q(s, a) = \mathbb{E}_{T \sim \mathcal{T}(s, a, \pi)} \left[ \sum_{k \ge 1} \gamma^k R_k \right] \tag{130}$$

$$= r(s, a) + \sum_{r' \in \mathcal{R}} \sum_{k \ge 1} \gamma^k p^\pi(R_k = r' \mid s, a) r' \tag{131}$$

$$= r(s,a) + \sum_{r' \in \mathcal{R}} \sum_{u' \in \mathcal{U}} \sum_{k=1}^{n-1} \gamma^k p^\pi(R_k = r', U_k = u' \mid s, a) r' + \tag{132}$$

$$\gamma^n \sum_{s' \in \mathcal{S}} p^\pi(S_n = s' \mid s, a) V^\pi(s') \tag{133}$$

$$= r(s,a) + \sum_{r' \in \mathcal{R}} \sum_{u' \in \mathcal{U}} \sum_{k=1}^{n-1} \gamma^k p^\pi(R' = r' \mid U' = u') p^\pi(U_k = u' \mid s, a) r' + \tag{134}$$

$$\gamma^n \sum_{s' \in \mathcal{S}} p^\pi(S_n = s' \mid s, a) V^\pi(s') \tag{135}$$

$$= r(s,a) + \sum_{u' \in \mathcal{U}} r(u') \sum_{k=1}^{n-1} \gamma^k p^\pi(U_k = u' \mid s, a) + \gamma^n \sum_{s' \in \mathcal{S}} p^\pi(S_n = s' \mid s, a) V^\pi(s') \tag{136}$$

$$= r(s,a) + \sum_{u' \in \mathcal{U}} r(u') \sum_{k=1}^{n-1} \gamma^k p^\pi(U_k = u' \mid s) \frac{\sum_{k'=1}^{n-1} \gamma^{k'} p^\pi(U_{k'} = u \mid s, a)}{\sum_{k'=1}^{n-1} \gamma^{k'} p^\pi(U_{k'} = u \mid s)} + \tag{137}$$

$$\gamma^n \sum_{s' \in \mathcal{S}} p^\pi(S_n = s' \mid s) \frac{p^\pi(S_n = s' \mid s, a)}{p^\pi(S_n = s' \mid s)} V^\pi(s') \tag{138}$$

$$= r(s,a) + \sum_{u' \in \mathcal{U}} r(u') \sum_{k=1}^{n-1} \gamma^k p^\pi(U_k = u' \mid s)(w_{n,\beta}(s, a, u') + 1) \tag{139}$$

$$+ \gamma^n \sum_{s' \in \mathcal{S}} p^\pi(S_n = s' \mid s)(w_n(s, a, s') + 1) V^\pi(s') \tag{140}$$

$$\tag{141}$$

where we use that $U'$ is fully predictive of the reward $R'$, and define $r(u') = \sum_{r' \in \mathcal{R}} p(R' = r' \mid U' = u') r'$ By subtracting the value function, we get

$$A(s,a) = r(s,a) - r^\pi(s) + \sum_{u' \in \mathcal{U}} r(u') \sum_{k=1}^{n-1} \gamma^k p^\pi(U_k = u' \mid s) w_{n,\beta}(s, a, u') \tag{142}$$

$$+ \gamma^n \sum_{s' \in \mathcal{S}} p^\pi(S_n = s' \mid s) w_n(s, a, s') V^\pi(s') \tag{143}$$

$$= r(s,a) - r^\pi(s) + \mathbb{E}_{\mathcal{T}(s,\pi)} \left[ \sum_{k=1}^{n-1} \gamma^k w_{n,\beta}(s, a, U_k) R_k + \gamma^n w_n(s, a, S_n) V^\pi(S_n) \right] \tag{144}$$

$$\square$$

Finally, we can sample from this advantage function to obtain an n-step COCOA gradient estimator, akin to Theorem 1.

Note that we require to learn the state-based contribution coefficients $w_n(s, a, s')$ to bootstrap the value function into the n-step return, as the value function requires a Markov state $s'$ as input instead of a rewarding outcome encoding $u'$. Unfortunately, these state-based contribution coefficients will suffer from spurious contributions, akin to HCA, introducing a significant amount of variance into the n-step COCOA gradient estimator. We leave it to future research to investigate whether we can incorporate value functions into an n-step return, while using rewarding-outcome contribution coefficients $w(s, a, u')$ instead of state-based contribution coefficients $w_n(s, a, s')$.

**Learning the contribution coefficients.** We can learn the contribution coefficients $w_{\beta,n}(s, a, u')$ with the same strategies as described in Section 3, but now with training data from $n$-step trajectories instead of complete trajectories. If we use a discount $\gamma \neq 1$, we need to take this discount factor into account in the training distribution or loss function (c.f. App. J).

**Correction to Theorem 7 of Harutyunyan et al. [1].** Harutyunyan et al. [1] propose a theorem similar to Theorem 10, with two important differences. The first one concerns the distribution on $K$ in the graphical model of Fig. 15a. Harutyunyan et al. [1] implicitly use this graphical model, but with a different prior probability distribution on $K$:

$$p_{n,\beta}^{HCA}(K = k) = \begin{cases} \beta^{k-1}(1-\beta) & \text{if } 1 \leq k \leq n-1 \\ \beta^{n-1} & \text{if } k = n \\ 0 & \text{else} \end{cases} \tag{145}$$

The graphical model combined with the distribution on $K$ defines the hindsight distribution $p_{n,\beta,HCA}^{\pi}(A = a \mid S = s, S' = s')$. The second difference is the specific $Q$-value estimator Harutyunyan et al. [1] propose. They use the hindsight distribution $p_{n,\beta,HCA}^{\pi}(A = a \mid S = s, S' = s')$ in front of the value function (c.f. Theorem 10), which considers that $s'$ can be reached at any time step $k \sim p_{n,\beta}^{HCA}(k)$, whereas Theorem 10 uses $w_n(s, a, s')$ which considers that $s'$ is reached exactly at time step $k = n$.

To the best of our knowledge, there is an error in the proposed proof of Theorem 7 by Harutyunyan et al. [1] for which we could not find a simple fix. For the interested reader, we briefly explain the error. One indication of the problem is that for $\beta \to 1$, all the probability mass of $p_{n,\beta}^{HCA}(K = k)$ is concentrated at $k = n$, hence the corresponding hindsight distribution $p_{n,\beta,HCA}^{\pi}(A = a \mid S = s, S' = s')$ considers only hindsight states $s'$ encountered at time $k = n$. While this is not a mathematical error, it does not correspond to the intuition of a 'time independent hindsight distribution' the authors provide. In the proof itself, a conditional independence relation is assumed that does not hold. The authors introduce a helper variable $Z$ defined on the state space $\mathcal{S}$, with a conditional distribution

$$\mu_k(Z = z \mid S' = s') = \begin{cases} \delta(z = s') & \text{if } 1 \leq k \leq n-1 \\ \tilde{d}^{\pi}(z \mid s') & \text{if } k = n \end{cases} \tag{146}$$

with the normalized discounted visit distribution $\tilde{d}^{\pi}(z \mid s') = (1 - \gamma) \sum_k \gamma^k p^{\pi}(S_k = z \mid S_0 = s)$. We can model this setting as the graphical model visualized in Fig. 15b. In the proof (last line on page 15 in the supplementary materials of Harutyunyan et al. [1]), the following conditional independence is used:

$$p^{\pi}(A_0 = a \mid S_0 = s, S' = s', Z = z) = p^{\pi}(A_0 = a \mid S_0 = s, S' = s') \tag{147}$$

However, Fig. 15b shows that $S'$ is a collider on the path $A_0 \to S' \leftarrow K \to Z$. Hence, by conditioning on $S'$ we open this collider path, making $A_0$ dependent on $Z$ conditioned on $S_0$ and $S'$, thereby invalidating the assumed conditional independence. For example, if $Z$ is different from $S'$, we know that $K = n$ (c.f. Eq. 146), hence $Z$ can contain information about action $A_0$, beyond $S'$, as $S'$ ignores at which point in time $s'$ is encountered.

# L   HCA-return is a biased estimator in many relevant environments

## L.1   HCA-return

Besides HCA-state, Harutyunyan et al. [1] introduced HCA-return, a policy gradient estimator that leverages the hindsight distribution conditioned on the return:

$$\sum_{t \geq 0} \nabla_\theta \log \pi(A_t \mid S_t)\left(1 - \frac{\pi(A_t \mid S_t)}{p^{\pi}(A_t \mid S_t, Z_t)}\right)Z_t \tag{148}$$

When comparing this estimator with COCOA-reward, we see two important differences: (i) HCA-return uses a hindsight function conditioned on the return instead of individual rewards, and (ii) HCA-return leverages the hindsight function as an action-dependent baseline for a Monte Carlo policy gradient estimate, instead of using it for contribution coefficients to evaluate counterfactual actions. Importantly, the latter difference causes the HCA-return estimator to be biased in many environments of relevance, even when using the ground-truth hindsight distribution.

## L.2   HCA-return can be biased

An important drawback of HCA-return is that it can be biased, even when using the ground-truth hindsight distribution. Theorem 2 of Harutyunyan et al. 2019, considering the unbiasedness HCA-return, is valid under the assumption that for any possible random return $Z$ for all possible trajectories

starting from state $s$, it holds that $p^\pi(a \mid s, z) > 0$. This restrictive assumption requires that for each observed state-action pair $(s_t, a_t)$ along a trajectory, all counterfactual returns $Z$ resulting from a counterfactual trajectory starting from $s_t$ (not including $a_t$) result in $p^\pi(a_t \mid s_t, Z) > 0$. This implies that all returns (or rewarding states) reachable from $s_t$ should also be reachable from $(s_t, a_t)$.

Consider the following bandit setting as a simple example where the above assumption is not satisfied. The bandit has two arms, with a reward of 1 and $-2$, and a policy probability of $\frac{2}{3}$ and $\frac{1}{3}$ respectively. The advantage for both arms is 1 and $-2$. Applying eq. 6 from Harutyunyan et al. results in $A^\pi(s, a_1) = (1 - \frac{2}{3}) = \frac{1}{3}$ and $A^\pi(s, a_2) = -2(1 - 1/3) = -4/3$. This shows that the needed assumptions for an unbiased HCA-return estimator can be violated even in simple bandit settings.

# M   Additional details

## M.1   Author contributions

This paper was a collaborative effort of all shared first authors working closely together. To do this fact better justice we give an idea of individual contributions in the following.

**Alexander Meulemans**$^*$. Original idea, conceptualizing the theory and proving the theorems, conceptual development of the algorithms, experiment design, implementation of main method and environments, debugging, neural network architecture design, running experiments, connecting the project to existing literature, writing of manuscript, first draft and supplementary materials, feedback to the figures.

**Simon Schug**$^*$. Conceptual development of the algorithms, experiment design, implementation of main method, baselines and environments, neural network architecture design, debugging, tuning and running experiments, writing of manuscript, creation of figures, writing of supplementary materials.

**Seijin Kobayashi**$^*$. Conceptual development of the algorithms, experiment design, implementation of environments, baselines, main method and Dynamic Programming-based ground-truth methods, debugging, tuning and running experiments, feedback to the manuscript, writing of supplementary materials.

**Nathaniel Daw**. Regular project meetings, conceptual input and feedback for method and experimental design, connecting the project to existing literature, feedback to the manuscript and figures.

**Gregory Wayne**. Senior project supervision, conceptualising of the project idea, conceptual development of the algorithms, regular project meetings, technical and conceptual feedback for method and experimental design, connecting the project to existing literature, feedback to the manuscript and figures.

## M.2   Compute resources

We used Linux workstations with Nvidia RTX 2080 and Nvidia RTX 3090 GPUs for development and conducted hyperparameter searches and experiments using 5 TPUv2-8, 5 TPUv3-8 and 1 Linux server with 8 Nvidia RTX 3090 GPUs over the course of 9 months. All of the final experiments presented take less than a few hours to complete using a single Nvidia RTX 3090 GPU. In total, we spent an estimated amount of 2 GPU months.

## M.3   Software and libraries

For the results produced in this paper we relied on free and open-source software. We implemented our experiments in Python using JAX [95, Apache License 2.0] and the Deepmind Jax Ecosystem [82, Apache License 2.0]. For experiment tracking we used wandb [96, MIT license] and for the generation of plots we used plotly [97, MIT license].

