# OpenReview forum: "Would I have gotten that reward? Long-term credit assignment by counterfactual contribution analysis"
_NeurIPS.cc/2023/Conference — NeurIPS 2023 spotlight_

### Official Review · Reviewer_UxzB · 2023-06-23

**Soundness:** 3 good
**Presentation:** 2 fair
**Contribution:** 3 good
**Rating:** 7
**Confidence:** 4

**Summary:**

This paper lays out a goal of addressing a weakness in HCA that HCA can confuse the contributions of actions to reaching a state, increasing the variance of gradient estimation. Instead, the COCOA method proposed in this paper generalizes HCA to use any feature that is predictive of rewards, such as the reward or reward-predicting features. Thus, it allows for selecting an appropriate rewarding outcome to disentangle the contributions of each action from the observed rewards. The paper derives a new unbiased gradient estimator (given the exact hindsight model). Experiments are then conducted to show that when these models are known, they can reduce variance in cases not handled by standard control variates.

**Strengths:**

The biggest strength of this paper is thorough experimentation to illustrate when COCOA reduces variance when the hindsight models are known.

The experiments and environment clearly show the COCOA method's impact in the cases: when distractor rewards are encountered during the episode and when the distractor rewards are of the same magnitude as the main reward. These experiments show clear evidence that selecting the appropriate disentangling factor is essential to reducing variance.

The experiments to show the SNR as a function of credit assignment distance highlights how each gradient estimator degrades as the episode length and distance between action and rewarding stimulus increases. However, this should be evaluated at longer distances as the slopes and shapes of the curves are difficult to observe in this range.

**Weaknesses:**

The biggest weakness of this paper is that there is no investigation into how accurate the estimates of hindsight models need to be to reduce variance. Past variance reduction techniques for policy gradients have proven useless in practice (Tucker et al. 2018). So it is crucial to understand the practical limitations when developing a new method.


The comparison to q critic, when q is known, is odd. Because if q is known, then the natural gradient will be known (Kakade, 2001). It should be made precise what sources of variance these experiments are measuring.

The bias can be much higher for COCOA methods than other methods (Figure 3 (D)), and it needs to be clearly discussed since this method is supposed to be unbiased.

A more relevant baseline instead of the Advantage method would be the doubly robust control variate for policy gradient methods (Huang and Jiang, 2020). This has a significant variance reduction compared to standard baselines in policy gradient. The comparison would also better highlight the limits of information between the two approaches.

The success/failure of an algorithm depends greatly on hyperparameter choices. It needs to be clear how each method's hyperparameters were chosen. Since this paper is trying to argue that COCOA has better sample efficiency, it should be made clear in what exact context this claim is being made. Furthermore, if one wants to say that COCOA has better information utilization through measuring performance, then the hyperparameters for the baseline methods need to be tuned to their absolute maximum. As the paper currently stands, the experiment design is insufficient to make this argument, even if it is likely true.

Though not poor, the writing in the paper has room for improvement. One of the best ways for this paper to have a bigger impact is to make it written so that most RL people can easily understand the lessons. So I recommend that the authors spend time revising the writing for a future or camera-ready version of the paper.

___minor comments__
what are the decibel units? This is nonstandard in ML, so it is worth specifying.

Table1: In the Advantage method, $v^\pi$ should just be $v$ because the baseline can be any state-dependent function

Although common in the ML community $\nabla_\theta$ is a mathematical symbol for a directional derivative. However, it is used in this work as a partial derivative. Further, since the gradient is a function that takes a derivative with respect to all inputs and partial derivates are the quantities of interest, it makes mathematical sense to use mathematical symbols for partial differentiation, not gradients.

Line 100: "it has a failure mode of practical importance" —say what this is.

Section 3.1 Define reward outcomes first. It is unclear what they mean, making it hard to interpret. It also needs to be made clear when or why one would use different choices of f. Giving a few motivating examples would help here.

Line 133: Eq 1. Check the JMLR guidelines for standard practice for referencing equations. It should be just (1) not Eq, and equation blocks should only be numbered if they are referenced.

Line 146: missing for all k?

Definition 1. Remark on definition 1 to clarify what is needed to satisfy this property in practice.

Theorem 1: It is important to remark that the gradient estimator is unbiased only if the counterfactual term is known precisely, which is unlikely in practice. This would be like saying the DDPG is unbiased, which is misleading because it is only true if Q is known.

Theorem 4. "equals" should be changed because the expression is not a statement of equality but proportionality.

Line 246: "SNR indicates… fewer trajectories…" This is not universally true. It can require an exponential number of trajectories to get a gradient estimate, and lower variance estimates will not help this.



REFERENCES
Huang, Jiawei, and Nan Jiang. "From importance sampling to doubly robust policy gradient." International Conference on Machine Learning. PMLR, 2020.

Kakade, Sham M. "A natural policy gradient." Advances in neural information processing systems 14 (2001).

Tucker, George, et al. "The mirage of action-dependent baselines in reinforcement learning." International conference on machine learning. PMLR, 2018.

**Questions:**

From Theorem 3 it wasn't clear if one should expect the same trend to hold in other cases. Would we expect the same ordering of variances in more settings?

**Limitations:**

The authors do mention that further work is needed on developing methods to learn the hindsight models so they do acknowledge this limitation. But a pilot study on the same environment is really warranted in this work.

---

> ### Author Rebuttal · Authors · 2023-08-09
>
> Thank you for your comprehensive remarks and for the useful feedback that we incorporated in a revised version of our paper. We address your main concerns below.
>
> ## 1. SOTA Control Variate baseline.
>
> We thank the reviewer for suggesting an extra baseline that leverages hindsight information for a control variate (CV). Doubly-Robust CV requires learning an environment dynamics model to contruct a cheap estimate of the Q-value gradient, which we consider out-of-scope for our work. Instead, we investigate TrajCV (Cheng, Yan and Boots, 2019), a related SOTA CV method that also leverages hindsight information but does not require an environment model.
>
> Figure 1 in the attached pdf shows the results of the TrajCV method on our linear key-to-door environment. We observe that it improves upon the advantage baseline but still falls short of the performance of the COCOA methods with ground-truth contribution coefficients. The main reason for the performance gap between the COCOA methods and TrajCV is that the control variate of TrajCV can only reduce the variance caused by the stochastic policy and not from stochastic environment dynamics, whereas COCOA can reduce both. In the linear key-to-door environment, a large part of the variance is due to the stochastic distractor rewards. The value functions used in TrajCV necessarily need to approximate the average returns to use as control variates, whereas COCOA can ignore the stochastic distractor rewards completely when applicable, by multiplying them with contribution coefficients equal to zero. Interestingly, this points to a potential benefit of the multiplicative interactions between contribution coefficients and rewards in the COCOA estimators, compared to the additive interaction of the control variates.
>
> ## 2. Investigation into the needed accuracy of the hindsight models
> We include a new experiment quantifying the relation between the approximation quality of the hindsight models and value functions upon the SNR of the resulting policy gradient estimate. We start from the ground-truth hindsight models and value functions computed with our dynamic programming setup, and introduce a persistent bias into the output of the model by elementwise multiplying the output with $(1 + \sigma \epsilon)$, with $\epsilon$ zero-mean univariate Gaussian noise and $\sigma \in \{0.001, 0.003, 0.01, 0.03, 0.1,0.3\}$ a scalar of which we vary the magnitude in the experiment.
>
> Figure 1a in the attached PDF shows the SNR of the COCOA-reward estimator and the baselines, as a function of the perturbance magnitude $\log\sigma$, where we average the results over 30 random seeds. We see that the sensitivity of the COCOA-reward estimator to its model quality is similar to that of Q-critic. Furthermore, the SNR of COCOA-reward remains of better quality compared to Advantage and HCA-state for a wide range of perturbations.
>
> ## 3. Clarification on hyperparameter tuning
> Appendix E.6.4 and Table 2 provide details on our rigorous hyperparameter tuning. We emphasize that we used the same extensive hyperparameter search for all methods and baselines, and hence we are confident that our empirical results support the main claims we made in our paper.
>
> ## 4. Generality of Theorem 3
> Theorem 3 considers only a limited setting. To extend these theoretical results, we performed an extensive empirical evaluation on random MDPs with dense rewards considering the full policy gradient estimates. Figure 2 and Figure 6 in Appendix C.5 indicate that the relative ordering of Theorem 3 holds in more general settings as well. This is further corroborated by our results on the key-to-door environment in Figure 3. It is an open question whether it is possible to formally extend Theorem 3 to a more general setting.
>
> ## 5. Q-critic baseline
> Our main motivation to include the Q-critic when the Q-value is learned, is to have a baseline that considers all counterfactual actions for a given trajectory, similar to the COCOA estimators. The ground-truth Q-critic is an interesting upper bound for the other methods. Compared to the ground-truth policy gradient, the only source of variance in the ground-truth Q-critic is the sampled trajectory states $S_t$ where the policy is evaluated. In Figure 3, we see that the performance of ground-truth COCOA-reward matches the Q-critic upper bound. We note that in the bias-variance and SNR plot of Figure 3, we consider the policy gradient on the first decision moment of picking up the key, as this determines the performance on the key-to-door task. In this case, the ground-truth Q-critic is equal to the ground-truth policy gradient, as the first state is always the same.
>
> ## 6. Biasedness of COCOA with learned coefficients
> Theorem 1 considers only the case with ground-truth contribution coefficients. When we approximate the contribution coefficients, this can lead to a biased COCOA estimator. Theorem 1 nonetheless is an important result, as it was not clear beforehand which rewarding outcome encodings $U$ lead to a an unbiased policy gradient estimator *with ground-truth contribution coefficients*. We thank the reviewer for pointing out this possible confusion, and we will clarify it in the main text by adjusting the text of Theorem 1 and discussing the potential biasedness in section 3.4.
> ## 7. Remaining comments
> - We will run the experiments of Figure 1B for longer environment lengths for the final version of our paper.
> - We thank the reviewer for the detailed suggestions to improve the clarity of our manuscript and will incorporate them in our final version.

---

> > ### Comment · Reviewer_UxzB · 2023-08-14
> > **Updated score**
> >
> > Thank you for your detailed response as it addressed my main concerns (sans seeing a new draft). I will positively update my score.

---

### Official Review · Reviewer_5aMg · 2023-07-09

**Soundness:** 2 fair
**Presentation:** 2 fair
**Contribution:** 3 good
**Rating:** 6
**Confidence:** 4

**Summary:**

This paper proposes Counterfactual Contribution Analysis (COCOA), an RL credit assignment approach inspired by the Hindsight Credit Assignment (HCA) family of algorithms.

The paper notes that in some instances, the previously proposed State-HCA approach can degrade to as high a variance as REINFORCE due to using the future state for obtaining hindsight weights.

Instead of estimating hindsight contributions of actions based on future states (as in State-HCA), this paper introduces the idea of a rewarding outcome which is a function of a future state, action, and reward (s’, a’, r’) tuple. The paper proposes two choices for the rewarding outcome– the reward itself or the reward predictive features of the state-action pair (COCOA-reward and COCOA-feature). The paper provides intuitions on how the COCOA approaches should lead to lower-variance policy gradients compared to State-HCA due to reduced encoding of state information.

Experiments in a key-to-door environment shows that the COCOA approaches perform much better than State-HCA, REINFORCE (with and without baseline), and all-actions policy gradient on Q-values.


**Strengths:**

**S1.** Hindsight-based approaches are a promising solution to the credit assignment problem in RL, which is of great interest to the research community.

**S2.** While identifying the high-variance failure case of State-HCA may not be a novel contribution (see W2), the proposed solution to use rewards or reward predictive features instead of states is novel. The solution idea is simple, well-motivated with examples (e.g., Figure 2 and text below proposition 2), and fits the HCA family of approaches well. Focusing on features that can be useful for HCA is an interesting direction and opens a way to improve previous HCA approaches.

**S3.** The paper is well-written and easy to follow.


**Weaknesses:**

**W1.** The paper's claims would be more accurate with the clarification that HCA refers to the State-HCA algorithm from the HCA paper. The claim that this approach generalizes HCA (e.g., line 103) is inaccurate. HCA generally proposes any function of the future trajectory to estimate contributions, and future states are simply one instance in the proposed family.

**W2.** The observation that State-HCA-based approaches can suffer from high-variance or spurious contributions (even in comparison to Monte-Carlo approaches) is not a novel finding, as it has also been noted in previous work [1,2]. Could the authors clarify the similarities and differences to the observation made in the papers above?

**W3.** A crucial weakness is that it is hard to judge the applicability of the proposed approach based on evaluation in a single environment (linear key-to-door). While linear key-to-door is a challenging problem from the perspective of credit assignment, an evaluation in Atari games (along the lines of Deep HCA [3]) could be more informative in guiding intuitions of how this approach scales to complex settings.

**W4.** The paper also needs to include evaluation against other hindsight baselines, for instance, return-conditioned HCA, which was concretely proposed in the HCA paper. An even more interesting comparison would be to Counterfactual Credit Assignment [4], which learns features for hindsight contribution.


Overall, I appreciate the direction the authors took to address the limitation of State-HCA. Should these weaknesses be adequately clarified/addressed, I would happily increase my score.


—------------------—------------------—------------------—------------------—------------------

### References

[1] Young, K. (2019). Variance Reduced Advantage Estimation with $\delta$ Hindsight Credit Assignment. arXiv preprint arXiv:1911.08362.

[2] Zhang, P., Zhao, L., Liu, G., Bian, J., Huang, M., Qin, T., & Liu, T. Y. (2019). Independence-aware advantage estimation.

[3] Alipov, V., Simmons-Edler, R., Putintsev, N., Kalinin, P., & Vetrov, D. (2021). Towards practical credit assignment for deep reinforcement learning. arXiv preprint arXiv:2106.04499.

[4] Mesnard, T., Weber, T., Viola, F., Thakoor, S., Saade, A., Harutyunyan, A., ... & Munos, R. (2020). Counterfactual credit assignment in model-free reinforcement learning. arXiv preprint arXiv:2011.09464.


**Questions:**

- In Figure 4, why does the COCOA policy not always collect the treasure before the reward switch? Is it due to an exploratory policy? Furthermore, why does the REINFORCE with baseline (Advantage) get much closer to always collecting the treasure in that period?

**Limitations:**

Negative societal impact is not directly applicable. Limitations adequately discussed in the Discussion section.

---

> ### Author Rebuttal · Authors · 2023-08-09
>
> Thank you for your encouraging review and the useful feedback that helped us improve our work.
>
> ## 1. Adjusting the claims of COCOA
> We acknowledge the foundational work of Harutyunyan et al. 2019 in proposing a general class of credit assignment algorithms using hindsight probabilities conditioned on any function of the future trajectory. The main goal of our work is to extend the -theory- of HCA, as the authors only showed theoretical results for the unbiasedness of State-HCA and Return-HCA, leaving open the question of which other functions of the future trajectory lead to unbiased policy gradient estimators, and what impact these functions have on the variance of the estimators. We will clarify this main focus in the manuscript.
>
> ## 2. Comparison to Young 2020 and Zhang 2019
> 1. **Scope of the analysis**: Both works focus their analyses on a specific constructed setting with constant Monte Carlo (MC) returns, no matter which state trajectories are followed. Proposition 2 and its accompanying explanations demonstrate that the high variance of HCA is not just limited to this constructed setting but becomes problematic in a wider range of environments. Interestingly, our work highlights the inherent conflict between obtaining detailed state-representations that enable powerful policies and the resultant high variance in HCA.
> 2. **Estimator focus**: A significant difference lies in which estimator we analyse. While both Young 2020 and Zhang et al 2019 analyse the value estimator, our work studies the policy gradient estimator. This distinction is crucial, as it can lead to different conclusions. Even though a value estimate can be of higher variance compared to the MC value estimate, when combining these value estimates for all actions into the policy gradient estimator, the resulting gradient estimate can be of equal or lower variance compared to its MC equivalent. To illustrate, consider the MDP of Figure 3c in the attached pdf. Here, the return is constant and hence the MC estimator of the Q-value has zero variance, whereas the HCA Q-value estimator does not. However, when comparing the REINFORCE policy gradient estimator with HCA, HCA still demonstrates a reduction in variance, as it accurately estimates that the penultimate state is accessible through both actions in state A, while REINFORCE solely credits the observed action in state A for the entire return. It's worth noting that in scenarios with negative rewards, one can construct toy examples where rewards nullify each other resulting in a constant zero return independent of the state trajectory, which could result in HCA exhibiting higher variance compared to REINFORCE.
> 4. **Solutions to the variance problem**: Both Young 2020 and Zhang et al 2019 propose solutions to the variance issue that continue to utilize the state-based contribution coefficients. This approach does not address the core issue causing the high variance. Given that state representations in realistic environments are often detailed, aligning with Proposition 2, the solutions proposed by these works may not significantly reduce the policy gradient variance when compared to their MC counterparts, in contrast to our low-variance COCOA methods.
>
> In conclusion, while we acknowledge the important work of Young 2020 and Zhang et al 2019, our paper extends and deepens the understanding of the variance issues in the HCA policy gradient estimator and its relationship with state representations. These insights have enabled us to introduce new COCOA estimators with markedly reduced variance.
>
>
> ## 3. Extending COCOA to more environments
> We agree that investigating the scalability of COCOA is an important next step. In this work, however, we concentrated our efforts on a deep analysis of all methods by designing a dynamic programming toolbox, instead of dedicating our time towards investigating the scalability of our methods. This allows us to investigate the credit assignment capabilities of RL algorithms without confounding them with exploration difficulties, as is often the case with performance metrics. While we consider it out of the scope of our work to perform a detailed scalability analysis, we include a new environment compatible with our dynamic programming toolbox, to show that the improved performance of the COCOA methods is not specific to the key-to-door environment (c.f. Figure 2 in the attached pdf)
>
> ## 4. Comparison to new baselines
>
> We now include HCA-return in our experiments, as well as a new method Counterfactual Return that uses returns to construct contribution coefficients (see general rebuttal). Finally, we include Trajectory-wise CV (Cheng, Yan and Boots, 2019), a state-of-the-art method that incorporates hindsight information to construct a near-optimal control variate (see [Answer 1 Rebuttal UxzB](https://openreview.net/forum?id=yvqqkOn9Pi&noteId=P18iyPkGnV)). While we acknowledge that comparing COCOA to Counterfactual Credit Assignment could lead to interesting insights, the source code of this method is not publicly available and we consider it out of scope for our work to reproduce this method consisting of several intrically connected models and losses. We have contacted the lead authors of Mesnard et al. 2021 to request access to their code, to be able to run this baseline.
>
> ## 5. Clarification Figure 4
> The entropy regularization that we use to ensure sufficient exploration has a different effect on the various methods. We perform an independent hyperparameter tuning for all methods and baselines, including the entropy regularization, as detailed in Appendix E.6.3 in the supplementary materials. This led to a hyperparameter configuration for COCOA-reward that solves the environment fast above 90%, but does not solve it 100% due to the entropy regularizer. The tuned configuration for Advantage solves the environment closer to a 100% after 60k episodes, due to a different effect of the entropy regularizer on the exploration behavior.

---

> > ### Comment · Reviewer_5aMg · 2023-08-13
> > **Updated score**
> >
> > Thank you for your response. The clarifications in the response and inclusion of a new environment (and baselines) address some of my main concerns. Accordingly, I have increased my score from 4 to 6.
> >
> > As mentioned in the review (W3), it would still be beneficial to include a more varied range of environments to understand the behaviour of COCOA, especially when learning the contribution coefficients.
> > I recognise the challenge with the code for Counterfactual Credit Assignment not being available. But perhaps comparing with a simpler re-implementation that directly learns features for hindsight contributions would further improve the paper.

---

### Official Review · Reviewer_Embf · 2023-07-12

**Soundness:** 3 good
**Presentation:** 3 good
**Contribution:** 3 good
**Rating:** 6
**Confidence:** 4

**Summary:**

This paper introduces COCOA, a new family of hindsight credit assignment methods that build on HCA, which uses hindsight importance weights for the policy gradient estimator to reduce its variance. HCA uses state-conditioned importance weights, i.e., the ratio of $p(a_t|s_t, s_{t+k})$ to $\pi(a_t|s_t)$. COCOA generalizes HCA by replacing $s_{t+k}$ with $u_{t+k} = f(s_{t+k}, a_{t+k}, r_{t+k})$ with a general function $f$. Specifically, the authors suggest using either $f(s, a, r) = r$ or $f(s, a, r) = g(s, a)$ where $g(s, a)$ is a function that can fully determine the reward. These variants lead to a lower variance compared to vanilla HCA by conditioning only on information that is related to the reward function. They evaluate COCOA on synthetic domains, showing that COCOA outperforms HCA with improved credit assignment.


**Strengths:**

- The authors generalize HCA in an important direction, and the theoretical benefits of COCOA are significant and clear.
- The paper contains informative discussions about various aspects of hindsight gradient estimators, which provides further insights into hindsight credit assignment in general.
- The paper is well-written and easy to follow.

**Weaknesses:**

- The paper lacks discussion regarding return-conditioned HCA. While the authors argue that (state-conditioned) HCA suffers from high variance, the original HCA paper also proposes a return-conditioned variant (Sec 3.2 and Theorem 5 in Appendix B), which can enjoy similar benefits to COCOA. Also, when referring to HCA, the authors only mention its state-conditioned variant throughout the paper. Discussing and comparing with the return-conditioned variant of HCA would be more relevant given its similarity to COCOA-reward.
- The novelty of the method is a bit limited, as the original HCA paper also proposes a general version of HCA that is conditioned on any function of the future trajectory (Sec 3 in the HCA paper). Nonetheless, the paper further discusses sufficient conditions of this function to make the policy gradient estimator unbiased.
- The experiments are conducted only on toy, synthetic environments. It is questionable how COCOA works in more realistic domains.

**Questions:**

- How is COCOA-reward different from HCA-return?
- Given that COCOA-reward shows the lowest variance (Theorem 3), when is COCOA-feature preferred to COCOA-reward?
- Is COCOA-reward always *optimal* in terms of variance, meaning that there are no importance sampling-based gradient estimators with lower variance?
- What would happen if the reward function is a constant function (e.g., $r(s, a) = 1$) but there are terminal signals? Can COCOA-reward directly handle this MDP (without transforming the MDP -- e.g., by creating an additional absorbing state with a zero reward)?

**Limitations:**

The authors adequately address the limitations of COCOA in Section 5.

---

> ### Author Rebuttal · Authors · 2023-08-10
>
> Thank you for your thoughtful comments and the useful feedback that helped us improve our work. We address the points you raised one-by-one below.
>
> ## 1. Comparison to HCA-return
>
> We agree that a dedicated comparison to and discussion of the return-conditioned variant of HCA was missing and we added it now to the manuscript. Notably, we added HCA-return as a baseline to our experiments (see Figure 1 in attached PDF) and discuss its theoretical properties (see joint reply).
>
>
> > The novelty of the method is a bit limited, as the original HCA paper also proposes a general version of HCA that is conditioned on any function of the future trajectory (Sec 3 in the HCA paper). Nonetheless, the paper further discusses sufficient conditions of this function to make the policy gradient estimator unbiased
> ## 2. Contributions compared to HCA
> Although Harutyunyan et al. 2019 proposed HCA as a family of algorithms leveraging hindsight probabilities conditioned on any function of the trajectory, the authors only provided theoretical justifications and experimental results for HCA-state and HCA-return. Hence, our theoretical results for general rewarding outcomes combined with a rigorous empirical investigation of our newly proposed algorithms are important new contributions to the field of credit assignment in RL. In [Answer 1 of rebuttal nRru](https://openreview.net/forum?id=yvqqkOn9Pi&noteId=YPsUl3Q9i1) we provide further details on the contributions of our work.
>
> ## 3. Scalability of COCOA
>
> We agree that investigating the scalability of COCOA is an important next step. In this work, however, we concentrated our efforts on a deep analysis of all methods by designing a dynamic programming toolbox, instead of dedicating our time towards investigating the scalability of our methods. This allows us to investigate the credit assignment capabilities of RL algorithms without confounding them with exploration difficulties, as is often the case with performance metrics. While we consider it out of the scope of our work to perform a detailed scalability analysis, we perform experiments on a new environment compatible with our dynamic programming toolbox, to show that the improved performance of the COCOA methods is not specific to the key-to-door environment. We refer to [the joint reply to all reviewers](https://openreview.net/forum?id=yvqqkOn9Pi&noteId=ga5ZZkiTQt) for further details.
>
>
> ## 4. Comparison with HCA-return
> In the [general post for all reviewers](https://openreview.net/forum?id=yvqqkOn9Pi&noteId=ga5ZZkiTQt) we provide a detailed theoretical and empirical analysis of HCA-return, comparing it to COCOA-reward.
>
>
> ## 5. When is COCOA-feature preferred to COCOA-reward?
>
> <!-- The reviewer points out correctly that COCOA-reward shows the lowest variance, raising the question when COCOA-feature would be preferred over COCOA-reward. -->
> When an oracle provides access to the ground-truth contribution coefficients, COCOA-reward is better-or-equal compared to COCOA-features in MDPs. However, when we need to learn the contribution coefficients, it is often beneficial to use COCOA-features, especially when different rewarding objects have the same scalar reward, but different features. In this case, COCOA-feature can disentangle the rewarding objects, whereas COCOA-reward cannot (c.f. Section 4 in the main text and Fig. 8 in Appendix E). Furthermore, when the environment has many structural regularities, it can help to have more informative rewarding outcome encodings $U$ to leverage latent learning, on which we elaborate in Appendix G. Finally, in a non-MDP setting where the reward function but not the environment dynamics can change throughout training, COCOA-feature allows for quick adaptation (c.f. Fig. 4).
>
> ## 6. Is COCOA-reward optimal?
> While our theoretical and empirical results suggest that the COCOA-reward estimator with ground-truth contribution coefficients has the lowest variance when considering the COCOA and HCA family of methods, we currently cannot make any conclusive claims on the optimality of COCOA-reward. The main two reasons are that (i) Theorem 3 only applies in a limited setting, hence cannot be used to theoretically show the optimality of COCOA-reward in the more general case, and (ii) our work focuses on gradient estimators that use rewarding outcome encodings based on state-action pairs, leaving it to future work to analyse the variance of other hindsight functions of future trajectories, such as the return.
>
> > What would happen if the reward function is a constant function (e.g., $r(s,a) = 1 $) but there are terminal signals? ...
>
> We interpret ‘terminal signals’ as state features indicating the end of a trajectory. If this interpretation is not what the reviewer intended, we kindly ask for a clarification of the question.
>
> COCOA-reward estimates the correct policy gradient in this extreme setting. To gain an intuition, consider the MDP illustrated in Fig 3b in the attached pdf, where each state has a constant reward of 1. Following action 1 results in a longer trajectory compared to action 2, and hence more rewards $r=1$ are encountered after taking action 1. Consequently, the probability $p^\pi(A_0=a_1 \mid S_0=s, R=1) > p^\pi(A_0=a_2 \mid S_0=s, R=1)$, as $\sum_k p^\pi(R_k=1 \mid A_0=a_1, S_0=s) > \sum_k p^\pi(R_k=1 \mid A_0=a_2, S_0=s)$, and hence COCOA-reward reinforces action $a_1$ more than action $a_2$, independently of which trajectory is followed (see also Appendix C.1). When learning the hindsight function $p^\pi(A_0 \mid S_0=s, R=1)$ with a classifier, there will be more triplets $(s,a_1, r=1)$ in the dataset compared to $(s,a_2, r=1)$, hence the learned hindsight function will approximate the true distribution.

---

> > ### Comment · Reviewer_Embf · 2023-08-11
> >
> > Thank you for the detailed response! I believe the discussion about HCA-return (especially about its biasedness) does improve the quality of the paper and raised my score from 5 to 6.

---

### Official Review · Reviewer_nRru · 2023-07-20

**Soundness:** 4 excellent
**Presentation:** 3 good
**Contribution:** 2 fair
**Rating:** 4
**Confidence:** 4

**Summary:**

The paper presents a policy gradient estimator using credit assignment measure on influence of specific actions toward future rewards. The paper offers theoretical analysis on resulting algorithm such as (1) policy gradient estimator is unbiased, (2) variance of the estimator in relation to some existing methods, and so on. The authors also include numerical studies using a simple but specifically designed example to highlight the benefits of the proposed algorithm.

**Strengths:**

1. With some assumptions, the paper provides analytical guarantees on a range of desirable aspects of the algorithm.

**Weaknesses:**

1. Given HCA, contribution seems marginal.
2. Claims are either straightforward or require strong assumptions (quarantees in extreme cases such as Prop 2, Theorem 3)
3. Experiments too simple

**Questions:**

1. Rare but possible that reward-based approach may result in larger variance (e.g., S=(1,2) but r(s,a) \in {-100, 0, 100)).
2. Why did you use only state-conditional HCA? (i.e., not using return-conditional HCA)?
3. In the last paragraph on p.5, it is argued that less informative encodings lead to lower variance. While this is intuitively true, less information also means less learning, which suggests that there should be a trade-off between lower variance and information loss in learning. How would you achieve a good balance?

---

> ### Author Rebuttal · Authors · 2023-08-09
>
> Thank you for the useful feedback that helped us improve our work. We address the points you raised below.
>
> ## 1. Contribution compared to HCA and importance of our theoretical results
> Although Harutyunyan et al. 2019 proposed HCA as a family of algorithms leveraging hindsight probabilities conditioned on any function of the trajectory, the authors only provided theoretical justifications and experimental results for HCA-state and HCA-return. Hence, our theoretical results for general rewarding outcomes combined with a rigorous empirical investigation of our newly proposed algorithms are important new contributions to the field of credit assignment in RL. More specifically:
> - Theorem 1 introduces a general condition (Definition 1) under which hindsight information COCOA and HCA estimators can be used to construct unbiased policy gradients.
> - Proposition 2 provides crucial insights on why HCA-state fails to perform well on many environments of interest, relating it to the high-variance REINFORCE estimator and pointing to a more general issue: state representations need to contain detailed features to enable powerful policies, whereas the same level of detail leads to high variance in HCA-state. These insights combined with our experimental analysis show that HCA-state suffers due to high variance even in environments that significantly violate the strict assumptions of Proposition 2.
> - Theorem 3 is to the best of our knowledge the first theoretical result analyzing the variance of the HCA and COCOA policy gradient estimators, providing important insights on how the information content of hindsight objects influence the variance of the estimator, and suggesting that COCOA-reward results in the lowest-variance estimator. Although Theorem 3 considers only a limited setting, the relative ordering between the methods in terms of variance as predicted by Theorem 3 seems to translate to the general setting of dense rewards and a full policy gradient (c.f. Figure 2 and 3, and Figure 6 in Appendix C).
> - We introduce COCOA-reward and COCOA-features, two practical new algorithms resulting in low-variance gradient estimators with strong performance.
> - We developed a dynamic programming setup which makes it possible to investigate the credit assignment capabilities of RL algorithms without confounding them with exploration difficulties, as is often the case with performance metrics.
> - Specifically, this allows us for the first time to empirically investigate the performance of the ground-truth HCA and COCOA variants (i.e. without approximating models for the hindsight distribution) and further allows us to investigate the bias and variance of the approximate HCA and COCOA methods.
>
>
> ## 2. Extending COCOA to more environments
> We agree that investigating the scalability of COCOA is an important next step. In this work, however, we concentrated our efforts on a deep analysis of all methods by designing a dynamic programming toolbox, instead of dedicating our time towards investigating the scalability of our methods. This allows us to investigate the credit assignment capabilities of RL algorithms without confounding them with exploration difficulties, as is often the case with performance metrics. While we consider it out of the scope of our work to perform a detailed scalability analysis, we include a new environment compatible with our dynamic programming toolbox, to show that the improved performance of the COCOA methods is not specific to the key-to-door environment (c.f. Figure 2 in the attached pdf).
>
> ## 3. Comparison with HCA-return
> We have included new experimental and theoretical results, comparing COCOA-reward to HCA-return. Please refer to the [joint reply for details](https://openreview.net/forum?id=yvqqkOn9Pi&noteId=ga5ZZkiTQt).
>
> ## 4. Tradeoff between variance and information loss in learning
> We thank the reviewer for raising this excellent point. Section 3.3 uses *ground-truth* contribution coefficients, and in this case less informative encodings indeed lead to lower variance. However, when *learning* the contribution coefficients, it could be that less informative encodings lead to slower learning and hence a bigger bias in the contribution coefficients, as fewer observations contain useful information for learning the hindsight distributions. We did not observe this issue in our key-to-door environment, as HCA-state, which has more informative encodings, has a higher bias on average compared to COCOA-reward in Fig. 3. However, in environments where there is more structure present that can be learned from non-rewarding observations, we would expect a trade-off between lower variance and higher bias due to information loss in learning.
>
> From a practitioners perspective, we can navigate this trade-off by considering the information content of the encoding $U$ as a hyperparameter that needs to be tuned for a certain environment. When learning the rewarding outcome features for COCOA-feature, we use a regularization term in the loss to control the information content of the encoding (c.f. appendix E.6.3), which can be tuned as a hyperparameter.
> In Appendix G in the supplementary materials, we further expand upon the interesting question of navigating this trade-off. Our main aim here is to explore strategies that can leverage more state-information to learn the credit assignment models, without causing additional variance. We kindly refer the reviewer to Appendix G for more information about these approaches.
>
> > Rare but possible that reward-based approach may ...
>
> This is an interesting consideration but even in this case, COCOA-reward still has smaller-or-equal variance compared to HCA-state. We assume the one-step MDP depicted in Figure 3a of the accompanying PDF: the agent has 2 actions, which give different transition probabilities to $s_1$ and $s_2$ respectively. This case exactly satisfies the assumptions of Theorem 3, independent of the reward structure.

---

### Official Review · Reviewer_V1Ut · 2023-07-25

**Soundness:** 3 good
**Presentation:** 3 good
**Contribution:** 4 excellent
**Rating:** 8
**Confidence:** 4

**Summary:**

The paper introduces a novel credit assignment algorithm for reinforcement learning known as COCOA (Counterfactual Contribution Analysis). The proposed method extends the concept of Hindsight Credit Assignment (HCA) and aims at making the learning process more sample efficient. By leveraging counterfactual reasoning, COCOA measures the contribution of actions upon obtaining subsequent reward, thereby achieving improved credit assignment. The paper provides an extensive theoretical analysis of COCOA while addressing its potential advantages and limitations.

**Strengths:**

- The proposal of a novel approach to credit assignment in RL.
- Comprehensive theoretical derivations and analysis that support the argument.
- Improvement in policy gradient estimators and progression in long-term credit assignment by distinguishing rewarding outcomes.


**Weaknesses:**

- The absence of empirical comparisons with related methods, particularly RUDDER which is cited in the paper.
- Unclear scalability of the proposed method in more complex environments and restricted quality of the inverse dynamics model.
- Limited empirical evaluation and ablation studies, i.e. examining the influence of the dynamics model on performance.


**Questions:**

- How does COCOA compare to similar SOTA credit assignment methods?
- What is the expected influence of imperfect modelling on the performance of COCOA?
- When employing COCOA, what are best practices for modelling action contributions for ‘rewarding outcomes’ and how does this impact the learning process?


**Limitations:**

- Uncertain scalability to intricate environments.
- Examination on how the choice of the dynamics model impacts COCOA's performance (arguably this is kept open in the discussion section).

---

> ### Author Rebuttal · Authors · 2023-08-09
>
> Thank you for the encouraging review and for the useful feedback that we incorporated in a revised version of our paper. We reply to your questions point-by-point below.
>
> ## 1. Comparison to SOTA credit assignment methods
> We now include HCA-return in our experiments, as well as a new method Counterfactual Return that uses returns to construct contribution coefficients. Finally, we include Trajectory-wise CV (Cheng, Yan and Boots, 2019), a state-of-the-art control variate method that incorporates hindsight information to construct a near-optimal control variate to reduce the variance of the Monte Carlo estimator. The [the joint reply to all reviewers](https://openreview.net/forum?id=yvqqkOn9Pi&noteId=ga5ZZkiTQt) and Answer 1 of [Rebuttal UxzB](https://openreview.net/forum?id=yvqqkOn9Pi&noteId=P18iyPkGnV)  contains detailed information on the new baselines. While we acknowledge that it would be interesting to compare COCOA to the RUDDER method (Arjona-Medina et al. 2019) we consider it out of scope for our current work to integrate RUDDER with our dynamic programming set up.
>
> ## 2. Influence of imperfect modelling on the performance of COCOA
>
> We now include a new experiment investigating this interesting question. We refer to Answer 2 of [Rebuttal UxzB](https://openreview.net/forum?id=yvqqkOn9Pi&noteId=P18iyPkGnV)  for further details.
>
> ## 3. Best practices for modelling action contributions for ‘rewarding outcomes’
>
> When considering which rewarding outcome encoding $U$ to use, there are 2 main forces to take into consideration. First, section 3.3 argues that taking the reward as rewarding outcome encoding results in the lowest variance estimator, when we have access to the ground-truth contribution coefficients from an oracle. Second, when we need to learn the contribution coefficients, taking the rewards as rewarding outcome encodings is not always the best choice, as the rewards do not contain a lot of information about the underlying environment structure, which can make it harder to learn good hindsight distributions. Especially when multiple rewarding objects have the same scalar reward, this can pose problems to COCOA-reward, favoring COCOA-features that learns a rewarding object representation instead (c.f. Appendix E.7.1 in the supplementaries for an empirical evaluation with reward aliasing).
>
> For environments with simple dynamics and without reward aliasing we recommend using COCOA-reward, as it is uses a straight-forward encoding $U=R$ and leads to good results in practice.
> In environments where one expects the state space to contain useful information for learning the hindsight distribution, or where reward aliasing can occur, there can be a trade-off when choosing the information content of $U$. Indeed, a lower information content of $U$ leads to a lower variance, but can lead to a higher bias in the hindsight model as there is less information available to leverage structure of the environment dynamics for efficient learning.
>
> From a practitioners perspective, we can navigate this trade-off by considering the information content of the encoding $U$ as a hyperparameter that needs to be tuned for a certain environment. When learning the rewarding outcome features for COCOA-feature, we use a regularization term in the loss to control the information content of the encoding (c.f. appendix E.6.3), which can be tuned as a hyperparameter.
> In Appendix G in the supplementary materials, we further expand upon the interesting question of navigating this trade-off. Our main aim here is to explore strategies that can leverage more state-information to learn the credit assignment models, without causing additional variance.
>
> We highlight three interesting avenues:
> 1. Learn a state-based hindsight distribution, as in HCA-state, which can leverage detailed state information, but use counterfactual reasoning on rewarding states to recombine the state-based hindsight distributions into a reward-based hindsight distribution, as in COCOA-reward (c.f. Appendix G.3). Intuitively, this approach evaluates the question “could I have gotten the same reward in different states as well?”
> 2. Use a variational information bottleneck approach to explicitly control the information content of the encodings, which can also be combined with the previous point (c.f. Appendix G.4).
> 3. Instead of learning backward hindsight distributions $p(a \mid s, u)$, learn forward environment dynamics $p(s’ \mid s,a)$ and $p(u \mid s,a)$ and use this to compute the contribution coefficients using Eq. 3. Learning the forward environment dynamics can leverage the detailed state-information, hence bypassing the variance trade-off (c.f. Appendix G.2)
> We kindly refer the reviewer to Appendix G for more information about these approaches.
>
>
> ## 4. Uncertain scalability to intricate environments
>
> We agree that investigating the scalability of COCOA is an important next step. In this work, however, we concentrated our efforts on a deep analysis of all methods by designing a dynamic programming toolbox, instead of dedicating our time towards investigating the scalability of our methods. This allows us to investigate the credit assignment capabilities of RL algorithms without confounding them with exploration difficulties, as is often the case with performance metrics. While we consider it out of the scope of our work to perform a detailed scalability analysis, we perform experiments on a new environment compatible with our dynamic programming toolbox, to show that the improved performance of the COCOA methods is not specific to the key-to-door environment. We refer to [the joint reply to all reviewers](https://openreview.net/forum?id=yvqqkOn9Pi&noteId=ga5ZZkiTQt) for further details.

---

> > ### Comment · Reviewer_V1Ut · 2023-08-14
> > **Updated score**
> >
> > I acknowledge the revisions made by the authors. I appreciate the time and effort they have put into addressing the points raised in my initial review. Therefore, I revise my evaluation score to reflect the improvements made.

---

### Official Review · Reviewer_r14e · 2023-07-26

**Soundness:** 3 good
**Presentation:** 4 excellent
**Contribution:** 3 good
**Rating:** 6
**Confidence:** 1

**Summary:**

This paper proposes Counterfactual Contribution Analysis (COCOA) to improve credit assignment in the reinforcement learning problem by building upon Hindsight Credit Assignment.

**Strengths:**

This paper is written in a clear manner and allows non-expert audiences to understand its difference and contribution in contrast to previous work.

**Weaknesses:**

I do not feel that I understand enough about this field to critique this paper.

**Questions:**

A space is missing between policy and [15-17] on line 41.

**Limitations:**

Seems okay to me.

---

> ### Author Rebuttal · Authors · 2023-08-09
>
> We thank the reviewer for his review and pointing us to the typo.

---

### Official Review · Reviewer_GAJb · 2023-08-02

**Soundness:** 3 good
**Presentation:** 3 good
**Contribution:** 3 good
**Rating:** 7
**Confidence:** 3

**Summary:**

This paper presents family of algorithms namely Counterfactual Contribution Analysis (COCOA), which is based on measuring contribution of an action by asking counterfactual question. The new measure of contribution performs better than existing methods such as HCA in terms of lower variance. They define rewarding outcome as function of state, action and reward. They compute contribution coefficients which can be used to learn a policy with COCOA policy gradient estimator.

**Strengths:**

1. The authors show that the presented COCOA policy gradient estimator can result in lower variance than existing method such as HCA, which suffers from spurious contributions which can be alleviated by using less informative encodings.
2. The authors demonstrate demonstrate the performance of COCOA compared to standard baselines in key-to-door environment. They further present improvement in sample efficiency due to favorable bias-variance trade off
3. The way COCOA is learning the correct distanglement enables long-term credit assignment better than existing methods.

**Weaknesses:**

Compared to policy gradient methods like REINFORCE and Advantage, the COCOA estimators seem to be very computationally expensive.

**Questions:**

1. Could you please highlight how expensive it is to compute these contribution coefficients.
2. Policy gradient methods have shown great performance in continuous state and continuous action MDPs. How can the coefficients be computed in such a setting? Or the proposed method is limited to tabular setting?
3. In many real world applications, we only have sparse rewards, how does the performance of these estimators vary between dense and sparse reward scenarios?

**Limitations:**

-

---

> ### Author Rebuttal · Authors · 2023-08-09
>
> Thank you for your positive review and instructive suggestions that have helped to improve the paper. Below we provide responses to your raised questions and concerns.
>
> ## Computational cost
>
> We provide the wall clock time of all methods after 3000 update steps on the linear key-to-door environment (averaged over three seeds) on our hardware (Nvidia RTX 3090) in the following table.
>
> | Method        | Wall clock time (seconds) |
> |---------------|---------------------------|
> | COCOA-reward  | 343.67                    |
> | COCOA-feature | 434.67                    |
> | HCA+          | 319.33                    |
> | Advantage     | 314.00                    |
> | Q-critic      | 319.00                    |
> | REINFORCE     | 330.33                    |
> | TrajCV        | 328.00                    |
>
> During training, value based methods such as Q-critic and Advantage have a complexity linear in the environment length, considering the amount of evaluations of the policy network and value network. The COCOA and HCA methods have a quadratic complexity in the environment length. Despite this, except for COCOA-feature, all methods have similar runtime, suggesting the computational bottleneck lies in the environment simulation in our case.
>
> ## Continuous state and action spaces
> In appendix F.1 and F.4 in the supplementary materials we expand upon this question of how COCOA can be applied to continuous state and action spaces. In short, the contribution coefficients of Equation 3 readily extend towards continuous state and action spaces, by using probability density functions $p(a\mid s, u)$ and $p(u \mid s,a)$ instead of probability distributions as in the discrete case. To construct a policy gradient estimator, we either need to replace the sum over actions in Equation 4 by an integral over the action space, or sample from it resulting in
> $$\sum_{t\geq0} \nabla_\theta \log \pi(A_t \mid S_t) R_{t} + \frac{1}{M} \sum_m \nabla_\theta \log \pi(a^m\mid S_t)\sum_{k=1}^{\infty} w(S_{t}, a^m, U_{t+k}) R_{t+k}$$
> with $M$ the number of samples. We refer the reviewer to Appendix F for a more detailed discussion.
>
> ## Difference between sparse and dense reward scenarios
> In sparse reward settings, there are two main challenges: (i) efficient exploration of the environment to discover the sparse rewards and (ii) making efficient use of the sparse rewards to update the policy. These exploration and credit assignment challenges are mostly orthogonal, and in our work and experiments we focus on the credit assignment problem, assuming that the exploration challenge is solved to a sufficient extend. The COCOA methods particularly excel when long-term credit assignment is needed, as the contribution coefficients can reduce the variance of the policy gradient estimator without the need to resort to temporal discounting. The key-to-door environment investigates the credit assignment problem with sparse treasure rewards, with the added difficulty of a distractor task introducing noise. Hence, we expect that our key-to-door results translate to other sparse reward settings where the exploration problem is solved sufficiently. In a dense reward setting where no long-term credit assignment is needed, we expect that the Advantage and Q-net baselines will move closer towards the performance of our COCOA methods, as in this case weaker credit assignment techniques are likely sufficient in combination with temporal discounting.

---

### Author Rebuttal · Authors · 2023-08-09

We thank all reviewers for their constructive comments and useful suggestions that we believe have helped us to significantly improve our paper. Here we summarize the changes we did addressing the main concerns of the reviewers, linking to the corresponding detailed responses.

## Main changes

1. Reviewers nRru, Embf and 5aMg asked for a comparison with HCA-return. We now include HCA-return in our experimental results, showing that it performs similarly to HCA-state and significantly lags behind our COCOA methods and other baselines. Furthermore, we provide new theoretical results showing that HCA-return with ground-truth hindsight distributions is a biased estimator in many relevant environments. See below for a detailed discussion.
2. Reviewers 5aMg, UxzB, V1Ut, nRru and Embf asked to introduce extra baselines to compare COCOA against state-of-the-art credit assignment methods. We included three new baselines: (i) Trajectory-wise Control Variate (Cheng, Yan and Boots, 2019; TrajCV), a SOTA control variate method to reduce the variance of MC estimators, (ii) HCA-return as explained above, and (iii) Counterfactual Return, a new baseline investigating the difference between taking rewards or returns as hindsight information for constructing contribution coefficients. See below and [Answer 1 Rebuttal UxzB](https://openreview.net/forum?id=yvqqkOn9Pi&noteId=P18iyPkGnV) for details.
3. Reviewers UxzB and V1Ut asked for additional experiments investigating the influence of the hindsight model accuracy upon the gradient estimator quality. We now include new experimental results investigating this question, showing that COCOA-reward remains performant for significantly biased models, compared to the other baselines. See Answer 2 in [Rebuttal UxzB](https://openreview.net/forum?id=yvqqkOn9Pi&noteId=P18iyPkGnV) for details.
4. Reviewers 5aMg, V1Ut, nRru and Embf raised the concern that our experiments are focused around a single environment. We have added an additional environment based on Mesnard et al. (2021), see below for details.


## Comparison with HCA-return

### Difference between HCA-return and COCOA-reward
The first main differences between HCA-return and COCOA-reward are that (i) HCA-return uses a hindsight function conditioned on the return instead of individual rewards, and (ii) HCA-return leverages the hindsight function as an action-dependent baseline for a Monte Carlo policy gradient estimate, instead of using it for contribution coefficients to evaluate counterfactual actions:
$\sum_{t\geq 0} \nabla_\theta \log \pi(A_t \mid S_t)\Big(1 - \frac{\pi(A_t \mid S_t)}{p^\pi(A_t \mid S_t, Z_t)}\Big)Z_t$

### HCA-return is a biased estimator in many relevant environments.
An important drawback of HCA-return is that it can be biased, even when using the ground-truth hindsight distribution. Theorem 2 of Harutyunyan et al. 2019 considering the unbiasedness HCA-return is valid under the assumption that for any possible random return $Z$ for all possible trajectories starting from state $s$, it holds that $p^\pi(a \mid s,z) > 0$. This restrictive assumption requires that for each observed state-action pair $(s_t, a_t)$ along a trajectory, all counterfactual returns $Z$ resulting from a counterfactual trajectory starting from $s_t$ (not including $a_t$) result in $p^\pi(a_t \mid s_t, Z) > 0$. This implies that all returns (or rewarding states) reachable from $s_t$ should also be reachable from from $(s_t, a_t)$.

Consider the following bandit setting as a simple example where the above assumption is not satisfied. The bandit has two arms, with a reward of $1$ and $-2$, and a policy probability of $\frac{2}{3}$ and $\frac{1}{3}$ respectively. The advantage for both arms is $1$ and $-2$. Applying eq. 6 from Harutyunyan et al. results in $A^\pi(s,a_1) = (1 - \frac{2}{3})=\frac{1}{3}$ and $A^\pi(s,a_2) = -2(1 - 1/3)=-4/3$. This shows that the needed assumptions for an unbiased HCA-return estimator can be violated even in simple bandit settings.

### Using returns instead of rewards as hindsight information
To isolate the difference of using returns instead of rewards as hindsight information for constructing the contribution coefficients, we introduce the following new baseline:
$\sum_{t\geq 0} \sum_{a} \nabla_\theta \pi(a \mid S_t)\Big(\frac{p^\pi(a \mid S_t, Z_t)}{\pi(a \mid S_t)} -1\Big)Z_t$.
Similar to HCA-return, this *Counterfactual Return* variant leverages a hindsight distribution conditioned on the return. Different from HCA-return, we use this hindsight distribution to compute contribution coefficients that can evaluate all counterfactual actions. We prove that this estimator is unbiased.


### Empirical comparison HCA-return and Counterfactual Return

Figure 1 in the attached pdf shows that the performance of HCA-return and Counterfactual return lags far behind the performance of COCOA-reward, both in a setting with ground-truth hindsight distributions and learned hindsight distributions. This is due to the high variance of HCA-return and Counterfactual Return, and the biasedness of the former. As the return is a combination of all the rewards of a trajectory, it cannot be used to disentangle rewarding outcomes, causing the variance of the distractor subtask to spill over to the subtask of picking up the treasure.

## New environment
To show that the improved performance of the COCOA methods is not specific to the key-to-door environment, we include a stylized version of the Task Interleaving environment of Mesnard et al. 2021. Strong credit assignment is needed to solve this task, due to the delayed rewards.
Figure 2 in the attached pdf describes the environment and shows the results for our main methods and baselines. We will include the remaining baselines, ground-truth performance and bias/variance analyses in the final manuscript.

---

### Decision · Program_Chairs · 2023-09-21

**Decision:**

Accept (spotlight)

**Comment:**

Based on the reviews provided by the reviewers, I recommend accepting the paper. The proposed Counterfactual Contribution Analysis (COCOA) approach addresses limitations of existing methods, particularly the State-Hindsight Credit Assignment (HCA) variant, by introducing rewarding outcomes and disentangling action contributions using various features. The theoretical analysis and experiments demonstrate its potential in reducing variance and improving credit assignment in reinforcement learning. While some weaknesses and limitations are acknowledged, the paper's contributions are technically sound, and it shows promise in improving sample efficiency and addressing issues with credit assignment. Therefore, I recommend accepting the paper as it offers a novel approach with moderate-to-high impact in the field.